# Improving Detection of Watermarked Language Models

**Dara Bahri, John Wieting**
*Google DeepMind*

**Reviewed on OpenReview:** *https://openreview.net/forum?id=6nmztBgngB*

## Abstract

Watermarking has recently emerged as an effective strategy for detecting the generations of large language models (LLMs). The strength of a watermark typically depends strongly on the entropy afforded by the language model and the set of input prompts. However, entropy can be quite limited in practice, especially for models that are post-trained, for example via instruction tuning or reinforcement learning from human feedback (RLHF), which makes detection based on watermarking alone challenging. In this work, we investigate whether detection can be improved by combining watermark detectors with *non-watermark* ones. We explore a number of *hybrid* schemes that combine the two, observing performance gains over either class of detector under a wide range of experimental conditions.

## 1 Introduction

General usage of large language models (LLMs) has increased dramatically in recent years, and so has the need for identifying texts generated by LLMs, or AI-generated content (AGC), in the wild. For example, an academic institution may wish to know whether students are using an LLM to do their assignments, or an LLM provider may want to understand where and how their model is being used. One may want to detect whether the text was generated by a *specific* model or *any* model. Moreover, in the former case, the detecting party may or may not have white-box access (concretely, an ability to compute log-probabilities) to the model they wish to test against. Parties with white-box access are typically the model's owners, and so we refer to this setting as *first-party* (1P) detection. In contrast, we refer to the detection of one or more models where white-box access is not available as *third-party* (3P) detection.

The aim of watermarking is to bias the model so that first-party detection becomes more tractable. Most schemes do not modify the LLM's training procedure but instead inject the watermark signal at inference time as part of the autoregressive decoding loop (Aaronson, 2023; Bahri & Wieting, 2024; Kuditipudi et al., 2023; Kirchenbauer et al., 2023; Dathathri et al., 2024). Meanwhile, non-watermark detection has mostly been viewed as a binary classification task, with the goal of discriminating one class from another — typically human-written text from text authored by one or more models. Strategies here include training a binary classifier on labeled text samples or computing uncertainty-based scores, such as likelihood, under a specific LLM (Zellers et al., 2019; Solaiman et al., 2019; Gehrmann et al., 2019; Su et al., 2023; Mitchell et al., 2023). The focus of our work is on improving first-party detection by intelligently combining watermark- and non-watermark-based detection approaches.

The entropy of an LLM's distribution over possible responses conditioned on a specific prompt is a key quantity in watermark detection. For example, Bahri & Wieting (2024) provides a lower-bound on the detection ROC-AUC as a function of the entropy of the sampled next-token distribution. In all schemes, watermark performance improves with more available entropy. As a concrete example, consider two input prompts to a watermarked LLM: (1) *"Where is Mount Whitney?"* and (2) *"Write me a dreamy haiku about the John Muir Trail in California."* The first prompt is a fact-based inquiry whereas the second is more open-ended in nature, so the entropy will likely be higher for the second prompt than for the first and the generated haiku will be easier to detect via the watermark. The role of entropy for non-watermark detection is less studied. We show how non-watermark detectors, when configured in a hybrid setup, can substantially

bolster watermarking in low entropy regimes. For example, training a logistic regression model on watermark scores and scores from a RoBERTa-based AGC classifier can boost watermarking test accuracy from 75% to over 95%, a 20% absolute improvement, on the 20% of test prompts that afford the *least* amount of entropy. We study how these two approaches to detection can be combined effectively for better predictive performance than either part as well as computational advantages, and we recommend practical algorithms for deployment.

## 2 Related Work

We provide a concise overview of prior work on watermarking and AGC detection.

### 2.1 Watermarking

Although watermarking, sometimes referred to as linguistic steganography, has a storied past, interest in its application to modern large language models grew substantially after the seminal works of Kirchenbauer et al. (2023) and Aaronson (2023). Many successful techniques employ pseudo-random functions (PRFs) and cryptographic hashes and are applied token-by-token during the autoregressive decoding loop. Kirchenbauer et al. (2023) uses a PRF to bias the next-token probability distribution so that certain tokens known only to the watermarker are made more probable. The degree of bias is a hyper-parameter that trades off text distortion and watermark strength. Aaronson (2023) proposes a clever strategy that selects the token in the next-token distribution that has a high PRF value and a high likelihood in a way that makes the process appear distortion-free. Kuditipudi et al. (2023) applies a scheme similar to Aaronson (2023) but to increase robustness to attacks such as paraphrasing, the pseudo-random numbers are determined by cycling through a secret, predetermined sequence of values rather than by $n$-grams. Meanwhile, SYNTHID (Dathathri et al., 2024) samples a large number of tokens from the next-token distribution and pits them head-to-head in a series of tournaments, the winner of which is selected as the next token. Lee et al. (2023) adapts Kirchenbauer et al. (2023)'s scheme for the code-generation task by applying the watermark only at decoding steps that have sufficient entropy.

Black-box schemes that only require a way to sample sequences from the LLM have been devised (Yang et al., 2023a; Giboulot & Furon, 2024; Chang et al., 2024; Bahri & Wieting, 2024). Most notably, Bahri & Wieting (2024) uses elements from Aaronson (2023) to construct a general framework for black-box distortion-free watermarking, which can be used effectively when white-box access is available.

Ways of evading, spoofing, and even stealing watermarks have been extensively studied (Zhang et al., 2023; Gu et al., 2023; Jovanović et al., 2024; Cheng et al., 2025). To address the weakness of many schemes to attacks such as token substitution or paraphrasing, watermarking based on semantics or invariant features has also been proposed (Liu et al., 2023; Hou et al., 2023; Ren et al., 2023; Yoo et al., 2023). Fernandez et al. (2023) tests various watermarking schemes on classical NLP benchmarks and introduces new statistical tests — for example, they suggest skipping duplicate $n$-grams during testing. Liang et al. (2024) conducts a comprehensive assessment of watermarking strategies, finding that incorporating a non-watermark RoBERTa-based detector like the way we do can help boost robustness to attacks. The purpose of our work is to generalize this focused insight by proposing and then carefully evaluating various hybrid schemes.

### 2.2 AGC Detection

A powerful paradigm to detect AGC is to finetune a pretrained LM on sizable datasets for classification (Zellers et al., 2019; Solaiman et al., 2019); notably, Hu et al. (2023) and Tian et al. (2023) do so via adversarial and positive-unlabeled (PU) learning respectively, while Tay et al. (2020) applies it to the task of discriminating different LM generators from one another.

A second camp of approaches focuses on capturing statistical patterns of AGC using little to no training data — such as the intrinsic dimensionality of generated text (Tulchinskii et al., 2024), perplexity / rank / log-rank (Gehrmann et al., 2019; Su et al., 2023), perplexity curvature (Mitchell et al., 2023), and $n$-gram patterns (Yang et al., 2023b).

Mireshghallah et al. (2024) shows, interestingly, that small LM generators make for better zero-shot detectors than larger ones. Mao et al. (2024) leverages the observation that LLMs change more words when tasked to rewrite human text than to rewrite AGC. Zhang et al. (2024) shows that existing detectors are inadequate for detecting *AI-revised human-written text*, text that a human wrote with assistance from an LLM. Interestingly, Russell et al. (2025) found that humans who use LLMs for writing tasks can outperform many commercial and open source detectors even in the presence of paraphrasing and humanization.

As usage of large language models is still nascent, it remains to be seen precisely how they are used in the wild, be it with good or nefarious intentions. Bahri et al. (2021) conducts a large-scale study to this end by running a GPT-2 detector on half a billion web pages.

Ways to evade detection have been studied. For example, both Lu et al. (2023) and Kumarage et al. (2023) propose a prompting strategy that encourages LLMs to produce more human-looking text, while Krishna et al. (2024) presents a paraphrasing model called DIPPER that can successfully fool detectors.

## 3   Experimental Setup

The task of detection is not without inherent ambiguities. For example, suppose an LLM regurgitates text from its (human) training corpus verbatim — should we consider this sample human or AGC? Arguments can be made for both sides; for the purposes of our work, we consider it AGC.

As watermarking is most commonly studied in the 1P setting, our evaluation is restricted to the 1P setting as well. However, we consider both 1P and 3P detectors in our hybrid scheme, as even a 3P detector can provide lift for 1P detection when the negative class consists of human samples. We treat the problem as a binary classification task, where positive samples come from *our* model, denoted $M_o$, and negative samples are either human or generations from a different model, denoted $M_t$ (*theirs*). We report performance metrics for both the human and model negative classes separately, a distinction often missing in prior works. Each dataset consists of prompts and human responses. For each prompt, we generate responses under $M_t$ and $M_o$, where the latter is equipped with various watermarking methods. We now discuss our choice of datasets, watermarks, and detection strategies.

### 3.1   Models, Datasets, Hyperparameters, and Compute

We consider two distinct settings: when $M_o$ is GEMMA-7B-INSTRUCT (Team et al., 2024b)[1] and $M_t$ is MISTRAL-7B-INSTRUCT (Jiang et al., 2023)[2] as well as the reverse assignment.

We use two test datasets: *databricks-dolly-15k*[3] (Conover et al., 2023), an open source dataset of instruction-following examples for brainstorming, classification, closed QA, generation, information extraction, open QA, and summarization. Exactly replicating Bahri & Wieting (2024), we use prompts from the brainstorming, generation, open QA (i.e. general QA), and summarization categories, whose human responses are at least 50 tokens long (save one example, which was removed because the prompt was extremely long); this amounts of 5,233 prompts total.

We use the training and test splits of *eli5-category*[4] (Fan et al., 2019). Prompts are formed by concatenating the *prompt* and *selftitle* fields. Only examples whose *prompt* field is non-empty and contains a *?* are kept — for a total of 83,089 train and 4,855 test samples.

We always decode by random sampling with temperature 1 and do not employ top-$p$ or top-$k$ strategies. For each prompt in both test datasets, we generate four non-watermarked responses, along with a watermarked one under each scheme. We always force a minimum (maximum) of 250 (300) new tokens by disabling the stop token for the first 250 tokens, re-enabling it, and stopping the generation at 300, regardless of whether the stop token was encountered. For both the train and test split of *eli5-category*, the human response is

---

[1]https://huggingface.co/google/gemma-7b-it
[2]https://huggingface.co/mistralai/Mistral-7B-Instruct-v0.1
[3]https://huggingface.co/datasets/databricks/databricks-dolly-15k
[4]https://huggingface.co/datasets/rexarski/eli5_category

taken to be the one with the highest score. Also, for the train split only, we run non-watermarked generation enforcing a maximum of 300 tokens but no minimum number of tokens.

To simulate real-world use, we de-tokenize the outputs to obtain plain text, and re-tokenize them during scoring. We study performance as a function of token length $T \leq 250$ by truncating samples to their first $T$ tokens.

Experiments were run on 80GB A100 or H100 GPUs using PyTorch and HuggingFace's Transformers models with bfloat16 quantization for a total compute cost of about 2,000 GPU hours.

## 3.2 Entropy

Entropy is a key quantity in our analysis. With $x$ denoting the input prompt and $y \sim P_M(\cdot \mid x)$ the sampled response of length $T$ from model $M$, the response entropy $H(x)$ is,

$$\mathbb{E}_y - \log P_M(y \mid x) = \mathbb{E}_y \sum_{i=1}^{T} - \log P_M(y_i \mid y_{<i}, x) = \sum_{i=1}^{T} \mathbb{E}_{y_{<i}} \underbrace{\underbrace{\mathbb{E}_{y_i \mid y_{<i}} - \log P_M(y_i \mid y_{<i}, x)}_{=H_+(x, y_{<i})}}_{=H_i(x)},$$

where $H_+(x, y_{<i})$ is the entropy of the next-token distribution given prompt $x$ and partial response $y_{<i}$ (of length $i-1$) and $H_i(x)$ is this quantity averaged over partial responses. In other words, $H_i(x)$ is the entropy of the next-token distribution for token position $i$ that we see on average.

If we condone sample reuse for the sake of computational efficiency, then $H_i(x)$ and $H(x)$ can be estimated with i.i.d. samples $\{y^1, \ldots, y^n\}$ from $P_M(\cdot \mid x)$ as,

$$\hat{H}_i(x) = \frac{1}{n} \sum_{j=1}^{n} H_+(x, y_{<i}^j), \quad \text{and} \quad \hat{H}(x) = \sum_{i=1}^{T} \hat{H}_i(x).$$

We bucket prompts $x$ in our test set by their estimated response entropy $\hat{H}(x)$ (where the responses used for the estimate are sampled from the model sans watermarking) and analyze detection performance for each bucket separately so as to directly observe the role of entropy on performance. We choose $T = 100$ and $n = 4$.

## 3.3 Watermarks

The watermark schemes we consider here operate at a token level. To facilitate describing them below, let $p$ be the next-token probability distribution. Moreover, if $F$ is a cumulative distribution function (CDF), let $F[s]$ be a single draw from a pseudorandom number generator (PRNG) for $F$ seeded with integer seed $s$, $F(x)$ F evaluated at $x$, and $F_k$ the CDF for the sum of $k$ i.i.d. random variables where each is distributed $F$. Furthermore, let named distributions represent their CDFs; for example, $U(0, 1)(0.5)$ is the CDF for the standard uniform distribution evaluated at 0.5. Let our LLM have vocabulary $\mathcal{V}$ of size $V$ and $h$ be a cryptographic hash function (e.g. SHA-256) from $\mathbb{Z}^*$ to $\mathbb{Z}$. $K \in \mathbb{Z}$ is used to denote the *secret* integer key that is known only to the watermarking party. For both watermark and non-watermark detection, higher scores indicate higher confidence that the test query is watermarked / AGC.

**Aaronson**. At each step, Aaronson (2023) computes a pseudorandom number for each token $i \in \mathcal{V}$ as, $u_i = U(0, 1)[h(K|w|i)]$, where $w$ is the preceding $(n-1)$-gram, and $|$ denotes concatenation. Token $i^*$ is selected, where $i^* = \text{argmax}_i u_i^{1/p_i}$. At test time, $n$-grams $\{w_i\}_{i=1}^{T}$ are extracted from the test query and the detection score is given by,

$$s_A = - \sum_{i=1}^{T} \log(1 - R_i), \quad \text{where} \quad R_i = U(0, 1)[h(K|w_i)].$$

Bahri & Wieting (2024) notes that $s_A$ is not length-aware so that a single decision threshold across scores involving various lengths results in poor performance. To remedy this, they propose *length-aware* score

$s_{AC} = \text{Gamma}(T, 1)(s_A) = \chi^2_{2T}(2s_A)$. We use this length-aware score with $n$ set to 4; this choice strikes a good balance between generation quality / diversity and robustness to attacks.

**Bahri**. Bahri & Wieting (2024) recently proposed a distortion-free watermarking scheme that requires only black-box access to an LLM. The scheme works by autoregressively sampling $m$ sequences $\{Q_i\}$ from the LLM, each roughly $k$ tokens long. Let $\{X_i\}$ be the set of *unique* sequences from $\{Q_i\}$, $W_n(X)$ represent the set of $n$-grams for sequence $X$, and $S_i$ the set of seeds $\{h(K|w)\}_{w \in W_n(X_i)}$ that have had duplicates across the $X_i$'s removed and where a random unseen seed is added, if necessary, to ensure all $S_i$'s are nonempty. Then $X_{i^*}$ is returned, where,

$$i^* = \text{argmax}_i u_i^{m/c_i}, \quad \text{and} \quad c_i = \sum_{j=1}^m \mathbf{1}[Q_j = X_i], \quad \text{and} \quad u_i = F_{|S_i|}\left(\sum_{s \in S_i} F[s]\right).$$

The score $s_B$ for sequence $X$ with *unique* (i.e. de-duplicated) seeds $S = \{h(K|w)\}_{w \in W_n(X)}$ is given as $s_B = F_{|S|}\left(\sum_{s \in S} F[s]\right)$. We use the settings optimal for 1P detection where white-box access *is* available — their *flat* scheme with sequence length $k = 1$ and $F = U(0, 1)$. We set $m = 1024$ and $n = 4$.

**Kirchenbauer**. Kirchenbauer et al. (2023) uses the last $n$ tokens to pseudorandomly partition the vocabulary for the next token into two lists: a green list of size $\gamma V$ and a red list consisting of the remainder. A positive bias of $\delta$ is added to the logits of the green list tokens while those of the red list are left unchanged. This has the effect of modifying $p$ so that green list tokens are more probable. The score for a text consisting of $T$ tokens, $T_g$ of which were found to be green is $s_{KB} = (T_g - \gamma T)/\sqrt{T\gamma(1 - \gamma)}$. We incorporate the latest updates to the algorithm,[5] such as including the current token in the $n$-gram and skipping duplicate $n$-grams at test time. The scheme modifies the model probability so it's not distortion-free, but a good balance between watermark strength and generation quality can often be achieved. We set $n = 4$, $\gamma = 0.25$, and $\delta \in \{0.5, 2, 3\}$.

**Kuditipudi**. Using the last $n$ tokens as the basis for the PRF has the downside that modifying just one of the tokens changes the output and subsequently hurts detection. Kuditipudi et al. (2023) addresses this limitation, which we describe in detail in the Appendix. We follow their methodology and use 256 seeds. To expedite the permutation test, we precompute 5,000 reference values for the secret list. These values are obtained by sampling snippets from the training set of *C4-realnewslike* (Raffel et al., 2019) across all evaluated target lengths.

### 3.4 Detectors

We consider the following competitive 1P and 3P detection strategies. We omit techniques like Detect-GPT (Mitchell et al., 2023), Ghostbuster (Verma et al., 2023), and DNA-GPT (Yang et al., 2023b) as the baselines covered here, like RADAR (Hu et al., 2023) and Binoculars (Hans et al., 2024), were shown to outperform them.

**Log-Likelihood and Rank**. We compute per-token log-likelihood (LLh) and rank features of the target text under a model $M$ (taken to be our model, $M_o$), as done in prior work (Gehrmann et al., 2019). We note that when scoring a response $y$, standard practice throughout the literature has been to use $P_M(y \mid \phi)$ as the likelihood, where $\phi$ is the null or empty prompt. This is incorrect; the true likelihood is: $P_M(y) = \mathbb{E}_{x \sim f_x} P_M(y \mid x)$, where $f_x$ is the distribution over prompts. Since the focus of this work is not on new detection techniques but in novel combinations of existing watermark and detection strategies we continue to define likelihood using the empty prompt. Our features are then:

$$L_M(y) = \frac{1}{|y|} \sum_{i=1}^{|y|} \log P_M(y_i \mid y_{<i}). \quad R_M(y) = \frac{1}{|y|} \sum_{i=1}^{|y|} R_M(y_i \mid y_{<i})$$

---

[5] https://github.com/jwkirchenbauer/lm-watermarking

$R_M(y_i \mid y_{<i})$ is the absolute rank of $y_i$ in $P_M(\cdot \mid y_{<i})$ — the rank is 1 ($V$) when $y_i$ is the most likely (least likely) token in the next-token distribution. Meanwhile, $L_M$ represents the log-likelihood. AGC should have high likelihood and low rank.

**DetectLLM**. Su et al. (2023) proposes a novel combination of zero-shot likelihood and rank features. Specifically, they propose the *Log-Likelihood Log-Rank Ratio* (LRR) as:

$$\mathrm{LRR}_M(y) = -\frac{\sum_{i=1}^{n} \log P_M(y_i \mid y_{<i})}{\sum_{i=1}^{n} \log R_M(y_i \mid y_{<i})},$$

where $M_o$ is used for $M$.

**Binoculars**. Hans et al. (2024) proposes a 3P detection scheme that leverages uncertainty scores from two LLMs. The Binoculars score is:

$$B_{M_1, M_2}(y) = \frac{\sum_{i=1}^{|y|} \log P_{M_1}(y_i \mid y_{<i})}{\sum_{i=1}^{|y|} P_{M_1}(\cdot \mid y_{<i})^T \log P_{M_2}(\cdot \mid y_{<i})}.$$

We use the recommended settings: FALCON-7B-INSTRUCT for $M_1$ and FALCON-7B for $M_2$.

**RADAR**. Hu et al. (2023) proposes an adversarial technique to jointly train an AGC detector with a paraphraser. The paraphraser is trained to generate content that will evade the detector while the detector is trained to discern the paraphraser's outputs. They show the method outperforms baselines on 4 datasets and 8 LLMs. We use their open-source RoBERTa-large based detector.[6]

**RoBERTa Classifier**. As done in prior work, we fine-tune a RoBERTa (Liu et al., 2019) model for binary classification. To detect $M_o$, positive samples are random non-watermarked generations under $M_o$ and negative ones are human responses *and* non-watermarked generations under $M_t$. To train the classifier, we use the entire train split of *eli5-category* (one positive and two negative samples for each prompt), fine-tuning RoBERTa-large with Adam (Kingma & Ba, 2014) for 1 epoch using batch size 32, weight decay 0.01 and a linear learning rate schedule with no warm-up and initial rate 5e-5. To make the detector robust to shorter and truncated texts, the train set is augmented by additionally including a truncated version of each example (to half the number of tokens).

### 3.5 Hybrid Detection

We explore a number of designs for hybrid detection. For methods that are trainable or require calibration, we use held-out samples from whichever dataset is not used for testing. That is, when evaluating on *databricks-dolly-15k* we use *eli5-category* for calibration (and vice-versa). This ensures that the new detector does not overfit to the test distribution. For all non-cascade models below, `scikit-learn` (Buitinck et al., 2013) (version 1.5.1) is used with the default hyperparameter settings.

**One-sided WM → DET Cascade (1S)**. Here, we cascade the watermark and non-watermark detectors together. If the sequence-level watermark score, denoted $s_w$ equals or exceeds threshold $\lambda_w$ we predict positive otherwise we predict positive if and only if the sequence-level detector score $s_d$ equals or exceeds threshold $\lambda_d$. Since most watermark detection schemes (at least the ones we evaluate here) are significantly more computationally efficient than non-watermark ones that involve LM inference, by positioning them first in the cascade, we save on compute as the latter is only run on residual samples. We quantify our savings later. The hypothesis here is that when $s_w$ is large, the sample is likely positive, but when it's small, it could be either a negative one or a positive one for which there was insufficient entropy to embed a strong watermark, in which case we defer to the non-watermark classifier.

**Two-sided WM → DET Cascade (2S)**. Here, if $s_w \geq \lambda_w^h$ (*h* for *high*) we predict positive and if $s_w \leq \lambda_w^l$ (*l* for *low*) we predict negative, otherwise we predict positive if and only if $s_d \geq \lambda_d$. This extends the aforementioned one-sided cascade by baking in the hypothesis that while positives with low entropy have small $s_w$, it is still not as small as those of negatives.

---

[6]https://github.com/IBM/RADAR

| WM | Det | WM Only | Det Only | 1S | 1S + | 2S | 2S + | LR | LR + |
|---|---|---|---|---|---|---|---|---|---|
| Aaronson | LLh | 90.4 | 91.7 | 93.9 | $2.2_{\pm0.4}$ | 94.3 | $2.6_{\pm0.4}$ | 96.8 | $5.1_{\pm0.6}$ |
| | LLR | 90.4 | 88.0 | 92.2 | $1.8_{\pm1.1}$ | 92.8 | $2.4_{\pm1.1}$ | 94.2 | $3.8_{\pm1.0}$ |
| | RoBERTa | 90.4 | 86.0 | 94.7 | $4.3_{\pm1.0}$ | 94.7 | $4.3_{\pm1.0}$ | 98.3 | $8.0_{\pm0.9}$ |
| | Binoculars | 90.4 | 93.1 | 94.8 | $1.7_{\pm0.4}$ | 95.0 | $2.0_{\pm0.4}$ | 96.8 | $3.7_{\pm0.6}$ |
| | Radar | 90.4 | 72.9 | 85.4 | $-5.0_{\pm1.3}$ | 90.7 | $0.3_{\pm1.1}$ | 91.5 | $1.1_{\pm1.1}$ |
| Kir. (0.5) | LLh | 58.0 | 91.5 | 89.8 | $-1.7_{\pm0.4}$ | 89.8 | $-1.7_{\pm0.4}$ | 91.4 | $-0.1_{\pm0.2}$ |
| | LLR | 58.0 | 88.1 | 87.9 | $-0.2_{\pm0.2}$ | 87.9 | $-0.2_{\pm0.2}$ | 88.2 | $0.1_{\pm0.3}$ |
| | RoBERTa | 58.0 | 85.9 | 93.8 | $7.9_{\pm0.8}$ | 93.8 | $7.9_{\pm0.8}$ | 96.2 | $10.3_{\pm1.0}$ |
| | Binoculars | 58.0 | 93.4 | 92.6 | $-0.8_{\pm0.3}$ | 92.6 | $-0.8_{\pm0.4}$ | 93.4 | $-0.1_{\pm0.2}$ |
| | Radar | 58.0 | 72.9 | 72.9 | $0.0_{\pm0.0}$ | 72.9 | $0.0_{\pm0.1}$ | 72.9 | $0.0_{\pm0.7}$ |
| Kir. (2) | LLh | 80.6 | 91.7 | 93.3 | $1.5_{\pm0.4}$ | 93.3 | $1.6_{\pm0.4}$ | 94.1 | $2.4_{\pm0.5}$ |
| | LLR | 80.6 | 87.5 | 88.6 | $1.1_{\pm0.5}$ | 88.6 | $1.1_{\pm0.5}$ | 91.3 | $3.8_{\pm0.7}$ |
| | RoBERTa | 80.6 | 85.1 | 92.1 | $7.0_{\pm0.7}$ | 92.1 | $7.0_{\pm0.7}$ | 96.7 | $11.6_{\pm1.0}$ |
| | Binoculars | 80.6 | 92.9 | 93.9 | $1.1_{\pm0.4}$ | 94.0 | $1.1_{\pm0.4}$ | 95.0 | $2.1_{\pm0.5}$ |
| | Radar | 80.6 | 72.1 | 75.0 | $-5.6_{\pm1.6}$ | 81.8 | $1.2_{\pm1.5}$ | 83.4 | $2.8_{\pm1.5}$ |
| Kir. (3) | LLh | 90.2 | 90.0 | 93.5 | $3.3_{\pm0.8}$ | 92.6 | $2.4_{\pm0.8}$ | 94.1 | $3.9_{\pm0.8}$ |
| | LLR | 90.2 | 83.1 | 90.4 | $0.2_{\pm1.2}$ | 92.8 | $2.7_{\pm1.1}$ | 94.3 | $4.1_{\pm1.1}$ |
| | RoBERTa | 90.2 | 82.2 | 89.3 | $-0.8_{\pm1.1}$ | 89.3 | $-0.8_{\pm1.1}$ | 98.2 | $8.0_{\pm0.9}$ |
| | Binoculars | 90.2 | 91.0 | 94.6 | $3.6_{\pm0.6}$ | 94.7 | $3.7_{\pm0.7}$ | 95.9 | $4.9_{\pm0.6}$ |
| | Radar | 90.2 | 72.1 | 89.9 | $-0.3_{\pm1.2}$ | 90.8 | $0.6_{\pm1.2}$ | 91.2 | $1.0_{\pm1.2}$ |
| Bahri | LLh | 89.9 | 92.2 | 94.6 | $2.4_{\pm0.4}$ | 95.0 | $2.8_{\pm0.5}$ | 95.3 | $3.1_{\pm0.6}$ |
| | LLR | 89.9 | 87.4 | 93.6 | $3.7_{\pm1.1}$ | 94.3 | $4.4_{\pm1.1}$ | 94.5 | $4.6_{\pm1.1}$ |
| | RoBERTa | 89.9 | 85.2 | 92.9 | $3.0_{\pm1.0}$ | 92.9 | $3.0_{\pm1.0}$ | 98.6 | $8.7_{\pm0.9}$ |
| | Binoculars | 89.9 | 93.2 | 95.1 | $1.9_{\pm0.4}$ | 95.1 | $1.9_{\pm0.4}$ | 96.0 | $2.8_{\pm0.5}$ |
| | Radar | 89.9 | 72.4 | 85.4 | $-4.4_{\pm1.4}$ | 90.4 | $0.5_{\pm1.2}$ | 90.3 | $0.4_{\pm1.2}$ |
| Kuditipudi | LLh | 61.0 | 91.3 | 90.0 | $-1.4_{\pm0.3}$ | 89.9 | $-1.4_{\pm0.3}$ | 91.3 | $0.0_{\pm0.3}$ |
| | LLR | 61.0 | 89.2 | 89.4 | $0.3_{\pm0.2}$ | 89.4 | $0.2_{\pm0.2}$ | 89.9 | $0.7_{\pm0.4}$ |
| | RoBERTa | 61.0 | 87.6 | 95.4 | $7.8_{\pm0.8}$ | 95.3 | $7.7_{\pm0.8}$ | 96.0 | $8.5_{\pm0.8}$ |
| | Binoculars | 61.0 | 93.9 | 94.1 | $0.1_{\pm0.2}$ | 94.0 | $0.1_{\pm0.2}$ | 94.3 | $0.4_{\pm0.3}$ |
| | Radar | 61.0 | 72.1 | 72.2 | $0.1_{\pm0.1}$ | 72.2 | $0.1_{\pm0.1}$ | 72.4 | $0.3_{\pm1.1}$ |

Table 1: Main table of accuracy results when GEMMA-7B-INSTRUCT is applied to *databricks-dolly-15k* and human examples are taken as negatives, with a target length of 100 tokens. 1S and 2S stand for the one-sided and two-sided cascades respectively. 1S + is the percent improvement conferred by 1S over the watermark and non-watermark detector, whichever is better. Firstly, we observe that it is not uncommon for the non-watermark detectors to outperform the watermark ones on their own. For example, under Aaronson, Binoculars gives 93.1% accuracy whereas the watermark detector obtains 90.4%. When combined with the two-sided cascade (LR), performance boosts to 95.0% (96.8%). Watermarking is not a silver bullet and non-watermark detection strategies can provide substantial complementary value. While one and two-sided cascades provide a performance boost, the best gain is had with a simple learnable model like logistic regression.

**Logistic Regression (LR)**. An alternative to manually designing a cascade based on intuition is to have a model learn the combination. To that end, we train a logistic regression model (`sklearn.linear_model.LogisticRegression`) on $z$-score normalized $\{s_w, s_d\}$ features.

**MLP**. To see whether a higher capacity parametric model confers gains over logistic regression, we also train a ReLU network (`sklearn.neural_network.MLPClassifier`) with 2 hidden layers of 100 units on $\{s_w, s_d\}$ features.

**Decision Tree**. We also experiment with a decision tree with a max depth of 3 and Gini criterion for splitting (`sklearn.tree.DecisionTreeClassifier`).

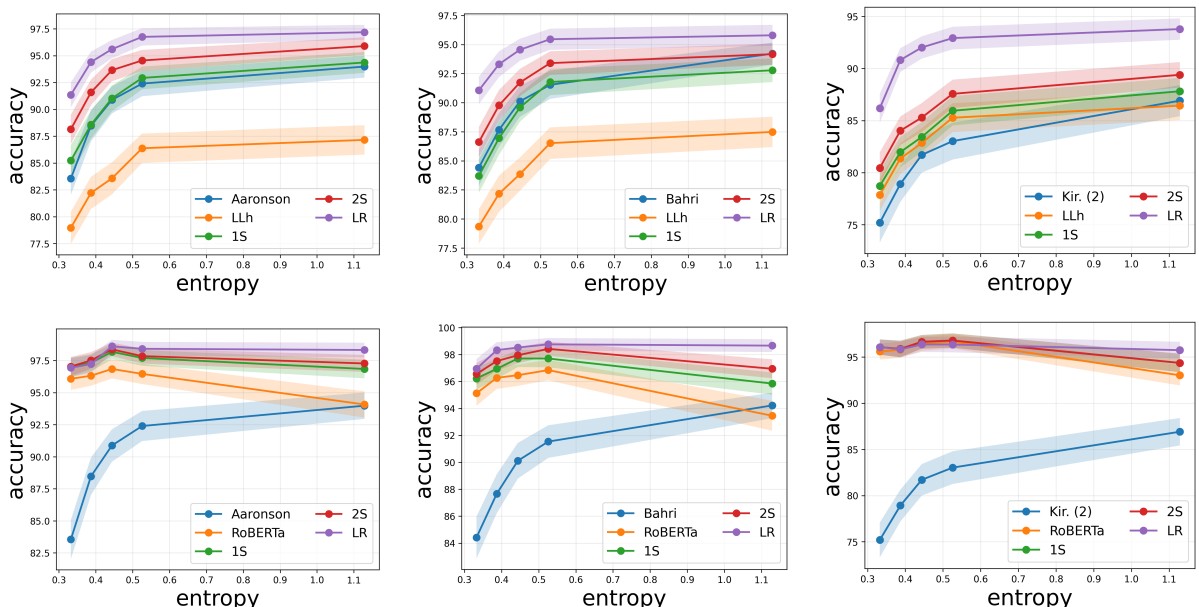

Figure 1: Detection accuracy as a function of average response entropy of prompts. The response entropy of each prompt is estimated using 4 non-watermarked generations and the prompts are partitioned based on 20% percentiles. For example, accuracy at entropy $x$ is computed on the 20% of prompts with the largest entropy that is less than or equal to $x$. GEMMA-7B-INSTRUCT is applied to *databricks-dolly-15k* under Aaronson, Kirchenbauer, and Bahri watermarking schemes. MISTRAL-7B-INSTRUCT generations are taken as negatives with a target length of 100. Likelihood (LLh) and RoBERTa detectors are shown. We see that watermarking performance improves with entropy, as we expect. While LLh also improves with entropy, the RoBERTa classifier is strong and also fairly flat, which the hybrid approaches successfully leverage to significantly improve watermarking in the low entropy regime. More results are in the Appendix.

## 3.6 Attacks

In practice, users of our model $M_o$ may act adversarially and attempt to evade detection. Following the methodology of Bahri & Wieting (2024), we study how detectability of our approaches degrades under two attack strategies — *random token replacement* and *paraphrasing*. While more advanced attacks exist, both these attacks are very easy to deploy and furthermore prior work has shown simple paraphrasing to be highly effective.

While some may argue it does not make sense to evaluate this setting because, quite simply, the text we are trying to detect is no longer text produced from our model per se, we include it nonetheless. We corrupt only the positive test samples (i.e. $M_o$'s watermarked generations), leaving the negative test ones untouched. The calibration dataset used for fitting parameters or thresholds is corrupted in the same way as the one used for testing.

**Random Token Replacement**. Here, we corrupt a random $p$-percent of watermarked tokens by replacing each with a random different token. $p$ is taken to be $\{10, 20, 30, 40\}$. This attack strategy is cheap for the adversary to carry out, but it will significantly degrade the quality of the text.

**Paraphrasing**. In this attack, the adversary attempts to evade detection by paraphrasing the watermarked text. We use Gemini-1.5-Pro (Team et al., 2024a) to paraphrase each non-truncated watermarked generation using the prompt: *"Paraphrase the following: {RESPONSE}"*. We skip examples for which Gemini does not return a response, for instance, for safety reasons.

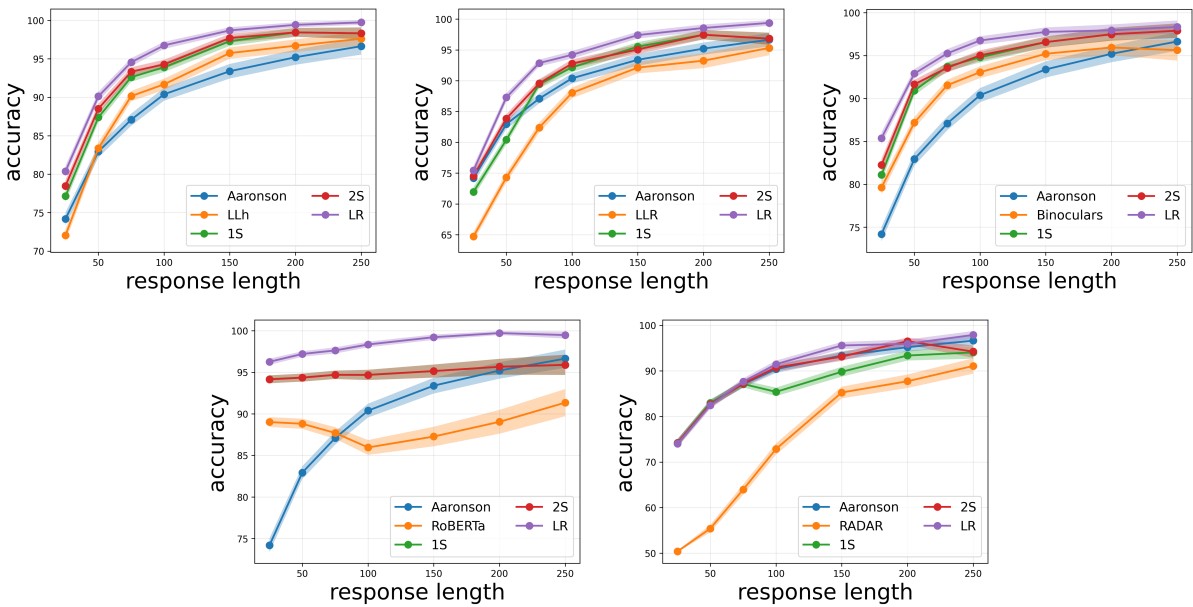

Figure 2: Detection accuracy of the cascades and logistic regression models as a function of the text length $T$ (in tokens) used for detection. The first $T$ tokens of each text is used and two standard error bars are shaded. GEMMA-7B-INSTRUCT is applied to *databricks-dolly-15k* under Aaronson and human negatives. We observe that for the watermark detector and all non-watermark detectors except the RoBERTa classifier, performance improves sharply with more test tokens, as we expect. RoBERTa's strong performance at low token count is noteworthy and likely due to the training procedure which explicitly incorporates texts of varying lengths. We find that cascades and LR combinations boost performance over either detector fairly consistently across lengths, providing assurance to the practitioner that these combinations confer benefits no matter the length of the test text.

## 3.7 Evaluation Criteria

**Detection Performance**. Our main metrics are centered solely around detection performance (as opposed to the quality of the watermarked generations) with and without adversarial corruption, as that is the focus of our work. We present two metrics.

*(1) Partial ROC-AUC.* We use partial ROC-AUC (pAUC) up to a max FPR, as utilized in Bahri & Wieting (2024). Since cascades (1S and 2S) have multiple thresholds, we sweep over a grid of thresholds (in 1% increments based on percentiles), record the (FPR, TPR) pairs that result from each threshold setting, and then determine the ROC's Pareto front, which is subsequently used to calculate pAUC using the trapezoidal rule. All other methods use a single decision threshold and for them we check every test datapoint in generating the (FPR, TPR) pairs, as is the standard practice (e.g. `sklearn`'s implementation). It is crucial to note that the granularity of the threshold grid search for cascades has a significant impact on the metric (especially so when the region of the ROC we are interested in is slim); performance should only increase the more finely grained the search is, and when every datapoint is checked, it should never underperform either detector since both cascades can represent any decision rule involving a single threshold on watermark or non-watermark scores alike. In other words, any losses for 1S and 2S under this metric should be ignored.

Our main one-number summary is pAUC with FPR $\leq 1\%$. The test datasets are always class balanced (i.e. 50/50 split), where negatives are either human responses or $M_t$'s generations.

In the Appendix, we discuss in detail how this methodology can give an unfair advantage to techniques with more tuneable decision thresholds and propose an alternative approach that attempts to remedy it.

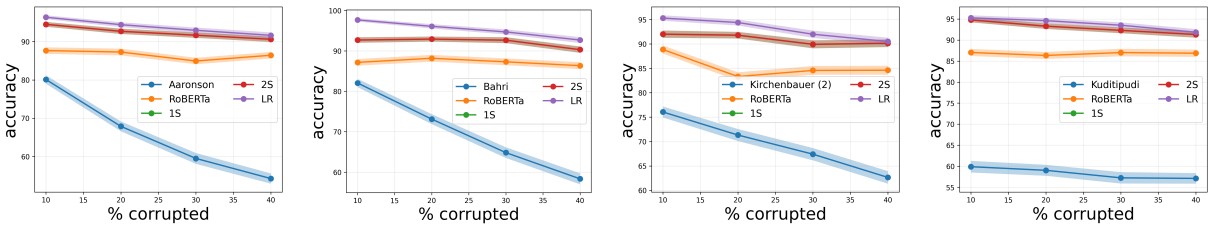

Figure 3: Detection accuracy of cascades and logistic regression as a function of the percentage of tokens corrupted. Two standard error bars are shaded. GEMMA-7B-INSTRUCT is applied to *databricks-dolly-15k* under different watermarking schemes using the RoBERTa classifier with human negatives and 100 tokens. We observe that while watermark detectors all degrade with more corruption, the RoBERTa classifier maintains consistently strong performance and that cascades and LR offer consistent improvements over either at all corruption levels. More results are in the Appendix.

We continue to use it, notwithstanding, as only at most two additional thresholds are being tuned (so the advantage conferred is small) and the alternative approach is not without its own disadvantages.

*(2) Accuracy.* We sidestep the concerns of (1) and the alternative approach discussed in the Appendix by finding the thresholds that optimize the *accuracy* on the calibration dataset and reporting test set accuracy using these thresholds. Since we are always class-balanced, accuracy here is equivalent to the arithmetic mean of TPR and TNR. Other single scalar scores such as *G-mean score* (the geometric mean of TPR and TNR) or *F-score* are also suitable, but we find no compelling reason to use them over simple accuracy, especially since the observed trends are similar.

Due to the critical role of text length, we also study performance as a function of text length by truncating the positive and negative samples to their first $T \in \{25, 50, 75, 100, 150, 200, 250\}$ tokens.

It is important to bear in mind that random text snippets (e.g. from wikipedia) are often used as negative samples in other work, whereas our negatives are model or human answers to the same prompts used to generate positive samples. The consequence is that there are far more semantic and syntactic similarities between the two classes in our work, which makes our classification task intrinsically harder. Concretely, positive and negative samples here will have more overlapping *n*-grams, the basis for many watermarking schemes, and so discrimination via the watermark will be more challenging.

**Computational Performance**. Our proposed cascading approach offers computational benefits over non-watermark detection alone, as watermark scoring is usually significantly cheaper than its counterpart. To that end, we report the fraction $\gamma$ of test samples that are caught by the watermark detector (i.e. whose prediction does not depend on the non-watermark detector). If $C_w(y)$, $C_d(y)$, $C_c(y)$ represent the computational cost (e.g. FLOPs) of detection based on watermark, non-watermark and cascade for sample $y$, respectively, then $\mathbb{E}_y C_c(y) \approx \mathbb{E}_y C_w(y) + (1 - \gamma)\mathbb{E}_y C_d(y)$.

All standard errors reported are obtained by bootstrap resampling each class of the test set separately (500 times), to ensure that the resampled test datasets remain class-balanced. $\pm 2$ standard errors are shown in all tables and figures. In the Appendix, we estimate the sensitivity of our models' performance on the calibration data, again via bootstrapping.

## 4 Experimental Results

Tables 1, 3 (Appendix), 10 (Appendix), and 11 (Appendix) show the performance of various watermarking schemes, detectors, and hybrid approaches, when GEMMA-7B-INSTRUCT is applied to *databricks-dolly-15k*, human text is taken as the negative class, and the target length is 100 tokens. Unless indicated otherwise, results are discussed under this setting. Many results are deferred to the Appendix.

**Standalone non-watermark detectors can outperform watermark ones.**   We first note that watermarking is not a silver bullet; in fact, we see that non-watermark detectors can outperform watermark ones in many cases. For example, under Aaronson, Binoculars and likelihood-based detection (LLh) achieve 93.1% and 91.7% accuracy respectively whereas watermark can only obtain 90.4%. The gains are even larger for Kuditipudi, which performs rather poorly at 61%. Thus, we should keep in mind that sometimes simply scoring the text using log-likelihood can outperform watermark-specific scoring.

**Hybrid approaches boost performance across the board.**   We see that hybrid detection lifts performance above either detection approach across the board. Observing We generally see marginal improvements going from one-sided to two-sided cascades (with the exception of RADAR whose baseline performance at 72.9% for Aaronson is quite low) but much larger ones moving to the learned logistic regression model. For example, LR with RoBERTa boosts Kirchenbauer (2) accuracy from 85.1% to 96.7% (11.6% gain). The same effective combination boosts Bahri from 89.9% to 98.6% (8.7% improvement).

**Cascade hit rates can be substantial and improve the more trustworthy watermarking is.**   In Table 3 (Appendix) we see that the hit rates grow as the strength of the watermarking increases. For example, under LLh, as Kirchenbauer's $\delta$ is cranked up $0.5 \to 2 \to 3$, we see 1S $\gamma$ increase $0\% \to 8.5\% \to 22.2\%$. Furthermore, the hit rates are large (i.e. the cascades prefer using the watermark scores for classification) when the non-watermark detectors are less reliable. For example, for Aaronson, 1S $\gamma$ for Radar is 42.9% vs. 21.6% for LLh. This confirms our expectation that cascades learn to rely more on the watermarking signal the more trustworthy it is. Furthermore, we observe that hit rates for two-sided cascades are generally higher than those for one-sided cascades, and this is anticipated as it has more degrees of freedom (an additional learnable threshold on the watermark scores).

For Aaronson, Bahri, and Kirchenbauer, $\gamma$ generally hovers around 20-40%. $\gamma$ under Kuditipudi is very low since its watermark detector is not much better than random. If the cost of watermark detection is negligible compared to its counterpart, as it often is, then cascading improves non-watermark computational efficiency by 20-40%.

**MLP and Tree generally offer gains similar to LR but risk overfitting.**   Among the learnable approaches, we find not much benefit in going from simple LR to a deep MLP or the tree — the gains, if any, are generally small, but there can be losses as well due to potential overfitting on the calibration dataset (more parameters to learn). For example, under Aaronson, MLP improves over LR by an absolute 0.6% (96.8% vs. 97.4%) but Tree loses by 2.7% (94.1%). Broadly, the tree under the hyperparameter setting we explored was more at risk of overfitting and realizing losses than MLP. These insights all suggest that the watermark and non-watermark scores are quite discriminative on their own and do not greatly benefit from excessive learning. Thus, among the learnable options, we recommend simple LR to the practitioner. In the Appendix, we analyze the decision boundaries learned by LR for the various non-watermark methods.

**Weaker non-watermark detectors can offer stronger performance in a hybrid setup.**   Interestingly, we observe that for each watermarking scheme, although the RoBERTa detector is never the best among non-watermark detectors when evaluated alone, it confers the biggest gains when leveraged in hybrid detection. For example, for Aaronson, log-likelihood (LLh) and RoBERTa achieve 91.7% and 86.0% respectively in isolation, but RoBERTa makes LR 8% better than either detector whereas LLh only provides a 5.1% boost. The difference is even starker for Binoculars (93.1% alone but only a 3.7% boost). This suggests that the RoBERTa classifier provides more complementary signal than the other approaches. One factor here is its strong performance in low entropy regimes, as discussed shortly.

**Stronger watermarking can hurt non-watermark detection but generally improves hybrid detection.**   We can observe the effect of watermarking on non-watermark detection by studying Kirchenbauer. As the strength of its watermark $\delta$ increases $0.5 \to 3$, watermark performance increases $58\% \to 90.2\%$ as expected. However, non-watermark performance can degrade a bit; for example, LLh degrades $91.5\% \to 90.0\%$ while RoBERTa drops more noticeably $85.9\% \to 82.2\%$. The reason for this drop is that stronger watermarks distort the generated text more, pushing away from the original language model distribution that the non-

watermark detectors were either scoring with or trained under. However, despite this, the hybrid LR approach performs increasingly better with a stronger watermark; e.g. LR with RoBERTa goes 96.2% → 98.2%.

**pAUC shows mostly consistent trends.** Our observed trends are largely similar when pAUC (with a max FPR of 1%) replaces accuracy as our metric. For example, under Aaronson, 2S improves LLh by 17% and Binoculars by 14%, as shown in Table 10.

**Hybrid approaches can significantly outperform watermarking alone when available entropy is low.** Figures 1 and 5 (Appendix) show the effect of entropy on performance. We see that watermarking performance indeed depends on the amount of entropy available. For example, accuracy on the 20% of prompts that afford the least amount of response entropy (whose estimation we described earlier) is about 10% worse in absolute terms than the 20% affording the most entropy. While LLh also improves with entropy, the RoBERTa classifier is both strong and fairly flat, which the hybrid approaches successfully leverage. Concretely, the cascades and logistic regression models with RoBERTa provide a near constant performance in the high 90's across the entire entropy spectrum. Lastly, hybrid approaches post improvements at all entropy levels.

**Hybrid approaches provide gains for short and long texts alike.** An important research question is how the length of the text (i.e. the number of tokens observed) impacts the performance of our methods. Figure 2 studies this, and we find that for the watermark detector and all non-watermark detectors except the RoBERTa classifier, performance improves sharply with more tokens. RoBERTa's impressive performance at low token counts is likely due to its training on texts of various lengths. We find that cascades and LR boost performance over either detector consistently across target lengths, providing assurance that the hybrid confers benefits no matter the length of the text.

**Paraphrasing attacks degrade non-watermark detectors but destroy watermark ones and gains from hybrid are minimal.** Tables 18 and 19 (Appendix) show accuracy under the paraphrasing attack when GEMMA-7B-INSTRUCT is applied to *databricks-dolly-15k*. We find that paraphrasing effectively removes most of the watermarking signal, as watermark detection is near-random. As a result, the hybrid approaches rely mostly on the non-watermark signal (which alone achieves around 70-80%) and the overall performance boost is minimal. An intriguing exception is RoBERTa, where both LR and MLPs are able to squeeze out additional signal. For example, LR under Aaronson and RoBERTa confers an 8% gain.

**Random token replacement attacks degrade watermark detectors but destroy non-watermark ones except for RoBERTa.** Figure 3 shows the accuracy of the hybrid methods for the RoBERTa classifier deteriorating with higher levels of token corruption. We observe that the classifier is surprisingly robust to this kind of corruption despite having been trained on uncorrupted samples only and that hybrid methods offer consistent improvements at all corruption levels. Figure 6 (Appendix) reports the performance for all other detectors under Aaronson. Likelihood-based detectors (LLh, LLR, and Binoculars) fail completely as even one bad token can cause the likelihood of the whole LLM-generated sequence to drop significantly, becoming even lower than that of human text. The cascades match the performance of the watermark detector here. However, LR does very well and this is an artifact of being trained on corrupted data —- specifically, it learns to flip the usual association and predict the negative class when the likelihood is *high.*

**Observations carry over to when LLM generations comprise the negative class.** Our insights largely carry over to the case where generations of a different LLM comprise the negative class. These results are presented in the Appendix. For instance, in Table 20 where GEMMA-7B-INSTRUCT is applied to *databricks-dolly-15k* using MISTRAL-7B-INSTRUCT generations as negatives, we see that 2S (LR) improves LLh by an absolute 2.9% (5.2%) and RoBERTa by 1.7% (1.9%) under Aaronson.

## 5 Conclusion

In this work, we explore schemes for first-party AI-generated content detection that combine watermark with non-watermark detection. We observe that the two complement each other well, providing a substantial boost

in performance over either. For practitioners looking to boost performance while keeping computationally costs incurred by non-watermark detection low, we recommend the two-sided cascade. For those looking for the best possible combination, we recommend the logistic regression model, which is less prone to overfitting than deep neural networks or trees but nearly equally performant. AGC detection is becoming harder but its societal implications and importance are only growing. We hope that this work will spur more interdisciplinary research on the topic and lead to effective real-world deployments.

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

## A   Appendix

### A.1   Omitted Discussions

**Kuditipudi**. We describe the distortion-free algorithm of Kuditipudi et al. (2023) in detail. Consider a secret, finite ordered list of seeds with length $k$. Begin watermarking by first selecting a position in the seed list uniformly at random and then apply the selection rule of Aaronson (2023) with the PRNG seeded to the current value. Advance to the next seed in the list (wrap-around if you are at the end) and repeat. Scoring is performed via a permutation test that evaluates how compatible the query text is with the specific list of seeds used during encoding versus any other random list of seeds of the same length. As the random starting position is not known during scoring, an alignment score based on the Levenshtein distance is formulated that considers alignments of various subsequences of the text and seeds. The proposed method is similar to Aaronson (2023) with the difference of using a fixed list of seeds and a permutation test for scoring. The upside is robustness to attacks; the downside is significantly higher computational cost during scoring. Larger $k$ offers more diversity and quality during generation but results in costlier and weaker detection.

**ROC-AUC involving multiple thresholds**. In the main text, we note that our methodology for computing ROC-AUC or pAUC favors methods with more tuneable thresholds; we now elaborate further. To see this, imagine you have a large neural network and ROC-AUC is calculated by first recording (FPR, TPR) pairs on the test dataset as every combination of the network's parameters is swept over. If AUC is subsequently calculated by integrating under the Pareto front, then it would be misleadingly high, as you would have, in effect, fitted the network to the test dataset.

Now, consider an alternative approach. For each method, the set of thresholds $\{\tau_i\}$ that achieve maximal TPR@$\beta$-FPR on a *(non-test) calibration dataset*, for a uniform grid of $\beta$, is recorded; these are the thresholds corresponding to the ROC's Pareto front when it is computed on the calibration dataset. Then (FPR, TPR) pairs are computed on the test set using the aforementioned set of thresholds $\{\tau_i\}$, and AUC / pAUC is estimated on these pairs using the trapezoidal rule. While more ideal, it has one crucial drawback: while $\{\tau_i\}$ gives (FPR, TPR) points that nicely cover the calibration ROC as the FPRs lie on a uniform grid on $[0, 1]$, this need not hold when the same thresholds are applied to the test set. In fact, we found that there can be large regions of the test ROC with little to no coverage, and this introduces unacceptably large errors in the estimation of AUC using the trapezoidal rule.

One might argue that we can just compare (FPR, TPR) pairs themselves rather than inaccurately estimated AUC, but the trouble with this is that the (FPR, TPR)'s for different methods end up aligning on neither the FPR nor TPR axis, which makes direct comparisons challenging.

## B   Omitted Figures

Figure 4 shows the decision boundary our logistic regression model learns for a specific setting. We see that the learned model consistently puts more weight on the non-watermark detector scores than the watermark ones and this is more pronounced for RoBERTa.

Figures 5 and 6 show, respectively, the effect of entropy and amount of token corruption on accuracy under the Aaronson scheme.

## C   Omitted Tables

Table 2 shows the standard deviation of the test accuracy sampling distribution for our hybrid models when the *calibration data* is bootstrap resampled (100 times) and the test samples are held fixed. Results for each individual detector, 1S, 2S, and LR methods under Aaronson watermarking are depicted when Gemma-7B-instruct is applied to *databricks-dolly-15k* and human examples are taken as negatives, with a target length of 100 tokens. We see that the standard deviations are low, suggesting low sensitivity to the calibration data and fitting procedure.

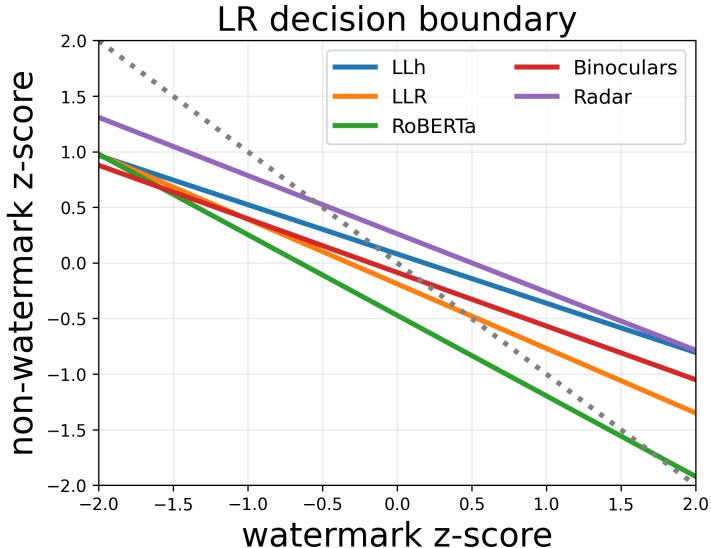

Figure 4: The decision boundary learned by our logistic regression (LR) model when GEMMA-7B-INSTRUCT is applied to *databricks-dolly-15k* under the Aaronson scheme. Human responses are taken as negatives and 100 tokens are used. Our LR models are trained on top of $z$-score normalized watermark and non-watermark scores. The area $y >= x$ represents positive predictions (i.e. the sample is from our model). $y = -x$ is shown for reference. By noticing that the slope magnitude is less than one, we see the learned LR consistently puts more weight on the non-watermark detector scores than the watermark ones and this is more pronounced for RoBERTa.

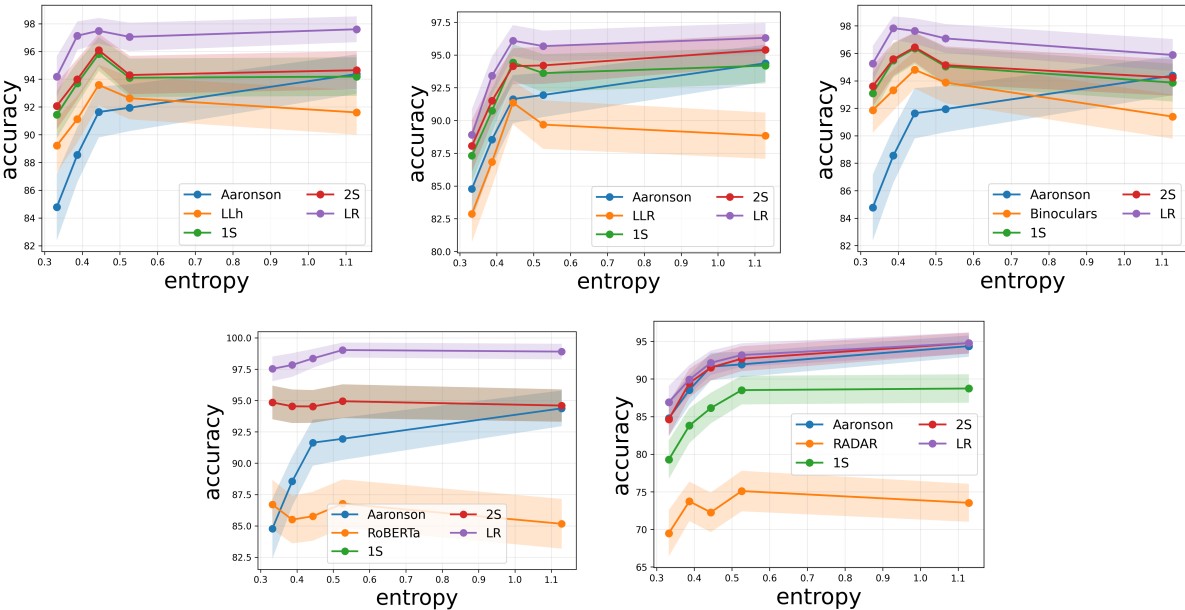

Figure 5: Detection accuracy of cascades and logistic regression as a function of average response entropy of prompts. GEMMA-7B-INSTRUCT is applied to *databricks-dolly-15k* under the Aaronson scheme. Human responses are taken as negatives and 100 tokens are used. Watermarking improves with entropy and hybrid methods boast gains across the board.

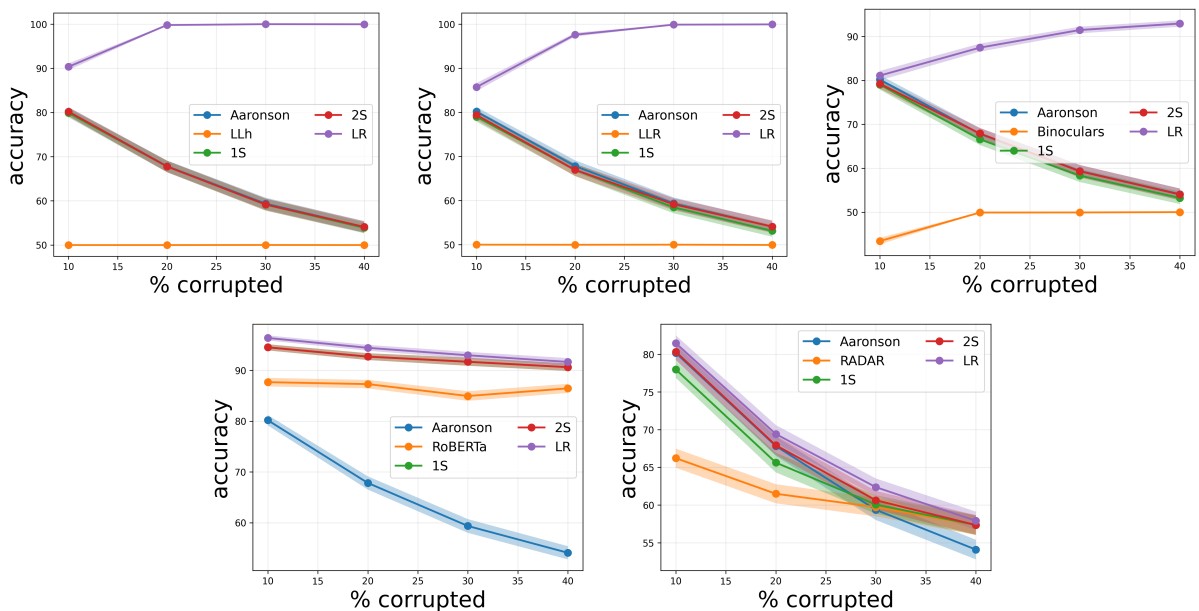

Figure 6: Detection accuracy of cascades and logistic regression as a function of the percentage of tokens corrupted. Two standard error bars are shaded. GEMMA-7B-INSTRUCT is applied to *databricks-dolly-15k* under Aaronson using human negatives and 100 tokens for various non-watermark detectors. We observe strong performance from our hybrid approaches across corruption levels. Interestingly, logistic regression's performance *improves* with more corruption, for the likelihood-based detectors. While this is counterintuitive, the explanation is that corrupting tokens of the positive samples via random flips will cause the likelihood scores of the positive samples to drop significantly — far below those of the negative (human) samples. Since the LR model is trained on corrupted data as well, it learns to achieve near perfect accuracy simply by flipping its sign: *high* likelihood now indicates "negative" and *low* likelihood indicates "positive", the reverse of the non-corrupted case.

| WM | Det | WM Only | Det Only | 1S | 2S | LR |
|---|---|---|---|---|---|---|
| Aaronson | LLh | 0.150 | 1.169 | 1.022 | 0.924 | 0.598 |
| | LLR | 0.150 | 0.627 | 1.079 | 0.839 | 0.842 |
| | RoBERTa | 0.150 | 1.233 | 0.649 | 0.647 | 0.051 |
| | Binoculars | 0.150 | 0.439 | 0.444 | 0.388 | 0.282 |
| | Radar | 0.150 | 0.000 | 2.456 | 0.343 | 0.268 |

Table 2: Standard deviation of the test accuracy sampling distribution when the calibration data is bootstrap resampled 100 times and new hybrid models and decision thresholds are fit each time. Results are for each individual detector, 1S, 2S, and LR methods under Aaronson watermarking when GEMMA-7B-INSTRUCT is applied to *databricks-dolly-15k* and human examples are taken as negatives, with a target length of 100 tokens. We find that the standard deviations are rather low, especially for logistic regression, suggesting that our findings are not too sensitive to randomness from the calibration data and fitting procedure.

Tables 3, 4, 5, 6, 7, 8, 9 report accuracy for human negatives and no corruption. Tables 10, 11, 12, 13, 14, 15, 16, 17 show pAUC for human negatives and no corruption. Tables 18 and 19 report accuracy for human negatives under the paraphrasing attack. Tables 20, 21, 22, 23 report accuracy for LLM negatives and no corruption.

| WM | Det | 1S $\gamma$ | 2S $\gamma$ | MLP | MLP + | Tree | Tree + |
|---|---|---|---|---|---|---|---|
| Aaronson | LLh | $21.6_{\pm0.9}$ | $24.5_{\pm1.1}$ | 97.4 | $5.7_{\pm0.6}$ | 94.1 | $2.4_{\pm0.4}$ |
| | LLR | $27.4_{\pm0.9}$ | $38.7_{\pm1.3}$ | 95.6 | $5.2_{\pm1.0}$ | 93.9 | $3.6_{\pm1.0}$ |
| | RoBERTa | $24.2_{\pm1.0}$ | $24.2_{\pm1.0}$ | 97.8 | $7.5_{\pm0.9}$ | 87.0 | $-3.4_{\pm1.2}$ |
| | Binoculars | $26.4_{\pm1.0}$ | $28.6_{\pm1.1}$ | 96.7 | $3.6_{\pm0.6}$ | 95.0 | $1.9_{\pm0.6}$ |
| | Radar | $42.9_{\pm0.8}$ | $74.2_{\pm1.2}$ | 91.4 | $1.0_{\pm1.1}$ | 91.2 | $0.8_{\pm1.1}$ |
| Kir. (0.5) | LLh | $0.0_{\pm0.0}$ | $0.1_{\pm0.1}$ | 92.1 | $0.6_{\pm0.3}$ | 91.9 | $0.3_{\pm0.2}$ |
| | LLR | $0.0_{\pm0.0}$ | $0.1_{\pm0.1}$ | 88.1 | $0.0_{\pm0.4}$ | 86.9 | $-1.2_{\pm0.4}$ |
| | RoBERTa | $0.0_{\pm0.0}$ | $0.1_{\pm0.1}$ | 92.4 | $6.5_{\pm0.7}$ | 86.3 | $0.4_{\pm0.2}$ |
| | Binoculars | $0.0_{\pm0.0}$ | $0.1_{\pm0.1}$ | 93.5 | $0.1_{\pm0.2}$ | 93.2 | $-0.3_{\pm0.3}$ |
| | Radar | $0.1_{\pm0.1}$ | $0.1_{\pm0.1}$ | 72.6 | $-0.3_{\pm0.3}$ | 72.9 | $0.0_{\pm0.0}$ |
| Kir. (2) | LLh | $8.5_{\pm0.7}$ | $8.6_{\pm0.7}$ | 93.9 | $2.1_{\pm0.5}$ | 94.9 | $3.1_{\pm0.5}$ |
| | LLR | $16.7_{\pm1.0}$ | $16.7_{\pm1.0}$ | 91.7 | $4.2_{\pm0.6}$ | 91.6 | $4.1_{\pm0.8}$ |
| | RoBERTa | $6.6_{\pm0.7}$ | $6.7_{\pm0.7}$ | 96.4 | $11.3_{\pm0.9}$ | 87.1 | $2.0_{\pm0.4}$ |
| | Binoculars | $12.1_{\pm0.8}$ | $12.1_{\pm0.8}$ | 95.1 | $2.2_{\pm0.5}$ | 94.6 | $1.8_{\pm0.4}$ |
| | Radar | $16.7_{\pm0.9}$ | $40.6_{\pm1.4}$ | 82.0 | $1.3_{\pm1.5}$ | 79.4 | $-1.2_{\pm1.5}$ |
| Kir. (3) | LLh | $22.2_{\pm1.0}$ | $29.7_{\pm1.2}$ | 96.1 | $5.9_{\pm0.7}$ | 93.5 | $3.4_{\pm0.8}$ |
| | LLR | $33.1_{\pm0.9}$ | $51.0_{\pm1.3}$ | 95.1 | $5.0_{\pm1.0}$ | 93.8 | $3.6_{\pm1.1}$ |
| | RoBERTa | $22.0_{\pm1.0}$ | $22.1_{\pm1.0}$ | 97.5 | $7.4_{\pm0.9}$ | 87.0 | $-3.2_{\pm1.2}$ |
| | Binoculars | $30.5_{\pm0.9}$ | $36.3_{\pm1.1}$ | 95.7 | $4.7_{\pm0.6}$ | 94.6 | $3.5_{\pm0.5}$ |
| | Radar | $51.4_{\pm0.8}$ | $77.4_{\pm1.1}$ | 90.4 | $0.3_{\pm1.2}$ | 89.7 | $-0.4_{\pm1.2}$ |
| Bahri | LLh | $28.1_{\pm0.9}$ | $32.8_{\pm1.1}$ | 95.7 | $3.5_{\pm0.5}$ | 93.1 | $0.9_{\pm0.3}$ |
| | LLR | $28.1_{\pm1.0}$ | $36.8_{\pm1.2}$ | 95.4 | $5.5_{\pm1.0}$ | 93.8 | $3.9_{\pm1.1}$ |
| | RoBERTa | $21.4_{\pm1.0}$ | $21.4_{\pm1.0}$ | 98.1 | $8.2_{\pm0.9}$ | 87.0 | $-2.8_{\pm1.2}$ |
| | Binoculars | $26.4_{\pm1.0}$ | $26.4_{\pm1.0}$ | 96.2 | $3.0_{\pm0.5}$ | 96.4 | $3.2_{\pm0.6}$ |
| | Radar | $44.5_{\pm0.8}$ | $80.1_{\pm1.1}$ | 89.9 | $-0.0_{\pm1.2}$ | 89.4 | $-0.4_{\pm1.2}$ |
| Kuditipudi | LLh | $0.6_{\pm0.2}$ | $0.7_{\pm0.2}$ | 92.3 | $1.0_{\pm0.3}$ | 94.5 | $3.2_{\pm0.5}$ |
| | LLR | $1.6_{\pm0.4}$ | $1.7_{\pm0.4}$ | 89.2 | $0.0_{\pm0.4}$ | 89.3 | $0.1_{\pm0.1}$ |
| | RoBERTa | $0.6_{\pm0.2}$ | $0.7_{\pm0.2}$ | 93.9 | $6.3_{\pm0.7}$ | 87.4 | $-0.2_{\pm0.1}$ |
| | Binoculars | $0.6_{\pm0.2}$ | $0.7_{\pm0.2}$ | 93.8 | $-0.2_{\pm0.3}$ | 93.9 | $-0.0_{\pm0.5}$ |
| | Radar | $0.6_{\pm0.2}$ | $0.7_{\pm0.2}$ | 73.2 | $1.1_{\pm0.7}$ | 72.2 | $0.0_{\pm0.1}$ |

Table 3: Cascade hit rates and accuracies for MLP and Tree when GEMMA-7B-INSTRUCT is applied to *databricks-dolly-15k* with human negatives and 100 tokens. We see that the hit rates grow as the strength of the watermarking increases. For example, under LLh, as Kirchenbauer's $\delta$ is cranked up from $0.5 \rightarrow 2 \rightarrow 3$, we see 1S $\gamma$ increase $0\% \rightarrow 8.5\% \rightarrow 22.2\%$. Furthermore, the hit rates are large (i.e. the cascades prefer using the watermark scores for classification) when the non-watermark detectors are less reliable. For example, for Aaronson, 1S $\gamma$ for Radar is 42.9% vs. 21.6% for LLh. We also observe that hit rates for the two-sided cascades are generally higher than those for one-sided cascade, and this is expected as it has more degrees of freedom (an additional learnable threshold on the watermark scores). MLPs boost performance overall (they improve RoBERTa by 7.5% under Aaronson), though this is not always the case with Tree (3.4% *drop* for this same setting), and this is due to overfitting on the calibration dataset.

| WM | Det | WM Only | Det Only | 1S | 1S + | 2S | 2S + | LR | LR + |
|---|---|---|---|---|---|---|---|---|---|
| Aaronson | LLh | 90.3 | 96.6 | 97.9 | $1.3_{\pm 0.5}$ | 98.2 | $1.6_{\pm 0.5}$ | 98.5 | $1.9_{\pm 0.6}$ |
| | LLR | 90.3 | 93.8 | 96.4 | $2.7_{\pm 0.8}$ | 96.2 | $2.5_{\pm 0.7}$ | 97.5 | $3.7_{\pm 0.7}$ |
| | RoBERTa | 90.3 | 98.7 | 98.8 | $0.0_{\pm 0.4}$ | 98.7 | $0.0_{\pm 0.4}$ | 99.1 | $0.3_{\pm 0.4}$ |
| | Binoculars | 90.3 | 98.6 | 98.5 | $-0.1_{\pm 0.5}$ | 98.4 | $-0.2_{\pm 0.4}$ | 99.1 | $0.5_{\pm 0.4}$ |
| | Radar | 90.3 | 85.8 | 90.4 | $0.1_{\pm 1.2}$ | 93.0 | $2.7_{\pm 1.1}$ | 92.5 | $2.2_{\pm 1.1}$ |
| Kir. (0.5) | LLh | 54.5 | 96.4 | 96.5 | $0.0_{\pm 0.1}$ | 96.5 | $0.0_{\pm 0.1}$ | 97.2 | $0.8_{\pm 0.4}$ |
| | LLR | 54.5 | 92.1 | 91.8 | $-0.3_{\pm 0.2}$ | 91.8 | $-0.3_{\pm 0.2}$ | 93.0 | $0.8_{\pm 0.5}$ |
| | RoBERTa | 54.5 | 98.5 | 98.5 | $0.0_{\pm 0.0}$ | 98.5 | $0.0_{\pm 0.0}$ | 98.1 | $-0.5_{\pm 0.2}$ |
| | Binoculars | 54.5 | 98.6 | 98.3 | $-0.3_{\pm 0.2}$ | 98.3 | $-0.3_{\pm 0.2}$ | 98.6 | $0.0_{\pm 0.3}$ |
| | Radar | 54.5 | 85.8 | 85.6 | $-0.2_{\pm 0.2}$ | 85.1 | $-0.6_{\pm 0.5}$ | 78.9 | $-6.8_{\pm 1.1}$ |
| Kir. (2) | LLh | 79.9 | 96.4 | 96.2 | $-0.2_{\pm 0.5}$ | 96.0 | $-0.4_{\pm 0.5}$ | 96.6 | $0.3_{\pm 0.6}$ |
| | LLR | 79.9 | 91.1 | 93.1 | $2.1_{\pm 0.8}$ | 92.6 | $1.6_{\pm 0.9}$ | 95.2 | $4.1_{\pm 0.9}$ |
| | RoBERTa | 79.9 | 97.9 | 98.2 | $0.3_{\pm 0.3}$ | 98.2 | $0.3_{\pm 0.3}$ | 98.1 | $0.3_{\pm 0.4}$ |
| | Binoculars | 79.9 | 98.5 | 98.6 | $0.2_{\pm 0.3}$ | 98.1 | $-0.4_{\pm 0.4}$ | 98.9 | $0.4_{\pm 0.4}$ |
| | Radar | 79.9 | 86.0 | 80.0 | $-6.1_{\pm 1.5}$ | 88.1 | $2.1_{\pm 1.0}$ | 86.3 | $0.3_{\pm 1.3}$ |
| Kir. (3) | LLh | 90.7 | 95.0 | 96.9 | $1.9_{\pm 0.7}$ | 97.3 | $2.3_{\pm 0.7}$ | 98.6 | $3.6_{\pm 0.7}$ |
| | LLR | 90.7 | 89.0 | 94.7 | $4.0_{\pm 1.1}$ | 95.3 | $4.6_{\pm 1.1}$ | 97.1 | $6.4_{\pm 1.0}$ |
| | RoBERTa | 90.7 | 97.1 | 98.7 | $1.7_{\pm 0.5}$ | 98.7 | $1.6_{\pm 0.5}$ | 98.8 | $1.8_{\pm 0.5}$ |
| | Binoculars | 90.7 | 97.5 | 98.0 | $0.5_{\pm 0.5}$ | 98.4 | $0.9_{\pm 0.5}$ | 98.7 | $1.2_{\pm 0.5}$ |
| | Radar | 90.7 | 85.8 | 90.7 | $-0.0_{\pm 1.2}$ | 94.2 | $3.5_{\pm 1.1}$ | 93.2 | $2.5_{\pm 1.2}$ |
| Bahri | LLh | 90.2 | 96.9 | 98.2 | $1.3_{\pm 0.5}$ | 98.5 | $1.6_{\pm 0.5}$ | 98.8 | $1.9_{\pm 0.5}$ |
| | LLR | 90.2 | 91.3 | 95.5 | $4.2_{\pm 1.0}$ | 96.8 | $5.6_{\pm 0.8}$ | 98.5 | $7.3_{\pm 0.8}$ |
| | RoBERTa | 90.2 | 97.7 | 98.4 | $0.7_{\pm 0.5}$ | 98.8 | $1.1_{\pm 0.4}$ | 99.0 | $1.3_{\pm 0.4}$ |
| | Binoculars | 90.2 | 98.7 | 98.7 | $0.0_{\pm 0.4}$ | 98.7 | $0.1_{\pm 0.4}$ | 99.0 | $0.3_{\pm 0.4}$ |
| | Radar | 90.2 | 85.4 | 90.3 | $0.1_{\pm 1.3}$ | 91.8 | $1.6_{\pm 1.2}$ | 92.1 | $2.0_{\pm 1.2}$ |
| Kuditipudi | LLh | 62.1 | 97.7 | 96.9 | $-0.8_{\pm 0.3}$ | 96.9 | $-0.8_{\pm 0.3}$ | 96.6 | $-1.1_{\pm 0.4}$ |
| | LLR | 62.1 | 92.7 | 91.8 | $-0.9_{\pm 0.5}$ | 91.8 | $-0.9_{\pm 0.5}$ | 95.4 | $2.7_{\pm 0.7}$ |
| | RoBERTa | 62.1 | 98.9 | 98.9 | $-0.1_{\pm 0.1}$ | 98.9 | $-0.1_{\pm 0.1}$ | 98.9 | $-0.0_{\pm 0.2}$ |
| | Binoculars | 62.1 | 98.6 | 98.4 | $-0.3_{\pm 0.2}$ | 98.4 | $-0.3_{\pm 0.2}$ | 99.0 | $0.3_{\pm 0.3}$ |
| | Radar | 62.1 | 85.5 | 85.2 | $-0.3_{\pm 0.2}$ | 84.1 | $-1.4_{\pm 0.6}$ | 78.7 | $-6.9_{\pm 1.3}$ |

Table 4: Main table of accuracies when GEMMA-7B-INSTRUCT is applied to the test set of *eli5-category* with human negatives and 100 tokens. The trends here are similar to those for other LLMs and test datasets, with cascades and LR boasting improvements. There are sometimes anomalous losses, such as LR degrading 6.9% under Radar and Kuditipudi, which is due to overfitting since LR could achieve neutrality by zeroing out the watermark score and rely solely on the non-watermark one.

| WM | Det | 1S $\gamma$ | 2S $\gamma$ | MLP | MLP + | Tree | Tree + |
|---|---|---|---|---|---|---|---|
| Aaronson | LLh | $24.5_{\pm 1.0}$ | $48.5_{\pm 1.5}$ | 98.5 | $1.9_{\pm 0.6}$ | 98.2 | $1.6_{\pm 0.5}$ |
| | LLR | $33.6_{\pm 1.0}$ | $51.2_{\pm 1.5}$ | 96.7 | $2.9_{\pm 0.7}$ | 96.4 | $2.6_{\pm 0.8}$ |
| | RoBERTa | $35.9_{\pm 1.0}$ | $55.7_{\pm 1.4}$ | 99.4 | $0.6_{\pm 0.3}$ | 99.5 | $0.8_{\pm 0.3}$ |
| | Binoculars | $35.9_{\pm 1.0}$ | $54.3_{\pm 1.5}$ | 99.1 | $0.5_{\pm 0.4}$ | 96.5 | $-2.1_{\pm 0.5}$ |
| | Radar | $50.1_{\pm 0.8}$ | $88.6_{\pm 0.9}$ | 93.4 | $3.1_{\pm 1.1}$ | 92.3 | $2.0_{\pm 1.1}$ |
| Kir. (0.5) | LLh | $0.0_{\pm 0.0}$ | $0.0_{\pm 0.0}$ | 97.1 | $0.7_{\pm 0.3}$ | 91.4 | $-5.0_{\pm 0.6}$ |
| | LLR | $0.0_{\pm 0.0}$ | $0.0_{\pm 0.0}$ | 93.6 | $1.4_{\pm 0.5}$ | 91.9 | $-0.2_{\pm 0.1}$ |
| | RoBERTa | $0.0_{\pm 0.0}$ | $0.0_{\pm 0.0}$ | 98.6 | $0.0_{\pm 0.1}$ | 98.3 | $-0.2_{\pm 0.1}$ |
| | Binoculars | $0.0_{\pm 0.0}$ | $0.0_{\pm 0.0}$ | 98.7 | $0.1_{\pm 0.3}$ | 98.3 | $-0.3_{\pm 0.3}$ |
| | Radar | $1.6_{\pm 0.4}$ | $5.8_{\pm 0.7}$ | 84.6 | $-1.2_{\pm 0.7}$ | 82.9 | $-2.9_{\pm 0.8}$ |
| Kir. (2) | LLh | $8.9_{\pm 0.8}$ | $13.8_{\pm 1.0}$ | 96.6 | $0.3_{\pm 0.6}$ | 94.8 | $-1.5_{\pm 0.6}$ |
| | LLR | $18.8_{\pm 1.1}$ | $29.3_{\pm 1.3}$ | 95.7 | $4.7_{\pm 0.8}$ | 91.0 | $-0.1_{\pm 1.0}$ |
| | RoBERTa | $12.8_{\pm 0.9}$ | $12.8_{\pm 0.9}$ | 98.6 | $0.7_{\pm 0.4}$ | 97.7 | $-0.2_{\pm 0.1}$ |
| | Binoculars | $10.3_{\pm 0.8}$ | $20.7_{\pm 1.2}$ | 98.6 | $0.1_{\pm 0.4}$ | 99.0 | $0.5_{\pm 0.3}$ |
| | Radar | $45.8_{\pm 1.1}$ | $59.5_{\pm 1.4}$ | 88.9 | $2.9_{\pm 1.1}$ | 89.7 | $3.6_{\pm 0.9}$ |
| Kir. (3) | LLh | $27.6_{\pm 1.1}$ | $50.1_{\pm 1.5}$ | 98.7 | $3.7_{\pm 0.7}$ | 98.1 | $3.1_{\pm 0.6}$ |
| | LLR | $35.0_{\pm 1.0}$ | $59.3_{\pm 1.5}$ | 97.6 | $6.9_{\pm 1.0}$ | 97.1 | $6.4_{\pm 1.0}$ |
| | RoBERTa | $32.1_{\pm 1.0}$ | $50.1_{\pm 1.4}$ | 99.0 | $1.9_{\pm 0.5}$ | 97.3 | $0.3_{\pm 0.6}$ |
| | Binoculars | $35.7_{\pm 1.0}$ | $60.6_{\pm 1.5}$ | 99.1 | $1.6_{\pm 0.5}$ | 98.3 | $0.7_{\pm 0.5}$ |
| | Radar | $49.1_{\pm 0.9}$ | $83.6_{\pm 1.1}$ | 94.5 | $3.8_{\pm 1.1}$ | 84.7 | $-6.0_{\pm 1.3}$ |
| Bahri | LLh | $30.3_{\pm 1.0}$ | $48.1_{\pm 1.5}$ | 99.1 | $2.2_{\pm 0.5}$ | 95.7 | $-1.2_{\pm 0.7}$ |
| | LLR | $35.1_{\pm 1.1}$ | $53.1_{\pm 1.5}$ | 97.7 | $6.5_{\pm 0.9}$ | 96.9 | $5.6_{\pm 0.8}$ |
| | RoBERTa | $35.2_{\pm 1.0}$ | $43.1_{\pm 1.3}$ | 99.0 | $1.3_{\pm 0.4}$ | 99.4 | $1.7_{\pm 0.4}$ |
| | Binoculars | $35.2_{\pm 1.0}$ | $61.8_{\pm 1.5}$ | 99.4 | $0.7_{\pm 0.4}$ | 99.3 | $0.6_{\pm 0.4}$ |
| | Radar | $47.6_{\pm 0.9}$ | $86.6_{\pm 1.0}$ | 93.1 | $2.9_{\pm 1.2}$ | 86.8 | $-3.3_{\pm 1.3}$ |
| Kuditipudi | LLh | $0.7_{\pm 0.2}$ | $0.7_{\pm 0.2}$ | 97.2 | $-0.5_{\pm 0.3}$ | 96.9 | $-0.7_{\pm 0.3}$ |
| | LLR | $2.5_{\pm 0.4}$ | $3.1_{\pm 0.5}$ | 95.3 | $2.6_{\pm 0.6}$ | 91.4 | $-1.3_{\pm 0.6}$ |
| | RoBERTa | $0.7_{\pm 0.2}$ | $0.7_{\pm 0.2}$ | 99.0 | $0.1_{\pm 0.1}$ | 99.2 | $0.3_{\pm 0.2}$ |
| | Binoculars | $0.7_{\pm 0.3}$ | $0.7_{\pm 0.3}$ | 98.7 | $0.1_{\pm 0.3}$ | 97.8 | $-0.8_{\pm 0.4}$ |
| | Radar | $3.6_{\pm 0.6}$ | $13.0_{\pm 1.0}$ | 85.0 | $-0.5_{\pm 0.8}$ | 85.4 | $-0.2_{\pm 0.7}$ |

Table 5: Cascade hit rates and accuracies of MLP and Tree methods when GEMMA-7B-INSTRUCT is applied to the test set of *eli5-category* with human negatives and 100 tokens. The trends here are similar to those for other LLMs and datasets.

| WM | Det | WM Only | Det Only | 1S | 1S + | 2S | 2S + | LR | LR + |
|---|---|---|---|---|---|---|---|---|---|
| Aaronson | LLh | 96.4 | 67.1 | 95.1 | $-1.2_{\pm0.7}$ | 95.1 | $-1.2_{\pm0.7}$ | 97.3 | $0.9_{\pm0.6}$ |
| | LLR | 96.4 | 54.4 | 93.9 | $-2.5_{\pm0.8}$ | 97.2 | $0.9_{\pm0.6}$ | 95.2 | $-1.2_{\pm0.8}$ |
| | RoBERTa | 96.4 | 62.1 | 83.6 | $-12.8_{\pm1.0}$ | 83.6 | $-12.8_{\pm1.0}$ | 79.2 | $-17.1_{\pm1.0}$ |
| | Binoculars | 96.4 | 75.0 | 97.6 | $1.2_{\pm0.6}$ | 97.6 | $1.2_{\pm0.6}$ | 97.3 | $0.9_{\pm0.7}$ |
| | Radar | 96.4 | 69.1 | 93.8 | $-2.5_{\pm0.8}$ | 96.2 | $-0.1_{\pm0.7}$ | 96.8 | $0.4_{\pm0.7}$ |
| Kir. (0.5) | LLh | 61.0 | 67.4 | 71.5 | $4.1_{\pm1.0}$ | 71.5 | $4.1_{\pm1.0}$ | 70.8 | $3.4_{\pm1.3}$ |
| | LLR | 61.0 | 54.4 | 59.9 | $-1.1_{\pm1.7}$ | 60.5 | $-0.5_{\pm1.7}$ | 61.5 | $0.4_{\pm1.8}$ |
| | RoBERTa | 61.0 | 62.2 | 66.2 | $4.0_{\pm0.6}$ | 66.2 | $4.0_{\pm0.6}$ | 76.3 | $14.1_{\pm0.8}$ |
| | Binoculars | 61.0 | 73.4 | 76.7 | $3.3_{\pm1.0}$ | 76.7 | $3.3_{\pm1.0}$ | 76.8 | $3.4_{\pm0.9}$ |
| | Radar | 61.0 | 69.7 | 69.8 | $0.1_{\pm0.3}$ | 70.6 | $0.9_{\pm0.8}$ | 67.5 | $-2.2_{\pm1.4}$ |
| Kir. (2) | LLh | 87.8 | 66.2 | 90.9 | $3.0_{\pm1.0}$ | 94.0 | $6.2_{\pm0.9}$ | 95.0 | $7.1_{\pm0.8}$ |
| | LLR | 87.8 | 52.6 | 84.8 | $-3.1_{\pm1.3}$ | 91.0 | $3.2_{\pm1.1}$ | 91.6 | $3.8_{\pm1.0}$ |
| | RoBERTa | 87.8 | 65.7 | 79.2 | $-8.7_{\pm1.2}$ | 79.2 | $-8.6_{\pm1.2}$ | 83.1 | $-4.7_{\pm1.2}$ |
| | Binoculars | 87.8 | 71.1 | 93.3 | $5.5_{\pm1.0}$ | 94.1 | $6.3_{\pm1.0}$ | 95.5 | $7.7_{\pm1.0}$ |
| | Radar | 87.8 | 69.7 | 85.8 | $-2.1_{\pm1.2}$ | 90.5 | $2.6_{\pm1.1}$ | 88.6 | $0.8_{\pm1.1}$ |
| Kir. (3) | LLh | 95.8 | 59.4 | 93.5 | $-2.3_{\pm0.8}$ | 98.3 | $2.5_{\pm0.5}$ | 98.5 | $2.7_{\pm0.5}$ |
| | LLR | 95.8 | 50.3 | 92.6 | $-3.2_{\pm0.8}$ | 97.3 | $1.5_{\pm0.7}$ | 97.5 | $1.7_{\pm0.7}$ |
| | RoBERTa | 95.8 | 58.3 | 70.1 | $-25.7_{\pm1.0}$ | 95.5 | $-0.3_{\pm0.7}$ | 89.4 | $-6.4_{\pm0.9}$ |
| | Binoculars | 95.8 | 64.5 | 96.1 | $0.3_{\pm0.7}$ | 97.7 | $1.9_{\pm0.6}$ | 98.4 | $2.6_{\pm0.6}$ |
| | Radar | 95.8 | 67.1 | 94.7 | $-1.1_{\pm0.7}$ | 97.1 | $1.3_{\pm0.7}$ | 95.7 | $-0.1_{\pm0.7}$ |
| Bahri | LLh | 96.7 | 70.9 | 97.1 | $0.4_{\pm0.6}$ | 97.9 | $1.2_{\pm0.5}$ | 97.3 | $0.6_{\pm0.5}$ |
| | LLR | 96.7 | 55.7 | 93.5 | $-3.2_{\pm0.8}$ | 97.2 | $0.4_{\pm0.6}$ | 97.4 | $0.6_{\pm0.6}$ |
| | RoBERTa | 96.7 | 62.6 | 80.4 | $-16.4_{\pm1.0}$ | 80.4 | $-16.4_{\pm1.0}$ | 91.6 | $-5.1_{\pm0.8}$ |
| | Binoculars | 96.7 | 77.6 | 96.5 | $-0.2_{\pm0.6}$ | 98.7 | $1.9_{\pm0.6}$ | 97.4 | $0.6_{\pm0.6}$ |
| | Radar | 96.7 | 68.8 | 94.5 | $-2.3_{\pm0.7}$ | 97.1 | $0.4_{\pm0.6}$ | 97.0 | $0.3_{\pm0.6}$ |
| Kuditipudi | LLh | 78.5 | 65.1 | 91.4 | $12.9_{\pm1.1}$ | 91.4 | $13.0_{\pm1.1}$ | 81.9 | $3.4_{\pm0.9}$ |
| | LLR | 78.5 | 54.3 | 79.4 | $1.0_{\pm1.5}$ | 79.6 | $1.1_{\pm1.5}$ | 80.4 | $2.0_{\pm1.3}$ |
| | RoBERTa | 78.5 | 66.6 | 77.5 | $-1.0_{\pm1.4}$ | 77.5 | $-0.9_{\pm1.3}$ | 80.4 | $2.0_{\pm1.3}$ |
| | Binoculars | 78.5 | 73.3 | 94.1 | $15.6_{\pm1.1}$ | 94.1 | $15.6_{\pm1.1}$ | 86.0 | $7.6_{\pm1.2}$ |
| | Radar | 78.5 | 68.8 | 77.7 | $-0.7_{\pm1.3}$ | 79.8 | $1.3_{\pm1.2}$ | 79.5 | $1.0_{\pm1.3}$ |

Table 6: Main table of accuracies when MISTRAL-7B-INSTRUCT is applied to *databricks-dolly-15k* with human negatives and 100 tokens. The trends here are similar to those for other LLMs and test datasets. We sometimes observe a loss for RoBERTa (e.g. 17.1% drop for LR under Aaronson). This is again due to overfitting on the calibration set. The non-watermark detector performance on the calibration data (94.4% accuracy on the test set of *eli5-category*) is better than on *databricks-dolly-15k* (only 62.1%) and the model learns to put more weight on the non-watermark detector than it should. Note that this finding is partly an artifact of the evaluation procedure. While the cascades have the same loss pattern, they do not under pAUC since no out-of-distribution calibration data is used for them in this case (see Table 14).

| WM | Det | 1S $\gamma$ | 2S $\gamma$ | MLP | MLP + | Tree | Tree + |
|---|---|---|---|---|---|---|---|
| Aaronson | LLh | $43.9_{\pm0.6}$ | $44.0_{\pm0.6}$ | 97.3 | $0.9_{\pm0.6}$ | 97.9 | $1.6_{\pm0.6}$ |
| | LLR | $43.9_{\pm0.6}$ | $95.1_{\pm0.5}$ | 96.1 | $-0.2_{\pm0.7}$ | 95.7 | $-0.7_{\pm0.7}$ |
| | RoBERTa | $43.9_{\pm0.6}$ | $44.0_{\pm0.6}$ | 80.0 | $-16.4_{\pm1.0}$ | 80.2 | $-16.2_{\pm1.0}$ |
| | Binoculars | $43.9_{\pm0.6}$ | $44.0_{\pm0.6}$ | 97.4 | $1.0_{\pm0.7}$ | 97.8 | $1.5_{\pm0.6}$ |
| | Radar | $43.9_{\pm0.6}$ | $95.1_{\pm0.5}$ | 96.7 | $0.3_{\pm0.7}$ | 93.9 | $-2.5_{\pm0.8}$ |
| Kir. (0.5) | LLh | $29.1_{\pm1.2}$ | $29.2_{\pm1.2}$ | 72.8 | $5.4_{\pm1.0}$ | 70.6 | $3.2_{\pm1.3}$ |
| | LLR | $43.3_{\pm1.3}$ | $48.8_{\pm1.3}$ | 62.3 | $1.2_{\pm1.8}$ | 61.3 | $0.3_{\pm1.7}$ |
| | RoBERTa | $0.1_{\pm0.1}$ | $0.2_{\pm0.1}$ | 74.6 | $12.4_{\pm0.8}$ | 74.0 | $11.8_{\pm0.8}$ |
| | Binoculars | $22.8_{\pm1.1}$ | $22.9_{\pm1.1}$ | 77.1 | $3.6_{\pm0.9}$ | 77.3 | $3.8_{\pm0.8}$ |
| | Radar | $3.9_{\pm0.5}$ | $21.9_{\pm1.1}$ | 72.8 | $3.1_{\pm0.9}$ | 69.7 | $0.0_{\pm0.0}$ |
| Kir. (2) | LLh | $37.0_{\pm0.9}$ | $72.0_{\pm1.2}$ | 94.7 | $6.8_{\pm0.9}$ | 90.2 | $2.3_{\pm1.1}$ |
| | LLR | $37.0_{\pm0.8}$ | $87.0_{\pm0.9}$ | 90.5 | $2.7_{\pm1.1}$ | 91.3 | $3.4_{\pm1.0}$ |
| | RoBERTa | $24.0_{\pm0.9}$ | $24.1_{\pm0.9}$ | 81.0 | $-6.8_{\pm1.2}$ | 74.7 | $-13.1_{\pm1.2}$ |
| | Binoculars | $37.0_{\pm0.9}$ | $81.2_{\pm1.0}$ | 95.5 | $7.6_{\pm1.0}$ | 94.5 | $6.7_{\pm1.0}$ |
| | Radar | $37.0_{\pm0.8}$ | $87.0_{\pm0.9}$ | 91.1 | $3.2_{\pm1.1}$ | 87.2 | $-0.7_{\pm1.1}$ |
| Kir. (3) | LLh | $44.8_{\pm0.5}$ | $94.5_{\pm0.6}$ | 98.4 | $2.6_{\pm0.5}$ | 97.9 | $2.1_{\pm0.4}$ |
| | LLR | $44.8_{\pm0.6}$ | $94.5_{\pm0.6}$ | 96.8 | $1.0_{\pm0.7}$ | 95.7 | $-0.1_{\pm0.7}$ |
| | RoBERTa | $40.0_{\pm0.7}$ | $83.6_{\pm1.0}$ | 79.8 | $-16.0_{\pm1.0}$ | 75.7 | $-20.1_{\pm1.0}$ |
| | Binoculars | $44.8_{\pm0.6}$ | $83.4_{\pm1.0}$ | 98.3 | $2.5_{\pm0.6}$ | 95.6 | $-0.2_{\pm0.7}$ |
| | Radar | $44.8_{\pm0.6}$ | $96.8_{\pm0.5}$ | 95.4 | $-0.4_{\pm0.7}$ | 95.6 | $-0.2_{\pm0.7}$ |
| Bahri | LLh | $44.3_{\pm0.6}$ | $79.0_{\pm1.1}$ | 98.9 | $2.1_{\pm0.4}$ | 96.3 | $-0.5_{\pm0.7}$ |
| | LLR | $44.3_{\pm0.6}$ | $90.6_{\pm0.8}$ | 97.4 | $0.7_{\pm0.6}$ | 93.9 | $-2.9_{\pm0.8}$ |
| | RoBERTa | $37.9_{\pm0.8}$ | $37.9_{\pm0.8}$ | 91.3 | $-5.4_{\pm0.8}$ | 80.2 | $-16.6_{\pm1.0}$ |
| | Binoculars | $44.5_{\pm0.6}$ | $87.1_{\pm0.9}$ | 99.1 | $2.4_{\pm0.5}$ | 96.8 | $0.1_{\pm0.6}$ |
| | Radar | $44.5_{\pm0.5}$ | $96.1_{\pm0.5}$ | 97.4 | $0.6_{\pm0.6}$ | 97.3 | $0.5_{\pm0.6}$ |
| Kuditipudi | LLh | $25.8_{\pm0.9}$ | $28.1_{\pm0.9}$ | 92.8 | $14.3_{\pm1.1}$ | 93.4 | $14.9_{\pm1.0}$ |
| | LLR | $25.8_{\pm1.0}$ | $28.1_{\pm1.0}$ | 83.2 | $4.7_{\pm1.4}$ | 78.8 | $0.3_{\pm1.4}$ |
| | RoBERTa | $24.5_{\pm0.9}$ | $24.6_{\pm0.9}$ | 81.1 | $2.6_{\pm1.3}$ | 74.4 | $-4.0_{\pm1.3}$ |
| | Binoculars | $25.8_{\pm0.9}$ | $25.9_{\pm0.9}$ | 95.3 | $16.8_{\pm1.0}$ | 94.9 | $16.4_{\pm1.0}$ |
| | Radar | $28.2_{\pm0.9}$ | $92.3_{\pm0.7}$ | 79.8 | $1.4_{\pm1.2}$ | 75.6 | $-2.9_{\pm1.3}$ |

Table 7: Cascade hit rates and accuracies of MLP and Tree methods when MISTRAL-7B-INSTRUCT is applied to *databricks-dolly-15k* with human negatives and 100 tokens. The trends here are similar to those for other LLMs and datasets.

| WM | Det | WM Only | Det Only | 1S | 1S + | 2S | 2S + | LR | LR + |
|---|---|---|---|---|---|---|---|---|---|
| Aaronson | LLh | 99.1 | 62.9 | 99.4 | $0.3_{\pm 0.2}$ | 99.8 | $0.7_{\pm 0.3}$ | 99.9 | $0.8_{\pm 0.3}$ |
| | LLR | 99.1 | 54.1 | 99.0 | $-0.0_{\pm 0.4}$ | 99.6 | $0.6_{\pm 0.3}$ | 99.8 | $0.7_{\pm 0.3}$ |
| | RoBERTa | 99.1 | 94.4 | 99.4 | $0.3_{\pm 0.3}$ | 99.8 | $0.7_{\pm 0.3}$ | 99.9 | $0.8_{\pm 0.3}$ |
| | Binoculars | 99.1 | 67.9 | 99.8 | $0.7_{\pm 0.3}$ | 99.9 | $0.8_{\pm 0.3}$ | 99.9 | $0.8_{\pm 0.3}$ |
| | Radar | 99.1 | 75.0 | 99.1 | $0.0_{\pm 0.4}$ | 99.2 | $0.1_{\pm 0.4}$ | 99.0 | $-0.0_{\pm 0.4}$ |
| Kir. (0.5) | LLh | 63.6 | 59.0 | 66.2 | $2.6_{\pm 1.2}$ | 66.2 | $2.6_{\pm 1.2}$ | 62.6 | $-1.0_{\pm 1.5}$ |
| | LLR | 63.6 | 53.6 | 65.3 | $1.7_{\pm 1.9}$ | 65.2 | $1.6_{\pm 1.9}$ | 64.3 | $0.7_{\pm 1.9}$ |
| | RoBERTa | 63.6 | 94.7 | 94.5 | $-0.1_{\pm 0.2}$ | 94.5 | $-0.1_{\pm 0.2}$ | 97.4 | $2.7_{\pm 0.5}$ |
| | Binoculars | 63.6 | 64.5 | 69.1 | $4.6_{\pm 1.1}$ | 69.1 | $4.6_{\pm 1.1}$ | 68.6 | $4.1_{\pm 1.0}$ |
| | Radar | 63.6 | 73.1 | 75.3 | $2.3_{\pm 0.8}$ | 75.3 | $2.2_{\pm 0.9}$ | 72.3 | $-0.7_{\pm 1.7}$ |
| Kir. (2) | LLh | 94.2 | 57.8 | 97.1 | $2.9_{\pm 0.5}$ | 98.6 | $4.4_{\pm 0.6}$ | 98.6 | $4.5_{\pm 0.6}$ |
| | LLR | 94.2 | 55.4 | 94.2 | $0.0_{\pm 0.9}$ | 96.9 | $2.7_{\pm 0.8}$ | 97.2 | $3.0_{\pm 0.7}$ |
| | RoBERTa | 94.2 | 92.8 | 97.4 | $3.2_{\pm 0.8}$ | 98.8 | $4.6_{\pm 0.7}$ | 99.7 | $5.5_{\pm 0.7}$ |
| | Binoculars | 94.2 | 62.7 | 97.8 | $3.7_{\pm 0.8}$ | 98.6 | $4.4_{\pm 0.7}$ | 98.6 | $4.5_{\pm 0.7}$ |
| | Radar | 94.2 | 73.3 | 93.9 | $-0.3_{\pm 0.9}$ | 96.7 | $2.5_{\pm 0.8}$ | 95.0 | $0.8_{\pm 0.9}$ |
| Kir. (3) | LLh | 98.9 | 55.4 | 98.9 | $0.0_{\pm 0.0}$ | 99.7 | $0.8_{\pm 0.3}$ | 99.9 | $1.0_{\pm 0.3}$ |
| | LLR | 98.9 | 53.3 | 98.9 | $0.0_{\pm 0.4}$ | 98.9 | $-0.0_{\pm 0.4}$ | 99.4 | $0.5_{\pm 0.4}$ |
| | RoBERTa | 98.9 | 86.5 | 99.3 | $0.4_{\pm 0.4}$ | 99.6 | $0.7_{\pm 0.4}$ | 99.9 | $1.0_{\pm 0.3}$ |
| | Binoculars | 98.9 | 57.8 | 99.3 | $0.4_{\pm 0.4}$ | 99.5 | $0.6_{\pm 0.4}$ | 99.6 | $0.7_{\pm 0.3}$ |
| | Radar | 98.9 | 70.9 | 98.5 | $-0.4_{\pm 0.5}$ | 99.1 | $0.2_{\pm 0.4}$ | 99.1 | $0.2_{\pm 0.4}$ |
| Bahri | LLh | 98.9 | 66.6 | 99.7 | $0.8_{\pm 0.2}$ | 99.8 | $0.9_{\pm 0.3}$ | 96.1 | $-2.8_{\pm 0.5}$ |
| | LLR | 98.9 | 53.5 | 98.8 | $-0.0_{\pm 0.4}$ | 99.5 | $0.7_{\pm 0.3}$ | 98.8 | $-0.0_{\pm 0.4}$ |
| | RoBERTa | 98.9 | 95.3 | 99.3 | $0.4_{\pm 0.4}$ | 99.9 | $1.0_{\pm 0.3}$ | 98.3 | $-0.6_{\pm 0.5}$ |
| | Binoculars | 98.9 | 68.9 | 99.3 | $0.4_{\pm 0.4}$ | 99.8 | $0.9_{\pm 0.3}$ | 96.6 | $-2.2_{\pm 0.6}$ |
| | Radar | 98.9 | 77.5 | 98.8 | $-0.0_{\pm 0.4}$ | 99.0 | $0.1_{\pm 0.4}$ | 98.6 | $-0.3_{\pm 0.4}$ |
| Kuditipudi | LLh | 94.5 | 57.8 | 98.7 | $4.2_{\pm 0.6}$ | 98.7 | $4.2_{\pm 0.6}$ | 84.5 | $-9.9_{\pm 1.1}$ |
| | LLR | 94.5 | 53.5 | 95.8 | $1.3_{\pm 0.9}$ | 95.6 | $1.2_{\pm 0.9}$ | 91.1 | $-3.3_{\pm 1.1}$ |
| | RoBERTa | 94.5 | 93.6 | 99.2 | $4.8_{\pm 0.7}$ | 99.2 | $4.8_{\pm 0.7}$ | 96.5 | $2.0_{\pm 0.8}$ |
| | Binoculars | 94.5 | 62.2 | 99.0 | $4.5_{\pm 0.7}$ | 98.9 | $4.5_{\pm 0.7}$ | 88.4 | $-6.0_{\pm 1.2}$ |
| | Radar | 94.5 | 71.5 | 90.7 | $-3.8_{\pm 1.1}$ | 93.0 | $-1.4_{\pm 1.0}$ | 92.8 | $-1.6_{\pm 1.0}$ |

Table 8: Main table of accuracies when MISTRAL-7B-INSTRUCT is applied to the test split of *eli5-category* with human negatives and 100 tokens. The trends here are similar to those for other LLMs and datasets.

| WM | Det | 1S $\gamma$ | 2S $\gamma$ | MLP | MLP + | Tree | Tree + |
|---|---|---|---|---|---|---|---|
| Aaronson | LLh | $50.4_{\pm 0.3}$ | $91.8_{\pm 0.8}$ | 99.9 | $0.8_{\pm 0.3}$ | 99.6 | $0.6_{\pm 0.3}$ |
| | LLR | $50.8_{\pm 0.3}$ | $98.6_{\pm 0.3}$ | 99.7 | $0.7_{\pm 0.3}$ | 99.6 | $0.6_{\pm 0.3}$ |
| | RoBERTa | $50.4_{\pm 0.2}$ | $92.0_{\pm 0.7}$ | 99.9 | $0.8_{\pm 0.3}$ | 99.7 | $0.6_{\pm 0.3}$ |
| | Binoculars | $49.8_{\pm 0.2}$ | $91.7_{\pm 0.8}$ | 99.9 | $0.8_{\pm 0.3}$ | 99.7 | $0.6_{\pm 0.3}$ |
| | Radar | $50.7_{\pm 0.3}$ | $99.0_{\pm 0.3}$ | 99.2 | $0.2_{\pm 0.3}$ | 99.6 | $0.5_{\pm 0.3}$ |
| Kir. (0.5) | LLh | $26.4_{\pm 1.2}$ | $26.4_{\pm 1.2}$ | 66.7 | $3.1_{\pm 1.3}$ | 65.8 | $2.2_{\pm 1.3}$ |
| | LLR | $36.3_{\pm 1.4}$ | $45.7_{\pm 1.4}$ | 65.3 | $1.7_{\pm 1.9}$ | 65.0 | $1.4_{\pm 1.9}$ |
| | RoBERTa | $1.9_{\pm 0.4}$ | $1.9_{\pm 0.4}$ | 95.9 | $1.2_{\pm 0.4}$ | 94.0 | $-0.7_{\pm 0.2}$ |
| | Binoculars | $17.5_{\pm 1.0}$ | $17.5_{\pm 1.0}$ | 69.7 | $5.2_{\pm 1.0}$ | 70.0 | $5.5_{\pm 1.2}$ |
| | Radar | $17.5_{\pm 1.0}$ | $18.6_{\pm 1.0}$ | 77.8 | $4.7_{\pm 0.9}$ | 76.8 | $3.7_{\pm 1.1}$ |
| Kir. (2) | LLh | $51.4_{\pm 0.5}$ | $77.0_{\pm 1.0}$ | 98.9 | $4.7_{\pm 0.6}$ | 98.5 | $4.4_{\pm 0.6}$ |
| | LLR | $55.2_{\pm 0.6}$ | $93.2_{\pm 0.7}$ | 96.9 | $2.7_{\pm 0.8}$ | 98.0 | $3.8_{\pm 0.7}$ |
| | RoBERTa | $51.2_{\pm 0.5}$ | $72.2_{\pm 1.0}$ | 99.4 | $5.2_{\pm 0.7}$ | 98.4 | $4.2_{\pm 0.7}$ |
| | Binoculars | $50.4_{\pm 0.5}$ | $77.1_{\pm 1.1}$ | 98.7 | $4.5_{\pm 0.7}$ | 98.3 | $4.1_{\pm 0.8}$ |
| | Radar | $55.4_{\pm 0.6}$ | $91.8_{\pm 0.7}$ | 97.1 | $2.9_{\pm 0.8}$ | 97.2 | $3.1_{\pm 0.8}$ |
| Kir. (3) | LLh | $51.0_{\pm 0.3}$ | $95.7_{\pm 0.5}$ | 99.9 | $1.0_{\pm 0.3}$ | 99.4 | $0.5_{\pm 0.2}$ |
| | LLR | $51.0_{\pm 0.3}$ | $96.7_{\pm 0.5}$ | 99.6 | $0.7_{\pm 0.4}$ | 99.4 | $0.5_{\pm 0.4}$ |
| | RoBERTa | $50.6_{\pm 0.2}$ | $92.8_{\pm 0.7}$ | 99.8 | $0.9_{\pm 0.3}$ | 99.6 | $0.7_{\pm 0.3}$ |
| | Binoculars | $50.6_{\pm 0.2}$ | $92.8_{\pm 0.7}$ | 99.7 | $0.8_{\pm 0.3}$ | 99.4 | $0.5_{\pm 0.4}$ |
| | Radar | $51.4_{\pm 0.4}$ | $98.6_{\pm 0.3}$ | 99.4 | $0.5_{\pm 0.4}$ | 99.3 | $0.4_{\pm 0.4}$ |
| Bahri | LLh | $50.2_{\pm 0.2}$ | $91.7_{\pm 0.8}$ | 99.3 | $0.4_{\pm 0.2}$ | 99.7 | $0.8_{\pm 0.2}$ |
| | LLR | $51.0_{\pm 0.3}$ | $98.4_{\pm 0.4}$ | 99.5 | $0.6_{\pm 0.4}$ | 99.6 | $0.7_{\pm 0.3}$ |
| | RoBERTa | $50.6_{\pm 0.2}$ | $97.8_{\pm 0.4}$ | 99.7 | $0.8_{\pm 0.3}$ | 99.9 | $1.0_{\pm 0.3}$ |
| | Binoculars | $50.6_{\pm 0.2}$ | $93.8_{\pm 0.6}$ | 99.5 | $0.6_{\pm 0.3}$ | 99.7 | $0.8_{\pm 0.3}$ |
| | Radar | $51.0_{\pm 0.3}$ | $99.6_{\pm 0.2}$ | 99.2 | $0.3_{\pm 0.4}$ | 99.1 | $0.2_{\pm 0.4}$ |
| Kuditipudi | LLh | $46.5_{\pm 0.6}$ | $46.6_{\pm 0.6}$ | 98.6 | $4.1_{\pm 0.6}$ | 99.0 | $4.5_{\pm 0.6}$ |
| | LLR | $48.9_{\pm 0.6}$ | $58.7_{\pm 0.9}$ | 96.6 | $2.1_{\pm 0.9}$ | 97.1 | $2.7_{\pm 0.9}$ |
| | RoBERTa | $46.0_{\pm 0.6}$ | $46.1_{\pm 0.6}$ | 99.2 | $4.7_{\pm 0.7}$ | 99.2 | $4.8_{\pm 0.7}$ |
| | Binoculars | $46.5_{\pm 0.6}$ | $46.6_{\pm 0.6}$ | 99.1 | $4.6_{\pm 0.7}$ | 98.8 | $4.3_{\pm 0.8}$ |
| | Radar | $48.9_{\pm 0.6}$ | $71.9_{\pm 1.1}$ | 94.1 | $-0.3_{\pm 1.0}$ | 94.6 | $0.1_{\pm 1.0}$ |

Table 9: Cascade hit rates and accuracies of MLP and Tree methods when MISTRAL-7B-INSTRUCT is applied to the test set of *eli5-category* with human negatives and 100 tokens. The trends here are similar to those for other LLMs and datasets.

| WM | Det | WM Only | Det Only | 1S | 1S + | 2S | 2S + | LR | LR + |
|---|---|---|---|---|---|---|---|---|---|
| | LLh | 78.9 | 72.3 | 88.4 | 9.6 | 95.9 | 17.0 | 94.4 | 15.5 |
| | LLR | 78.9 | 59.8 | 81.2 | 2.3 | 93.2 | 14.3 | 75.4 | -3.5 |
| Aaronson | RoBERTa | 78.9 | 92.4 | 96.9 | 4.5 | 97.4 | 5.0 | 96.3 | 3.9 |
| | Binoculars | 78.9 | 56.5 | 78.8 | -0.1 | 92.9 | 14.0 | 79.5 | 0.6 |
| | Radar | 78.9 | 50.5 | 78.9 | -0.0 | 82.6 | 3.7 | 80.6 | 1.7 |
| | LLh | 50.3 | 71.6 | 71.9 | 0.3 | 76.7 | 5.1 | 72.4 | 0.8 |
| | LLR | 50.3 | 59.2 | 59.2 | 0.0 | 63.0 | 3.8 | 59.2 | 0.0 |
| Kir. (0.5) | RoBERTa | 50.3 | 92.9 | 91.9 | -1.0 | 91.9 | -1.0 | 91.6 | -1.3 |
| | Binoculars | 50.3 | 56.4 | 55.8 | -0.6 | 63.6 | 7.1 | 56.9 | 0.5 |
| | Radar | 50.3 | 50.5 | 50.2 | -0.4 | 51.0 | 0.5 | 51.2 | 0.7 |
| | LLh | 60.2 | 67.8 | 70.0 | 2.2 | 85.1 | 17.3 | 81.4 | 13.6 |
| | LLR | 60.2 | 56.8 | 61.5 | 1.3 | 78.1 | 17.9 | 60.2 | -0.0 |
| Kir. (2) | RoBERTa | 60.2 | 92.8 | 93.5 | 0.7 | 93.8 | 1.0 | 93.9 | 1.1 |
| | Binoculars | 60.2 | 55.4 | 60.5 | 0.3 | 80.7 | 20.5 | 63.2 | 3.0 |
| | Radar | 60.2 | 50.5 | 60.3 | 0.1 | 63.0 | 2.8 | 61.6 | 1.5 |
| | LLh | 77.7 | 62.0 | 80.1 | 2.5 | 93.9 | 16.3 | 89.7 | 12.0 |
| | LLR | 77.7 | 54.0 | 77.2 | -0.4 | 90.7 | 13.0 | 63.5 | -14.2 |
| Kir. (3) | RoBERTa | 77.7 | 91.8 | 96.1 | 4.3 | 96.7 | 4.9 | 96.2 | 4.4 |
| | Binoculars | 77.7 | 53.6 | 74.7 | -2.9 | 91.2 | 13.5 | 74.2 | -3.5 |
| | Radar | 77.7 | 50.5 | 77.7 | 0.0 | 80.4 | 2.8 | 78.6 | 0.9 |
| | LLh | 79.6 | 71.9 | 87.0 | 7.4 | 95.5 | 15.9 | 87.2 | 7.6 |
| | LLR | 79.6 | 59.0 | 80.8 | 1.2 | 92.1 | 12.5 | 67.3 | -12.3 |
| Bahri | RoBERTa | 79.6 | 92.9 | 97.3 | 4.4 | 97.4 | 4.5 | 94.3 | 1.4 |
| | Binoculars | 79.6 | 56.2 | 79.3 | -0.3 | 93.5 | 13.9 | 71.8 | -7.7 |
| | Radar | 79.6 | 50.5 | 79.7 | 0.1 | 83.0 | 3.4 | 76.6 | -3.0 |
| | LLh | 52.1 | 71.5 | 71.8 | 0.3 | 81.8 | 10.3 | 75.6 | 4.1 |
| | LLR | 52.1 | 59.2 | 59.2 | 0.0 | 68.5 | 9.4 | 60.3 | 1.1 |
| Kuditipudi | RoBERTa | 52.1 | 93.4 | 92.7 | -0.7 | 93.0 | -0.4 | 91.0 | -2.5 |
| | Binoculars | 52.1 | 56.3 | 56.2 | -0.1 | 66.5 | 10.2 | 58.0 | 1.7 |
| | Radar | 52.1 | 50.5 | 52.1 | -0.0 | 53.1 | 1.0 | 52.3 | 0.2 |

Table 10: Main table of pAUC results (1% max FPR) when Gemma-7B-instruct is applied to *databricks-dolly-15k* and human examples are used as negatives at a target length of 100 tokens.

| WM | Det | MLP | MLP + | Tree | Tree + |
|---|---|---|---|---|---|
| | LLh | 93.0 | 14.1 | 50.4 | -28.5 |
| | LLR | 77.3 | -1.6 | 50.1 | -28.8 |
| Aaronson | RoBERTa | 67.8 | -24.5 | 50.7 | -41.7 |
| | Binoculars | 69.3 | -9.6 | 51.9 | -27.0 |
| | Radar | 81.8 | 2.9 | 66.1 | -12.8 |
| | LLh | 70.8 | -0.8 | 53.1 | -18.5 |
| | LLR | 59.2 | 0.1 | 54.5 | -4.6 |
| Kir. (0.5) | RoBERTa | 65.0 | -27.9 | 50.7 | -42.2 |
| | Binoculars | 55.5 | -0.9 | 53.7 | -2.8 |
| | Radar | 51.1 | 0.6 | 50.6 | 0.1 |
| | LLh | 79.2 | 11.4 | 52.3 | -15.5 |
| | LLR | 63.1 | 2.9 | 52.5 | -7.7 |
| Kir. (2) | RoBERTa | 64.2 | -28.6 | 50.7 | -42.1 |
| | Binoculars | 58.7 | -1.5 | 53.3 | -6.9 |
| | Radar | 62.3 | 2.1 | 52.3 | -7.9 |
| | LLh | 89.1 | 11.5 | 49.9 | -27.8 |
| | LLR | 66.0 | -11.7 | 55.6 | -22.0 |
| Kir. (3) | RoBERTa | 76.0 | -15.8 | 50.7 | -41.1 |
| | Binoculars | 66.9 | -10.8 | 50.0 | -27.6 |
| | Radar | 79.1 | 1.5 | 55.8 | -21.9 |
| | LLh | 83.8 | 4.2 | 50.6 | -29.0 |
| | LLR | 63.4 | -16.2 | 50.1 | -29.5 |
| Bahri | RoBERTa | 77.4 | -15.5 | 50.7 | -42.2 |
| | Binoculars | 64.8 | -14.7 | 57.0 | -22.6 |
| | Radar | 79.9 | 0.3 | 59.2 | -20.4 |
| | LLh | 76.7 | 5.2 | 49.7 | -21.7 |
| | LLR | 62.2 | 3.1 | 51.8 | -7.4 |
| Kuditipudi | RoBERTa | 59.3 | -34.1 | 50.8 | -42.7 |
| | Binoculars | 56.5 | 0.2 | 53.7 | -2.6 |
| | Radar | 52.3 | 0.2 | 50.8 | -1.3 |

Table 11: pAUC numbers (1% max FPR) of MLP and Tree methods when Gemma-7B-instruct is applied to *databricks-dolly-15k* and human examples are used as negatives at a target length of 100 tokens.

| WM | Det | WM Only | Det Only | 1S | 1S + | 2S | 2S + | LR | LR + |
|---|---|---|---|---|---|---|---|---|---|
| Aaronson | LLh | 79.3 | 98.8 | 99.4 | 0.6 | 99.7 | 1.0 | 99.9 | 1.1 |
| | LLR | 79.3 | 92.2 | 97.4 | 5.1 | 98.6 | 6.3 | 97.8 | 5.5 |
| | RoBERTa | 79.3 | 100.0 | 99.9 | -0.1 | 99.7 | -0.3 | 99.4 | -0.6 |
| | Binoculars | 79.3 | 98.8 | 99.2 | 0.3 | 99.3 | 0.5 | 99.3 | 0.5 |
| | Radar | 79.3 | 51.7 | 79.4 | 0.1 | 86.3 | 7.0 | 82.7 | 3.4 |
| Kir. (0.5) | LLh | 50.4 | 98.9 | 97.9 | -1.0 | 88.3 | -10.7 | 98.8 | -0.2 |
| | LLR | 50.4 | 92.1 | 90.2 | -1.9 | 91.2 | -0.9 | 91.0 | -1.1 |
| | RoBERTa | 50.4 | 100.0 | 99.6 | -0.3 | 77.7 | -22.2 | 98.3 | -1.7 |
| | Binoculars | 50.4 | 98.9 | 97.7 | -1.2 | 89.5 | -9.4 | 98.6 | -0.3 |
| | Radar | 50.4 | 51.7 | 50.4 | -1.3 | 53.0 | 1.2 | 51.8 | 0.1 |
| Kir. (2) | LLh | 59.7 | 98.4 | 98.2 | -0.1 | 98.9 | 0.5 | 98.8 | 0.4 |
| | LLR | 59.7 | 89.5 | 90.6 | 1.1 | 92.9 | 3.4 | 91.9 | 2.3 |
| | RoBERTa | 59.7 | 99.9 | 99.7 | -0.2 | 97.3 | -2.6 | 98.1 | -1.8 |
| | Binoculars | 59.7 | 98.4 | 98.4 | -0.0 | 98.5 | 0.1 | 97.7 | -0.7 |
| | Radar | 59.7 | 51.8 | 59.9 | 0.1 | 67.8 | 8.0 | 62.3 | 2.6 |
| Kir. (3) | LLh | 78.3 | 96.8 | 98.9 | 2.1 | 99.3 | 2.5 | 99.5 | 2.7 |
| | LLR | 78.3 | 84.1 | 93.5 | 9.3 | 96.4 | 12.3 | 96.3 | 12.1 |
| | RoBERTa | 78.3 | 99.9 | 99.9 | -0.0 | 99.5 | -0.4 | 98.9 | -1.0 |
| | Binoculars | 78.3 | 96.8 | 98.6 | 1.9 | 98.8 | 2.0 | 98.3 | 1.6 |
| | Radar | 78.3 | 51.7 | 78.6 | 0.3 | 84.8 | 6.5 | 80.5 | 2.2 |
| Bahri | LLh | 79.6 | 99.1 | 99.2 | 0.1 | 99.7 | 0.7 | 99.4 | 0.4 |
| | LLR | 79.6 | 91.6 | 97.5 | 5.9 | 98.8 | 7.2 | 97.5 | 5.9 |
| | RoBERTa | 79.6 | 100.0 | 99.9 | -0.1 | 99.5 | -0.5 | 99.3 | -0.6 |
| | Binoculars | 79.6 | 98.6 | 99.3 | 0.7 | 99.6 | 1.0 | 99.3 | 0.7 |
| | Radar | 79.6 | 51.6 | 79.8 | 0.2 | 85.0 | 5.4 | 84.0 | 4.4 |
| Kuditipudi | LLh | 51.2 | 98.9 | 97.4 | -1.5 | 94.2 | -4.8 | 99.1 | 0.1 |
| | LLR | 51.2 | 93.0 | 90.7 | -2.3 | 93.1 | 0.1 | 93.6 | 0.7 |
| | RoBERTa | 51.2 | 100.0 | 99.6 | -0.4 | 79.8 | -20.2 | 99.0 | -1.0 |
| | Binoculars | 51.2 | 98.9 | 97.6 | -1.2 | 93.0 | -5.8 | 98.9 | 0.0 |
| | Radar | 51.2 | 51.7 | 51.1 | -0.6 | 53.9 | 2.2 | 54.1 | 2.3 |

Table 12: Main table of pAUC results (1% max FPR) when GEMMA-7B-INSTRUCT is applied to the test set of *eli5-category* and human examples are used as negatives at a target length of 100 tokens.

| WM | Det | MLP | MLP + | Tree | Tree + |
|---|---|---|---|---|---|
| | LLh | 99.9 | 1.1 | 98.3 | -0.5 |
| | LLR | 99.1 | 6.9 | 95.8 | 3.6 |
| Aaronson | RoBERTa | 99.5 | -0.5 | 99.5 | -0.5 |
| | Binoculars | 99.5 | 0.7 | 92.3 | -6.5 |
| | Radar | 85.5 | 6.2 | 57.1 | -22.2 |
| | LLh | 98.8 | -0.1 | 97.0 | -1.9 |
| | LLR | 91.5 | -0.6 | 90.6 | -1.5 |
| Kir. (0.5) | RoBERTa | 99.5 | -0.5 | 99.9 | -0.1 |
| | Binoculars | 98.7 | -0.1 | 98.5 | -0.4 |
| | Radar | 53.4 | 1.6 | 53.0 | 1.3 |
| | LLh | 98.7 | 0.4 | 96.8 | -1.6 |
| | LLR | 93.3 | 3.8 | 88.8 | -0.7 |
| Kir. (2) | RoBERTa | 98.6 | -1.3 | 99.7 | -0.3 |
| | Binoculars | 98.0 | -0.4 | 93.7 | -4.7 |
| | Radar | 65.3 | 5.5 | 57.8 | -2.0 |
| | LLh | 99.6 | 2.8 | 97.4 | 0.6 |
| | LLR | 97.7 | 13.5 | 93.7 | 9.5 |
| Kir. (3) | RoBERTa | 98.9 | -1.0 | 99.2 | -0.7 |
| | Binoculars | 98.7 | 1.9 | 95.3 | -1.4 |
| | Radar | 82.0 | 3.7 | 74.2 | -4.1 |
| | LLh | 99.5 | 0.4 | 97.7 | -1.3 |
| | LLR | 97.3 | 5.7 | 94.0 | 2.4 |
| Bahri | RoBERTa | 99.2 | -0.8 | 99.4 | -0.6 |
| | Binoculars | 99.3 | 0.7 | 95.8 | -2.8 |
| | Radar | 83.9 | 4.3 | 70.3 | -9.3 |
| | LLh | 99.1 | 0.1 | 97.8 | -1.1 |
| | LLR | 93.0 | 0.0 | 85.3 | -7.7 |
| Kuditipudi | RoBERTa | 99.5 | -0.5 | 99.9 | -0.1 |
| | Binoculars | 98.9 | 0.0 | 98.6 | -0.3 |
| | Radar | 54.0 | 2.3 | 51.7 | 0.0 |

Table 13: pAUC (1% max FPR) of MLP and Tree methods when GEMMA-7B-INSTRUCT is applied to the test set of *eli5-category* and human examples are used as negatives at a target length of 100 tokens.

| WM | Det | WM Only | Det Only | 1S | 1S + | 2S | 2S + | LR | LR + |
|---|---|---|---|---|---|---|---|---|---|
| Aaronson | LLh | 96.2 | 50.8 | 94.1 | -2.1 | 98.7 | 2.5 | 93.3 | -2.9 |
| | LLR | 96.2 | 50.2 | 94.2 | -2.0 | 97.1 | 0.9 | 86.6 | -9.7 |
| | RoBERTa | 96.2 | 76.5 | 97.3 | 1.1 | 98.4 | 2.2 | 97.1 | 0.9 |
| | Binoculars | 96.2 | 56.2 | 96.4 | 0.2 | 98.5 | 2.3 | 94.2 | -2.0 |
| | Radar | 96.2 | 50.5 | 96.2 | -0.0 | 96.6 | 0.4 | 96.3 | 0.1 |
| Kir. (0.5) | LLh | 51.0 | 50.5 | 50.8 | -0.3 | 51.4 | 0.4 | 51.1 | 0.1 |
| | LLR | 51.0 | 50.3 | 50.7 | -0.3 | 51.1 | 0.1 | 50.7 | -0.3 |
| | RoBERTa | 51.0 | 76.1 | 73.9 | -2.2 | 76.4 | 0.3 | 53.0 | -23.1 |
| | Binoculars | 51.0 | 55.3 | 54.2 | -1.2 | 56.8 | 1.4 | 56.6 | 1.3 |
| | Radar | 51.0 | 50.5 | 51.0 | -0.1 | 51.7 | 0.7 | 51.7 | 0.7 |
| Kir. (2) | LLh | 81.6 | 50.2 | 79.7 | -1.9 | 85.1 | 3.5 | 80.1 | -1.5 |
| | LLR | 81.6 | 50.1 | 79.2 | -2.4 | 83.2 | 1.6 | 76.7 | -4.9 |
| | RoBERTa | 81.6 | 72.4 | 87.9 | 6.3 | 91.6 | 10.0 | 84.4 | 2.8 |
| | Binoculars | 81.6 | 52.7 | 80.2 | -1.3 | 88.6 | 7.0 | 88.4 | 6.8 |
| | Radar | 81.6 | 50.5 | 81.7 | 0.2 | 84.9 | 3.3 | 82.9 | 1.3 |
| Kir. (3) | LLh | 96.4 | 50.1 | 95.1 | -1.2 | 97.7 | 1.3 | 95.4 | -1.0 |
| | LLR | 96.4 | 50.1 | 95.1 | -1.2 | 97.0 | 0.6 | 94.6 | -1.7 |
| | RoBERTa | 96.4 | 68.7 | 97.0 | 0.6 | 98.4 | 2.1 | 96.2 | -0.1 |
| | Binoculars | 96.4 | 50.9 | 95.3 | -1.1 | 98.1 | 1.8 | 98.1 | 1.7 |
| | Radar | 96.4 | 50.4 | 96.4 | 0.1 | 97.0 | 0.7 | 96.9 | 0.5 |
| Bahri | LLh | 97.0 | 50.7 | 96.1 | -0.9 | 99.2 | 2.2 | 91.0 | -6.0 |
| | LLR | 97.0 | 50.1 | 96.0 | -1.0 | 97.8 | 0.8 | 94.5 | -2.5 |
| | RoBERTa | 97.0 | 70.5 | 97.2 | 0.2 | 98.6 | 1.6 | 96.8 | -0.2 |
| | Binoculars | 97.0 | 56.6 | 97.0 | 0.1 | 99.0 | 2.0 | 91.5 | -5.5 |
| | Radar | 97.0 | 50.4 | 96.9 | -0.1 | 97.3 | 0.3 | 95.5 | -1.5 |
| Kuditipudi | LLh | 77.4 | 50.5 | 76.5 | -0.9 | 81.0 | 3.6 | 64.7 | -12.7 |
| | LLR | 77.4 | 50.3 | 76.3 | -1.1 | 78.2 | 0.8 | 54.5 | -22.9 |
| | RoBERTa | 77.4 | 75.6 | 87.3 | 9.9 | 87.8 | 10.4 | 78.4 | 1.0 |
| | Binoculars | 77.4 | 55.3 | 79.5 | 2.1 | 84.0 | 6.6 | 69.0 | -8.4 |
| | Radar | 77.4 | 50.5 | 77.3 | -0.1 | 78.0 | 0.6 | 75.7 | -1.7 |

Table 14: Main table of pAUC results (1% max FPR) when MISTRAL-7B-INSTRUCT is applied to *databricks-dolly-15k* and human examples are used as negatives at a target length of 100 tokens.

| WM | Det | MLP | MLP + | Tree | Tree + |
|---|---|---|---|---|---|
| Aaronson | LLh | 89.3 | -6.9 | 54.4 | -41.8 |
| | LLR | 91.1 | -5.1 | 55.9 | -40.3 |
| | RoBERTa | 97.2 | 1.0 | 49.8 | -46.4 |
| | Binoculars | 94.0 | -2.2 | 56.7 | -39.5 |
| | Radar | 96.3 | 0.1 | 93.9 | -2.3 |
| Kir. (0.5) | LLh | 50.5 | -0.5 | 50.8 | -0.2 |
| | LLR | 50.3 | -0.7 | 50.1 | -0.9 |
| | RoBERTa | 50.0 | -26.1 | 50.2 | -25.8 |
| | Binoculars | 55.4 | 0.1 | 52.1 | -3.2 |
| | Radar | 52.2 | 1.1 | 50.9 | -0.1 |
| Kir. (2) | LLh | 63.0 | -18.6 | 80.3 | -1.3 |
| | LLR | 76.4 | -5.2 | 79.9 | -1.7 |
| | RoBERTa | 84.7 | 3.1 | 49.8 | -31.8 |
| | Binoculars | 76.2 | -5.4 | 78.0 | -3.6 |
| | Radar | 84.5 | 2.9 | 79.3 | -2.3 |
| Kir. (3) | LLh | 92.7 | -3.6 | 81.3 | -15.1 |
| | LLR | 93.5 | -2.8 | 85.3 | -11.1 |
| | RoBERTa | 94.5 | -1.9 | 50.2 | -46.1 |
| | Binoculars | 97.3 | 0.9 | 55.8 | -40.5 |
| | Radar | 97.1 | 0.7 | 96.7 | 0.3 |
| Bahri | LLh | 82.1 | -14.8 | 50.4 | -46.5 |
| | LLR | 86.5 | -10.4 | 50.1 | -46.9 |
| | RoBERTa | 54.4 | -42.6 | 50.4 | -46.6 |
| | Binoculars | 88.2 | -8.8 | 50.9 | -46.0 |
| | Radar | 96.1 | -0.9 | 96.9 | -0.0 |
| Kuditipudi | LLh | 55.0 | -22.4 | 74.4 | -3.0 |
| | LLR | 59.7 | -17.7 | 73.7 | -3.7 |
| | RoBERTa | 51.4 | -26.0 | 50.2 | -27.2 |
| | Binoculars | 58.2 | -19.2 | 55.3 | -22.1 |
| | Radar | 77.4 | 0.0 | 77.2 | -0.2 |

Table 15: pAUC (1% max FPR) of MLP and Tree methods when MISTRAL-7B-INSTRUCT is applied to *databricks-dolly-15k* and human examples are used as negatives at a target length of 100 tokens.

| WM | Det | WM Only | Det Only | 1S | 1S + | 2S | 2S + | LR | LR + |
|---|---|---|---|---|---|---|---|---|---|
| | LLh | 99.8 | 61.0 | 100.0 | 0.2 | 100.0 | 0.2 | 100.0 | 0.2 |
| | LLR | 99.8 | 52.5 | 99.7 | -0.1 | 99.9 | 0.1 | 99.9 | 0.1 |
| Aaronson | RoBERTa | 99.8 | 98.7 | 100.0 | 0.2 | 100.0 | 0.2 | 100.0 | 0.2 |
| | Binoculars | 99.8 | 64.3 | 100.0 | 0.2 | 100.0 | 0.2 | 100.0 | 0.2 |
| | Radar | 99.8 | 51.0 | 99.6 | -0.1 | 99.8 | 0.0 | 99.8 | 0.0 |
| | LLh | 51.2 | 57.5 | 56.2 | -1.3 | 57.6 | 0.1 | 53.9 | -3.6 |
| | LLR | 51.2 | 52.7 | 52.4 | -0.3 | 52.6 | -0.0 | 51.5 | -1.2 |
| Kir. (0.5) | RoBERTa | 51.2 | 98.5 | 98.0 | -0.5 | 90.5 | -8.0 | 98.1 | -0.5 |
| | Binoculars | 51.2 | 61.1 | 60.4 | -0.8 | 60.4 | -0.7 | 59.2 | -1.9 |
| | Radar | 51.2 | 50.9 | 51.2 | -0.0 | 52.8 | 1.6 | 52.1 | 0.8 |
| | LLh | 95.1 | 56.3 | 96.5 | 1.3 | 97.6 | 2.4 | 96.5 | 1.4 |
| | LLR | 95.1 | 52.6 | 94.1 | -1.0 | 96.4 | 1.2 | 95.8 | 0.7 |
| Kir. (2) | RoBERTa | 95.1 | 98.4 | 99.9 | 1.5 | 99.9 | 1.5 | 99.8 | 1.4 |
| | Binoculars | 95.1 | 59.3 | 96.9 | 1.7 | 97.8 | 2.7 | 97.2 | 2.0 |
| | Radar | 95.1 | 50.9 | 95.1 | -0.0 | 96.6 | 1.4 | 95.8 | 0.7 |
| | LLh | 99.7 | 53.9 | 99.8 | 0.1 | 99.9 | 0.3 | 100.0 | 0.3 |
| | LLR | 99.7 | 51.6 | 99.2 | -0.4 | 99.8 | 0.1 | 99.8 | 0.1 |
| Kir. (3) | RoBERTa | 99.7 | 95.1 | 100.0 | 0.3 | 100.0 | 0.3 | 100.0 | 0.3 |
| | Binoculars | 99.7 | 54.6 | 99.7 | 0.0 | 99.9 | 0.2 | 99.8 | 0.2 |
| | Radar | 99.7 | 50.8 | 99.5 | -0.1 | 99.7 | 0.0 | 99.7 | 0.1 |
| | LLh | 99.7 | 64.9 | 100.0 | 0.3 | 100.0 | 0.3 | 83.8 | -15.9 |
| | LLR | 99.7 | 52.0 | 99.0 | -0.7 | 99.8 | 0.1 | 78.6 | -21.1 |
| Bahri | RoBERTa | 99.7 | 99.1 | 100.0 | 0.3 | 100.0 | 0.3 | 100.0 | 0.3 |
| | Binoculars | 99.7 | 65.1 | 99.8 | 0.1 | 100.0 | 0.3 | 83.0 | -16.7 |
| | Radar | 99.7 | 50.9 | 99.6 | -0.2 | 99.7 | -0.0 | 98.3 | -1.4 |
| | LLh | 95.4 | 56.5 | 98.9 | 3.5 | 98.9 | 3.5 | 63.1 | -32.3 |
| | LLR | 95.4 | 52.0 | 96.2 | 0.8 | 96.5 | 1.1 | 58.8 | -36.6 |
| Kuditipudi | RoBERTa | 95.4 | 98.2 | 99.9 | 1.7 | 99.2 | 1.0 | 99.9 | 1.7 |
| | Binoculars | 95.4 | 59.7 | 99.4 | 4.0 | 99.3 | 3.9 | 70.5 | -24.9 |
| | Radar | 95.4 | 50.8 | 95.4 | -0.0 | 95.6 | 0.2 | 85.4 | -10.0 |

Table 16: Main table of pAUC results (1% max FPR) when MISTRAL-7B-INSTRUCT is applied to the test split of *eli5-category* and human examples are used as negatives at a target length of 100 tokens.

| WM | Det | MLP | MLP + | Tree | Tree + |
|---|---|---|---|---|---|
| | LLh | 100.0 | 0.2 | 94.8 | -4.9 |
| | LLR | 99.9 | 0.1 | 94.8 | -5.0 |
| Aaronson | RoBERTa | 99.9 | 0.2 | 94.9 | -4.9 |
| | Binoculars | 99.9 | 0.2 | 94.9 | -4.9 |
| | Radar | 99.8 | 0.1 | 94.8 | -5.0 |
| | LLh | 54.3 | -3.2 | 55.8 | -1.7 |
| | LLR | 52.2 | -0.5 | 50.7 | -2.0 |
| Kir. (0.5) | RoBERTa | 97.1 | -1.4 | 97.1 | -1.4 |
| | Binoculars | 60.1 | -1.1 | 60.2 | -1.0 |
| | Radar | 53.2 | 2.0 | 52.4 | 1.2 |
| | LLh | 97.2 | 2.1 | 83.2 | -12.0 |
| | LLR | 95.6 | 0.5 | 83.2 | -12.0 |
| Kir. (2) | RoBERTa | 99.6 | 1.2 | 98.8 | 0.4 |
| | Binoculars | 97.7 | 2.6 | 83.2 | -11.9 |
| | Radar | 96.5 | 1.3 | 83.2 | -11.9 |
| | LLh | 99.9 | 0.3 | 92.7 | -7.0 |
| | LLR | 99.8 | 0.1 | 92.7 | -7.0 |
| Kir. (3) | RoBERTa | 99.9 | 0.2 | 92.8 | -6.9 |
| | Binoculars | 99.9 | 0.2 | 92.7 | -6.9 |
| | Radar | 99.8 | 0.1 | 92.7 | -6.9 |
| | LLh | 90.3 | -9.4 | 91.8 | -7.9 |
| | LLR | 97.4 | -2.3 | 91.8 | -7.9 |
| Bahri | RoBERTa | 100.0 | 0.3 | 86.1 | -13.6 |
| | Binoculars | 97.1 | -2.6 | 91.8 | -7.9 |
| | Radar | 97.7 | -2.0 | 87.8 | -11.9 |
| | LLh | 87.5 | -7.9 | 93.3 | -2.1 |
| | LLR | 87.8 | -7.6 | 92.5 | -2.8 |
| Kuditipudi | RoBERTa | 99.8 | 1.6 | 98.3 | 0.1 |
| | Binoculars | 94.8 | -0.6 | 93.5 | -1.9 |
| | Radar | 90.3 | -5.1 | 92.6 | -2.8 |

Table 17: pAUC (1% max FPR) of MLP and Tree methods when MISTRAL-7B-INSTRUCT is applied to the test set of *eli5-category* and human examples are used as negatives at a target length of 100 tokens.

| WM | Det | WM Only | Det Only | 1S | 1S + | 2S | 2S + | LR | LR + |
|---|---|---|---|---|---|---|---|---|---|
| Aaronson | LLh | 50.0 | 74.4 | 74.7 | $0.3_{\pm 0.2}$ | 74.7 | $0.3_{\pm 0.2}$ | 74.4 | $-0.0_{\pm 0.0}$ |
| | LLR | 50.0 | 71.7 | 71.7 | $0.0_{\pm 0.1}$ | 71.7 | $0.0_{\pm 0.1}$ | 71.7 | $0.0_{\pm 0.0}$ |
| | RoBERTa | 50.0 | 71.1 | 71.3 | $0.3_{\pm 0.2}$ | 71.3 | $0.3_{\pm 0.2}$ | 79.1 | $8.0_{\pm 1.4}$ |
| | Binoculars | 50.0 | 77.5 | 77.6 | $0.1_{\pm 0.1}$ | 77.6 | $0.1_{\pm 0.1}$ | 77.5 | $0.0_{\pm 0.1}$ |
| | Radar | 50.0 | 65.4 | 65.4 | $0.0_{\pm 0.1}$ | 65.4 | $0.0_{\pm 0.1}$ | 65.1 | $-0.3_{\pm 0.3}$ |
| Kir. (0.5) | LLh | 51.5 | 73.9 | 75.1 | $1.2_{\pm 0.5}$ | 75.2 | $1.3_{\pm 0.5}$ | 75.1 | $1.2_{\pm 0.4}$ |
| | LLR | 51.5 | 72.3 | 72.2 | $-0.1_{\pm 0.2}$ | 72.2 | $-0.1_{\pm 0.2}$ | 74.1 | $1.8_{\pm 0.6}$ |
| | RoBERTa | 51.5 | 70.6 | 70.4 | $-0.2_{\pm 0.2}$ | 70.5 | $-0.2_{\pm 0.2}$ | 79.1 | $8.4_{\pm 1.4}$ |
| | Binoculars | 51.5 | 77.1 | 77.2 | $0.1_{\pm 0.3}$ | 77.3 | $0.2_{\pm 0.3}$ | 77.3 | $0.2_{\pm 0.3}$ |
| | Radar | 51.5 | 65.9 | 65.9 | $0.0_{\pm 0.0}$ | 65.9 | $0.0_{\pm 0.1}$ | 65.8 | $-0.1_{\pm 0.1}$ |
| Kir. (2) | LLh | 52.5 | 75.8 | 73.9 | $-1.9_{\pm 0.5}$ | 73.9 | $-1.8_{\pm 0.5}$ | 74.8 | $-1.0_{\pm 0.4}$ |
| | LLR | 52.5 | 71.7 | 71.5 | $-0.2_{\pm 0.2}$ | 71.5 | $-0.2_{\pm 0.2}$ | 73.5 | $1.8_{\pm 0.6}$ |
| | RoBERTa | 52.5 | 71.3 | 71.3 | $-0.0_{\pm 0.0}$ | 71.3 | $-0.0_{\pm 0.1}$ | 79.7 | $8.3_{\pm 1.4}$ |
| | Binoculars | 52.5 | 77.0 | 77.1 | $0.1_{\pm 0.4}$ | 77.2 | $0.2_{\pm 0.4}$ | 77.1 | $0.1_{\pm 0.3}$ |
| | Radar | 52.5 | 65.7 | 65.7 | $0.0_{\pm 0.0}$ | 65.7 | $0.0_{\pm 0.1}$ | 65.6 | $-0.1_{\pm 0.4}$ |
| Kir. (3) | LLh | 54.5 | 76.1 | 76.1 | $0.0_{\pm 0.1}$ | 76.2 | $0.1_{\pm 0.1}$ | 76.4 | $0.2_{\pm 0.4}$ |
| | LLR | 54.5 | 71.4 | 71.4 | $0.0_{\pm 0.2}$ | 71.4 | $0.1_{\pm 0.2}$ | 73.0 | $1.6_{\pm 0.6}$ |
| | RoBERTa | 54.5 | 72.1 | 72.3 | $0.2_{\pm 0.2}$ | 72.3 | $0.3_{\pm 0.2}$ | 79.9 | $7.9_{\pm 1.3}$ |
| | Binoculars | 54.5 | 76.6 | 76.3 | $-0.3_{\pm 0.4}$ | 76.3 | $-0.3_{\pm 0.4}$ | 76.6 | $0.0_{\pm 0.5}$ |
| | Radar | 54.5 | 65.3 | 65.3 | $0.0_{\pm 0.0}$ | 65.3 | $0.0_{\pm 0.3}$ | 65.9 | $0.6_{\pm 0.7}$ |
| Bahri | LLh | 50.4 | 74.5 | 74.4 | $-0.1_{\pm 0.1}$ | 74.4 | $-0.1_{\pm 0.1}$ | 75.7 | $1.2_{\pm 0.5}$ |
| | LLR | 50.4 | 71.8 | 71.7 | $-0.1_{\pm 0.1}$ | 71.7 | $-0.1_{\pm 0.1}$ | 72.1 | $0.3_{\pm 0.2}$ |
| | RoBERTa | 50.4 | 71.0 | 72.2 | $1.2_{\pm 0.4}$ | 72.2 | $1.2_{\pm 0.4}$ | 79.3 | $8.3_{\pm 1.3}$ |
| | Binoculars | 50.4 | 76.8 | 76.8 | $0.0_{\pm 0.0}$ | 76.8 | $0.0_{\pm 0.0}$ | 77.0 | $0.2_{\pm 0.2}$ |
| | Radar | 50.4 | 65.0 | 65.0 | $0.0_{\pm 0.0}$ | 65.0 | $0.0_{\pm 0.0}$ | 65.1 | $0.0_{\pm 0.0}$ |
| Kuditipudi | LLh | 52.5 | 76.6 | 76.8 | $0.2_{\pm 0.2}$ | 76.8 | $0.2_{\pm 0.2}$ | 77.1 | $0.5_{\pm 0.4}$ |
| | LLR | 52.5 | 72.0 | 72.1 | $0.2_{\pm 0.2}$ | 72.1 | $0.2_{\pm 0.2}$ | 73.6 | $1.7_{\pm 0.5}$ |
| | RoBERTa | 52.5 | 71.4 | 71.5 | $0.1_{\pm 0.1}$ | 71.5 | $0.1_{\pm 0.2}$ | 78.9 | $7.5_{\pm 1.4}$ |
| | Binoculars | 52.5 | 78.3 | 78.4 | $0.1_{\pm 0.1}$ | 78.4 | $0.1_{\pm 0.1}$ | 77.9 | $-0.5_{\pm 0.5}$ |
| | Radar | 52.5 | 65.5 | 65.5 | $0.0_{\pm 0.1}$ | 65.1 | $-0.4_{\pm 0.6}$ | 66.0 | $0.5_{\pm 0.7}$ |

Table 18: Main table of accuracies under the paraphrasing attack. GEMMA-7B-INSTRUCT is applied to *databricks-dolly-15k* and human examples are used as negatives at 100 tokens. We observe that paraphrasing, an attack type known to be, a priori, challenging to defend against, does effectively remove most watermarking signal, as watermark detection is near random. As a result, the hybrid approaches rely mostly on the non-watermark signal and the overall performance improvement is minimal. An intriguing exception is RoBERTa, where both LR and MLPs are able to juice out additional signal: LR under Aaronson and RoBERTa confers an 8% gain.

| WM | Det | 1S $\gamma$ | 2S $\gamma$ | MLP | MLP + | Tree | Tree + |
|---|---|---|---|---|---|---|---|
| Aaronson | LLh | $0.1_{\pm 0.1}$ | $0.1_{\pm 0.1}$ | 76.5 | $2.1_{\pm 0.6}$ | 77.8 | $3.4_{\pm 0.8}$ |
| | LLR | $0.1_{\pm 0.1}$ | $0.1_{\pm 0.1}$ | 72.0 | $0.3_{\pm 0.3}$ | 74.1 | $2.4_{\pm 0.7}$ |
| | RoBERTa | $0.1_{\pm 0.1}$ | $0.1_{\pm 0.1}$ | 79.0 | $8.0_{\pm 1.4}$ | 71.0 | $-0.0_{\pm 0.1}$ |
| | Binoculars | $0.1_{\pm 0.1}$ | $0.1_{\pm 0.1}$ | 77.0 | $-0.5_{\pm 0.3}$ | 77.0 | $-0.5_{\pm 0.9}$ |
| | Radar | $0.1_{\pm 0.1}$ | $0.1_{\pm 0.1}$ | 65.0 | $-0.4_{\pm 0.5}$ | 65.8 | $0.4_{\pm 0.6}$ |
| Kir. (0.5) | LLh | $0.0_{\pm 0.0}$ | $0.1_{\pm 0.1}$ | 75.0 | $1.1_{\pm 0.4}$ | 77.6 | $3.7_{\pm 0.8}$ |
| | LLR | $0.0_{\pm 0.0}$ | $0.1_{\pm 0.1}$ | 73.3 | $1.0_{\pm 0.4}$ | 72.3 | $0.0_{\pm 0.0}$ |
| | RoBERTa | $0.0_{\pm 0.0}$ | $0.1_{\pm 0.1}$ | 77.7 | $7.0_{\pm 1.3}$ | 70.4 | $-0.3_{\pm 0.2}$ |
| | Binoculars | $0.0_{\pm 0.0}$ | $0.1_{\pm 0.1}$ | 76.6 | $-0.5_{\pm 0.5}$ | 77.3 | $0.2_{\pm 0.4}$ |
| | Radar | $0.1_{\pm 0.1}$ | $0.1_{\pm 0.1}$ | 65.7 | $-0.2_{\pm 0.2}$ | 65.9 | $0.0_{\pm 0.0}$ |
| Kir. (2) | LLh | $0.0_{\pm 0.0}$ | $0.1_{\pm 0.1}$ | 75.4 | $-0.4_{\pm 0.4}$ | 75.8 | $0.0_{\pm 0.0}$ |
| | LLR | $0.0_{\pm 0.0}$ | $0.1_{\pm 0.1}$ | 72.7 | $1.0_{\pm 0.4}$ | 74.9 | $3.3_{\pm 0.9}$ |
| | RoBERTa | $0.0_{\pm 0.0}$ | $0.1_{\pm 0.1}$ | 79.3 | $8.0_{\pm 1.3}$ | 72.1 | $0.8_{\pm 0.3}$ |
| | Binoculars | $0.0_{\pm 0.0}$ | $0.1_{\pm 0.1}$ | 76.9 | $-0.1_{\pm 0.4}$ | 77.2 | $0.2_{\pm 0.4}$ |
| | Radar | $0.0_{\pm 0.1}$ | $0.1_{\pm 0.1}$ | 65.7 | $-0.0_{\pm 0.4}$ | 66.6 | $0.9_{\pm 0.7}$ |
| Kir. (3) | LLh | $0.0_{\pm 0.0}$ | $0.1_{\pm 0.1}$ | 75.9 | $-0.3_{\pm 0.5}$ | 77.0 | $0.9_{\pm 0.6}$ |
| | LLR | $0.0_{\pm 0.0}$ | $0.1_{\pm 0.1}$ | 73.0 | $1.7_{\pm 0.6}$ | 74.1 | $2.7_{\pm 0.8}$ |
| | RoBERTa | $0.0_{\pm 0.0}$ | $0.1_{\pm 0.1}$ | 79.7 | $7.6_{\pm 1.3}$ | 73.0 | $0.9_{\pm 0.4}$ |
| | Binoculars | $0.0_{\pm 0.0}$ | $0.1_{\pm 0.1}$ | 76.3 | $-0.3_{\pm 0.4}$ | 76.6 | $-0.0_{\pm 0.1}$ |
| | Radar | $0.0_{\pm 0.0}$ | $1.1_{\pm 0.3}$ | 65.4 | $0.1_{\pm 0.4}$ | 64.7 | $-0.6_{\pm 0.6}$ |
| Bahri | LLh | $0.0_{\pm 0.0}$ | $0.0_{\pm 0.0}$ | 75.7 | $1.1_{\pm 0.5}$ | 78.1 | $3.6_{\pm 0.8}$ |
| | LLR | $0.0_{\pm 0.0}$ | $0.0_{\pm 0.0}$ | 71.4 | $-0.3_{\pm 0.3}$ | 75.1 | $3.4_{\pm 0.7}$ |
| | RoBERTa | $0.0_{\pm 0.0}$ | $0.0_{\pm 0.0}$ | 78.1 | $7.1_{\pm 1.3}$ | 71.3 | $0.3_{\pm 0.3}$ |
| | Binoculars | $0.0_{\pm 0.0}$ | $0.0_{\pm 0.0}$ | 76.8 | $0.0_{\pm 0.4}$ | 76.8 | $0.0_{\pm 0.0}$ |
| | Radar | $0.0_{\pm 0.0}$ | $0.0_{\pm 0.0}$ | 65.1 | $0.1_{\pm 0.3}$ | 64.5 | $-0.5_{\pm 0.5}$ |
| Kuditipudi | LLh | $0.0_{\pm 0.0}$ | $0.1_{\pm 0.1}$ | 76.6 | $0.0_{\pm 0.3}$ | 78.2 | $1.6_{\pm 0.9}$ |
| | LLR | $0.0_{\pm 0.0}$ | $0.1_{\pm 0.1}$ | 72.4 | $0.4_{\pm 0.3}$ | 75.4 | $3.4_{\pm 1.0}$ |
| | RoBERTa | $0.0_{\pm 0.0}$ | $0.1_{\pm 0.1}$ | 78.7 | $7.4_{\pm 1.4}$ | 71.7 | $0.3_{\pm 0.2}$ |
| | Binoculars | $0.0_{\pm 0.0}$ | $0.1_{\pm 0.1}$ | 78.3 | $-0.0_{\pm 0.5}$ | 78.3 | $0.0_{\pm 0.0}$ |
| | Radar | $0.1_{\pm 0.1}$ | $2.5_{\pm 0.4}$ | 65.8 | $0.4_{\pm 0.5}$ | 64.8 | $-0.6_{\pm 0.6}$ |

Table 19: Cascade hit rates and accuracies of MLP and Tree methods when GEMMA-7B-INSTRUCT is applied to *databricks-dolly-15k* under the paraphrasing attack (human negatives, 100 tokens). Since watermark performance is essentially random, the cascades learn to ignore it and rely solely on non-watermark signal. As a result, hit rates are near zero.

| WM | Det | WM Only | Det Only | 1S | 1S + | 2S | 2S + | LR | LR + |
|---|---|---|---|---|---|---|---|---|---|
| Aaronson | LLh | 89.9 | 83.7 | 90.4 | $0.6_{\pm 0.8}$ | 92.8 | $2.9_{\pm 0.7}$ | 95.1 | $5.2_{\pm 0.6}$ |
| | LLR | 89.9 | 78.5 | 89.8 | $-0.1_{\pm 0.8}$ | 90.5 | $0.6_{\pm 0.8}$ | 94.1 | $4.2_{\pm 0.7}$ |
| | RoBERTa | 89.9 | 96.0 | 97.4 | $1.5_{\pm 0.3}$ | 97.6 | $1.7_{\pm 0.3}$ | 97.9 | $1.9_{\pm 0.4}$ |
| Kir. (0.5) | LLh | 59.5 | 84.7 | 85.1 | $0.4_{\pm 0.1}$ | 85.1 | $0.4_{\pm 0.1}$ | 85.7 | $1.0_{\pm 0.3}$ |
| | LLR | 59.5 | 78.5 | 78.7 | $0.2_{\pm 0.1}$ | 78.7 | $0.2_{\pm 0.1}$ | 78.9 | $0.4_{\pm 0.3}$ |
| | RoBERTa | 59.5 | 96.0 | 95.8 | $-0.3_{\pm 0.1}$ | 95.7 | $-0.3_{\pm 0.1}$ | 95.6 | $-0.4_{\pm 0.4}$ |
| Kir. (2) | LLh | 81.1 | 82.8 | 83.6 | $0.8_{\pm 0.2}$ | 85.3 | $2.6_{\pm 0.4}$ | 91.1 | $8.4_{\pm 0.6}$ |
| | LLR | 81.1 | 75.8 | 81.3 | $0.1_{\pm 1.0}$ | 83.5 | $2.3_{\pm 1.0}$ | 87.9 | $6.7_{\pm 1.0}$ |
| | RoBERTa | 81.1 | 95.5 | 96.0 | $0.4_{\pm 0.2}$ | 96.0 | $0.4_{\pm 0.2}$ | 96.1 | $0.6_{\pm 0.4}$ |
| Kir. (3) | LLh | 90.3 | 77.9 | 88.6 | $-1.7_{\pm 0.8}$ | 92.2 | $1.9_{\pm 0.7}$ | 94.6 | $4.3_{\pm 0.6}$ |
| | LLR | 90.3 | 72.7 | 83.0 | $-7.3_{\pm 0.8}$ | 91.1 | $0.8_{\pm 0.8}$ | 91.4 | $1.1_{\pm 0.8}$ |
| | RoBERTa | 90.3 | 94.2 | 97.1 | $3.0_{\pm 0.4}$ | 97.2 | $3.1_{\pm 0.4}$ | 97.6 | $3.4_{\pm 0.5}$ |
| Bahri | LLh | 89.6 | 83.9 | 89.0 | $-0.6_{\pm 0.8}$ | 91.1 | $1.5_{\pm 0.8}$ | 94.0 | $4.4_{\pm 0.7}$ |
| | LLR | 89.6 | 76.0 | 87.6 | $-2.0_{\pm 0.8}$ | 91.6 | $2.0_{\pm 0.8}$ | 92.3 | $2.7_{\pm 0.7}$ |
| | RoBERTa | 89.6 | 95.6 | 96.9 | $1.3_{\pm 0.3}$ | 97.5 | $1.8_{\pm 0.3}$ | 98.2 | $2.6_{\pm 0.4}$ |
| Kuditipudi | LLh | 58.8 | 83.7 | 85.0 | $1.3_{\pm 0.2}$ | 85.1 | $1.3_{\pm 0.2}$ | 86.0 | $2.2_{\pm 0.3}$ |
| | LLR | 58.8 | 78.5 | 79.3 | $0.8_{\pm 0.2}$ | 79.3 | $0.8_{\pm 0.2}$ | 80.7 | $2.2_{\pm 0.3}$ |
| | RoBERTa | 58.8 | 96.3 | 96.0 | $-0.2_{\pm 0.2}$ | 96.0 | $-0.2_{\pm 0.2}$ | 96.4 | $0.2_{\pm 0.3}$ |

Table 20: Main table of accuracies when GEMMA-7B-INSTRUCT is applied to *databricks-dolly-15k* and MISTRAL-7B-INSTRUCT generations are used as negatives at a target length of 100 tokens. The trends here are similar to those for human negatives.

| WM | Det | 1S $\gamma$ | 2S $\gamma$ | MLP | MLP + | Tree | Tree + |
|---|---|---|---|---|---|---|---|
| Aaronson | LLh | $29.4_{\pm 0.7}$ | $44.2_{\pm 0.9}$ | 94.3 | $4.5_{\pm 0.7}$ | 91.2 | $1.3_{\pm 0.8}$ |
| | LLR | $33.1_{\pm 0.7}$ | $49.2_{\pm 1.0}$ | 94.6 | $4.8_{\pm 0.7}$ | 91.2 | $1.4_{\pm 0.8}$ |
| | RoBERTa | $25.5_{\pm 0.7}$ | $35.7_{\pm 0.9}$ | 98.2 | $2.2_{\pm 0.3}$ | 97.1 | $1.1_{\pm 0.4}$ |
| Kir. (0.5) | LLh | $0.0_{\pm 0.0}$ | $0.1_{\pm 0.1}$ | 86.0 | $1.3_{\pm 0.3}$ | 83.9 | $-0.8_{\pm 0.3}$ |
| | LLR | $0.0_{\pm 0.0}$ | $0.1_{\pm 0.1}$ | 78.7 | $0.2_{\pm 0.4}$ | 77.0 | $-1.5_{\pm 0.3}$ |
| | RoBERTa | $0.0_{\pm 0.0}$ | $0.1_{\pm 0.1}$ | 96.1 | $0.1_{\pm 0.1}$ | 94.9 | $-1.1_{\pm 0.4}$ |
| Kir. (2) | LLh | $11.7_{\pm 0.6}$ | $18.6_{\pm 0.7}$ | 89.9 | $7.1_{\pm 0.5}$ | 90.0 | $7.3_{\pm 0.6}$ |
| | LLR | $18.7_{\pm 0.7}$ | $26.7_{\pm 0.9}$ | 87.6 | $6.5_{\pm 1.0}$ | 84.7 | $3.6_{\pm 1.0}$ |
| | RoBERTa | $11.7_{\pm 0.6}$ | $11.8_{\pm 0.6}$ | 96.6 | $1.1_{\pm 0.3}$ | 96.8 | $1.2_{\pm 0.3}$ |
| Kir. (3) | LLh | $36.2_{\pm 0.7}$ | $59.7_{\pm 1.0}$ | 94.7 | $4.4_{\pm 0.6}$ | 91.4 | $1.1_{\pm 0.5}$ |
| | LLR | $34.7_{\pm 0.7}$ | $63.5_{\pm 0.9}$ | 92.7 | $2.4_{\pm 0.7}$ | 91.6 | $1.3_{\pm 0.8}$ |
| | RoBERTa | $32.4_{\pm 0.7}$ | $56.0_{\pm 1.0}$ | 97.9 | $3.7_{\pm 0.4}$ | 97.3 | $3.1_{\pm 0.4}$ |
| Bahri | LLh | $29.6_{\pm 0.7}$ | $40.9_{\pm 0.9}$ | 94.9 | $5.3_{\pm 0.7}$ | 92.1 | $2.5_{\pm 0.7}$ |
| | LLR | $31.5_{\pm 0.7}$ | $55.3_{\pm 0.9}$ | 93.3 | $3.7_{\pm 0.7}$ | 90.6 | $1.0_{\pm 0.8}$ |
| | RoBERTa | $23.1_{\pm 0.7}$ | $39.0_{\pm 0.9}$ | 98.4 | $2.7_{\pm 0.3}$ | 97.5 | $1.8_{\pm 0.3}$ |
| Kuditipudi | LLh | $0.5_{\pm 0.1}$ | $0.6_{\pm 0.1}$ | 86.2 | $2.5_{\pm 0.4}$ | 84.3 | $0.5_{\pm 0.2}$ |
| | LLR | $1.5_{\pm 0.2}$ | $1.5_{\pm 0.2}$ | 80.1 | $1.6_{\pm 0.3}$ | 82.9 | $4.5_{\pm 0.5}$ |
| | RoBERTa | $0.5_{\pm 0.1}$ | $0.6_{\pm 0.1}$ | 96.1 | $-0.1_{\pm 0.2}$ | 96.2 | $-0.0_{\pm 0.0}$ |

Table 21: Cascade hit rates and accuracies of MLP and Tree methods when GEMMA-7B-INSTRUCT is applied to *databricks-dolly-15k* and MISTRAL-7B-INSTRUCT generations are used as negatives at a target length of 100 tokens. The trends here are similar to those for human negatives.

| WM | Det | WM Only | Det Only | 1S | 1S + | 2S | 2S + | LR | LR + |
|---|---|---|---|---|---|---|---|---|---|
| | LLh | 90.9 | 90.8 | 95.7 | $4.8_{\pm0.6}$ | 96.1 | $5.2_{\pm0.5}$ | 97.8 | $6.9_{\pm0.5}$ |
| Aaronson | LLR | 90.9 | 84.5 | 93.7 | $2.8_{\pm0.8}$ | 94.4 | $3.5_{\pm0.8}$ | 96.5 | $5.6_{\pm0.7}$ |
| | RoBERTa | 90.9 | 97.9 | 98.8 | $0.8_{\pm0.2}$ | 98.8 | $0.8_{\pm0.3}$ | 98.9 | $0.9_{\pm0.3}$ |
| | LLh | 56.9 | 90.4 | 89.9 | $-0.5_{\pm0.1}$ | 89.9 | $-0.5_{\pm0.1}$ | 90.7 | $0.3_{\pm0.3}$ |
| Kir. (0.5) | LLR | 56.9 | 83.9 | 83.9 | $-0.0_{\pm0.0}$ | 83.9 | $-0.0_{\pm0.0}$ | 84.7 | $0.7_{\pm0.4}$ |
| | RoBERTa | 56.9 | 97.8 | 97.8 | $0.0_{\pm0.0}$ | 97.8 | $0.0_{\pm0.0}$ | 97.0 | $-0.8_{\pm0.2}$ |
| | LLh | 81.6 | 88.7 | 91.0 | $2.3_{\pm0.6}$ | 92.4 | $3.7_{\pm0.6}$ | 94.8 | $6.2_{\pm0.6}$ |
| Kir. (2) | LLR | 81.6 | 86.8 | 88.9 | $2.1_{\pm0.8}$ | 88.5 | $1.7_{\pm0.8}$ | 91.6 | $4.8_{\pm0.7}$ |
| | RoBERTa | 81.6 | 97.4 | 97.8 | $0.3_{\pm0.2}$ | 97.8 | $0.3_{\pm0.2}$ | 97.4 | $0.0_{\pm0.3}$ |
| | LLh | 92.0 | 88.6 | 94.5 | $2.5_{\pm0.6}$ | 95.8 | $3.8_{\pm0.6}$ | 97.7 | $5.6_{\pm0.6}$ |
| Kir. (3) | LLR | 92.0 | 79.9 | 93.3 | $1.2_{\pm0.7}$ | 94.6 | $2.6_{\pm0.7}$ | 96.4 | $4.3_{\pm0.7}$ |
| | RoBERTa | 92.0 | 96.5 | 98.3 | $1.8_{\pm0.4}$ | 98.5 | $2.0_{\pm0.4}$ | 98.5 | $2.0_{\pm0.4}$ |
| | LLh | 90.2 | 90.8 | 95.4 | $4.6_{\pm0.6}$ | 96.8 | $6.0_{\pm0.5}$ | 96.6 | $5.8_{\pm0.5}$ |
| Bahri | LLR | 90.2 | 84.2 | 92.9 | $2.6_{\pm0.8}$ | 94.6 | $4.4_{\pm0.8}$ | 97.2 | $6.9_{\pm0.7}$ |
| | RoBERTa | 90.2 | 97.4 | 98.0 | $0.6_{\pm0.3}$ | 98.4 | $1.0_{\pm0.3}$ | 98.7 | $1.3_{\pm0.3}$ |
| | LLh | 59.2 | 91.2 | 91.5 | $0.3_{\pm0.1}$ | 91.6 | $0.4_{\pm0.1}$ | 88.2 | $-3.0_{\pm0.4}$ |
| Kuditipudi | LLR | 59.2 | 85.9 | 86.7 | $0.7_{\pm0.3}$ | 86.7 | $0.7_{\pm0.3}$ | 86.7 | $0.8_{\pm0.5}$ |
| | RoBERTa | 59.2 | 98.2 | 98.2 | $0.0_{\pm0.0}$ | 98.2 | $0.0_{\pm0.0}$ | 97.9 | $-0.3_{\pm0.2}$ |

Table 22: Main table of accuracies when GEMMA-7B-INSTRUCT is applied to the test set of *eli5-category* and MISTRAL-7B-INSTRUCT generations are used as negatives at a target length of 100 tokens. The trends here are similar to those for human negatives.

| WM | Det | 1S $\gamma$ | 2S $\gamma$ | MLP | MLP + | Tree | Tree + |
|---|---|---|---|---|---|---|---|
| | LLh | $35.6_{\pm0.7}$ | $58.8_{\pm0.9}$ | 98.1 | $7.2_{\pm0.5}$ | 96.6 | $5.7_{\pm0.5}$ |
| Aaronson | LLR | $40.5_{\pm0.6}$ | $58.8_{\pm1.0}$ | 96.6 | $5.7_{\pm0.7}$ | 92.3 | $1.4_{\pm0.8}$ |
| | RoBERTa | $31.1_{\pm0.7}$ | $54.2_{\pm1.0}$ | 99.0 | $1.1_{\pm0.3}$ | 99.0 | $1.0_{\pm0.3}$ |
| | LLh | $0.0_{\pm0.0}$ | $0.0_{\pm0.0}$ | 90.3 | $-0.0_{\pm0.3}$ | 90.4 | $0.0_{\pm0.0}$ |
| Kir. (0.5) | LLR | $0.0_{\pm0.0}$ | $0.0_{\pm0.0}$ | 85.4 | $1.5_{\pm0.4}$ | 75.3 | $-8.6_{\pm0.6}$ |
| | RoBERTa | $0.0_{\pm0.0}$ | $0.0_{\pm0.0}$ | 97.8 | $-0.0_{\pm0.1}$ | 97.8 | $0.0_{\pm0.0}$ |
| | LLh | $19.8_{\pm0.7}$ | $31.6_{\pm0.9}$ | 93.2 | $4.5_{\pm0.6}$ | 94.5 | $5.8_{\pm0.5}$ |
| Kir. (2) | LLR | $29.4_{\pm0.7}$ | $41.3_{\pm0.9}$ | 90.8 | $4.0_{\pm0.7}$ | 92.4 | $5.6_{\pm0.6}$ |
| | RoBERTa | $9.1_{\pm0.6}$ | $14.6_{\pm0.7}$ | 98.2 | $0.7_{\pm0.2}$ | 96.9 | $-0.5_{\pm0.3}$ |
| | LLh | $39.8_{\pm0.6}$ | $67.0_{\pm1.0}$ | 97.4 | $5.4_{\pm0.6}$ | 97.5 | $5.5_{\pm0.5}$ |
| Kir. (3) | LLR | $42.8_{\pm0.6}$ | $71.1_{\pm0.9}$ | 97.3 | $5.3_{\pm0.6}$ | 96.4 | $4.3_{\pm0.7}$ |
| | RoBERTa | $33.2_{\pm0.7}$ | $57.8_{\pm1.0}$ | 98.7 | $2.2_{\pm0.3}$ | 98.3 | $1.8_{\pm0.4}$ |
| | LLh | $37.5_{\pm0.7}$ | $54.9_{\pm1.0}$ | 98.2 | $7.4_{\pm0.5}$ | 97.3 | $6.5_{\pm0.5}$ |
| Bahri | LLR | $39.2_{\pm0.6}$ | $63.1_{\pm1.0}$ | 96.9 | $6.7_{\pm0.7}$ | 78.9 | $-11.4_{\pm1.0}$ |
| | RoBERTa | $32.1_{\pm0.7}$ | $42.9_{\pm0.9}$ | 98.7 | $1.3_{\pm0.3}$ | 98.4 | $1.0_{\pm0.3}$ |
| | LLh | $0.8_{\pm0.2}$ | $0.8_{\pm0.2}$ | 91.3 | $0.1_{\pm0.4}$ | 88.2 | $-2.9_{\pm0.5}$ |
| Kuditipudi | LLR | $2.6_{\pm0.3}$ | $2.7_{\pm0.3}$ | 85.0 | $-0.9_{\pm0.6}$ | 85.0 | $-0.9_{\pm0.5}$ |
| | RoBERTa | $0.3_{\pm0.1}$ | $0.4_{\pm0.1}$ | 98.0 | $-0.1_{\pm0.1}$ | 98.2 | $0.0_{\pm0.0}$ |

Table 23: Cascade hit rates and accuracies of MLP and Tree methods when GEMMA-7B-INSTRUCT is applied to the test set of *eli5-category* and MISTRAL-7B-INSTRUCT generations are used as negatives at a target length of 100 tokens. The trends here are similar to those for human negatives.

