# OpenReview forum: "Improving Detection of Watermarked Language Models"
_TMLR — Accepted by TMLR_

### Review · Reviewer_FJ8Y · 2025-10-08

**Summary Of Contributions:**

In this work, the authors propose to study the complementarity of passive detection methods -- i.e. forensics -- with active ones -- i.e watermarking -- in the case of LLM-generated text. To my knowledge, this hybrid approach has been scantily researched, with good reasons since the practical setting it entails is somewhat strange. I will discuss the significance of the approach later but I believe it does have its place and that such a study is welcomed in the field.

Since the subject matter:
-  is not aiming at SOTA performance with purely novel methodologies,
-  entails a peculiar detection setting,

I believe this work has its place in a venue such a TMLR which specifically targets such cases.
Since the paper is quite a straightforward empirical study of combining watermarking and forensics detector, I don't believe it requires a more lengthy summary. I will just say that the paper claims that an hybrid approach almost always improves watermarking-only methods which is quite surprising (see questions), and thus fully justifies a study.

Overall, the paper is very well-written, with complete and detailed experiments.  The last section summarizes the findings pretty well without overselling the approach. I feel the SOTA of both text watermarking and text forensics have been respected and correctly integrated in the work. I will also command the authors for the effort in providing all the elements needed for replicability.
In my opinion, the only main weakness of the paper is its reliance on a high FPR (1%), which I address in the rest of the review.
The question of the importance of the findings rests on the usefulness of the hybrid-approach which is not trivial as I will also discuss below.

References are provided in the additional comments.

**Additional Comments:**

References:

- [1] Giboulot, Eva, and Teddy Furon. "WaterMax: breaking the LLM watermark detectability-robustness-quality trade-off." NeurIPS 2024-38th Conference on Neural Information Processing Systems. 2024.
- [2] Fernandez, Pierre, et al. "Three bricks to consolidate watermarks for large language models." 2023 IEEE international workshop on information forensics and security (WIFS). IEEE, 2023.
- [3] Dathathri, Sumanth, et al. "Scalable watermarking for identifying large language model outputs." Nature 634.8035 (2024): 818-823.
- [4] Fernandez, Pierre, et al. "The stable signature: Rooting watermarks in latent diffusion models." Proceedings of the IEEE/CVF International Conference on Computer Vision. 2023.
- [5] Jovanović, Nikola, Robin Staab, and Martin Vechev. "Watermark stealing in large language models." arXiv preprint arXiv:2402.19361 (2024).

**Audience:**

Yes

**Audience Explanation:**

I believe the interest in these findings hinges on the following question: is the setting entailed by the hybrid approach useful in practice ?

The case for mixing watermarking and forensics is, in the best of worlds, somewhat of a nonsense. A watermark is designed to vastly outperform any forensics methods in exchange for a strong drawback: it needs to be able to modify the content in the first place. Forensics methods, on the other hand, work in the far more difficult setting of knowing **nothing** except the sample under scrutiny. Furthermore, watermarking algorithms, in a multi-bit setting, can also insert a payload in the watermarked content -- though I understand this is not the setting chosen by the authors. My point is that a good watermark design should preclude any hybrid-approach. Yet this paper makes the case that we are not there yet, especially in low-entropy settings.

 My counter-claim is two-fold:
	1. The low entropy, low text size, setting is generally of no interest to watermarking. Small answers generated by LLMs -- such as answers to quizzes or factual questions -- are not meant to be watermarked as they do not contain the "added-value" of the LLM. Even worse, the user can always truncate the answer to what interest them (no point in watermarking a proper name). Texts of interest are usually somewhat long (articles, essays, stories etc...) and provide plenty of room for the watermark signal to be very detectable even at low-entropy. I am ready to discuss this point with the authors since this claim is quite subjective and might reflect a bias in my vision of practical applications.
	2. The recent WaterMax scheme [1], given enough compute, should be able to reach an almost arbitrary power at any specified $P_{FA}$ . Granted, this assumes enough entropy such that at least two different tokens can be generated for most of the text. Yet, otherwise, this reinforces my qualms about the hybrid-approach: is it only useful for weak schemes in a very constrained setting which is low-entropy and low text-size?

As such, (see requested changes) if the authors can better motivate the use of such an hybrid approach, I believe the results of this paper are very interesting since **they challenge the key assumption that watermarking always provide better performance than forensics detectors**.

**Claims And Evidence:**

Yes

**Claims Explanation:**

This paper being purely empirical study, I will evaluate its merits based on two main criteria:

- The soundness of the experimental study: do the experiments allow me to validate/invalidate the thesis of the authors ? Is every parameter of importance
- The replicability of the study

### Soundness of the study

#### Structure
 First and foremost, I will command the authors on the overall structure of the paper. The main thesis of the paper is perfectly delineated: "It is possible to improve watermark detectors with forensics detector in the case of generated content detection". Along the thesis, a few quantities of interests are singled out as having an important impact on performance:
- the entropy of the model output provided the context
-  the length of the generated text
As pointed out by the authors, the entropy in particular is not well-studied in the case of forensics detector, this paper is thus a good reason to provide such a study. I will further highlight that despite the fact that these two quantities have quite a large impact on watermark performance, I still review far too many papers bypassing such study altogether.

#### Choice of attacks

The choice of attacks for text watermark still looks quite like the wild west in the field to me. I have been convinced by [1, Section 5] that the three main operations of deletion, addition and replacement of tokens provide a universal model of attack for the case of watermarking and that a curve of watermark performance as a function of the percentage of modified tokens is enough to completely characterize its robustness to attacks. The authors provide such a study in Figure 3. which is sufficient to me. The achievability of an attack for a fixed level of quality (provided a metric) is not a problem to solve by the defender, only by the attacker, since it quality should not have any impact on the "robustness curve" for the watermark.

Now obviously the question is different for forensics based detector since their performance directly depend on the distribution of the text for their detection. This is opposite to watermarking, which usually only depends on the distribution of the secret. In the former case, I question the use of a paraphrasing attack. Indeed, even though it is currently the most powerful attack against watermarking, that also preserves quality quite well in practice, it does make the text very amenable to forensics detector since it will follow the original, non-watermarked LLM distribution. The authors are aware of this as they (quite rightly) point it out in Section 3.6. To provide a fair comparison, I would have added a more naive attack which does not depend on an LLM, for example random replacement by synonyms which should somewhat preserve the quality of the text while not relying on any LLM.

#### Choice of metrics

The authors report two main "one-number" metrics to report for performance:
- A partial AUC -- computed under a FPR of interest
- Accuracy

I appreciate that the authors recognize the uselessness of studying regimes with very high FPR in the case of text detection -- as shown by the use of a pAUC.
Nevertheless, I am disappointed in the use of such a high FPR as 1%. I understand the authors work with a a "small" (~5000 samples is actually quite respectable) number of prompts and compute their FPR/TPR empirically on watermarker and detector scores.

To me, this goes against the philosophy of watermarking which should ensure high power at extremely low FPR ($ << 10^{-6} $ ). In the case of text, this is achieved by constructing a pivotal statistic on a well designed distribution of the watermarking signal (e.g. the red-green list score of Kirchenbauer). A well-designed scheme allows to specify exactly the distribution of the detector score under $H_0$ -- i.e. the non-watermarked case. This, in turn, allows to compute a sound $p$-value for the text under scrutiny -- allowing to reach extremely low **guaranteed** probabilities of false-alarm with usually very high power. A case in point for this approach is [2] -- which derives the correct statistical test for both Kirchenbauer's and Aaronson's schemes. In practice, it is also used with success in Watermax [1] and SynthID [3] (as the so-called "frequentist" detector) that the authors cite in this paper.
This is the reason why "active methods" such as watermarking are quite enticing compared to forensics methods: they allow to reach far better performance than passive methods, with strong  **theoretical guarantees** in terms of $P_{FA}$.  A recent, dramatic illustration of this, though in the different setting of images, is from Figure 3 in the Stable-Signature paper [4].


Each metric is computed as a function of: entropy, text length and percentage of corrupted text. These three choices are in my opinion necessary and sufficient for the evaluation of the method.

### Replicability

The LLMs are open-source.
All experimental details are provided with the training and testing set of forensics detector dully specified.
The methodology is clear and completely specified.
Error intervals are provided, including the methodology to compute them.

The only missing piece of information is the choice of hashing function for the Kirchenbauer and Aaronson watermarkers. I will assume that it is the baseline used in the original works (which is not a correct choice -- see [2] and [5] -- but is at least standard).

In conclusion, this should be a highly replicable paper.

**Requested Changes:**

- **High FPR, bad performance** To follow up on my discussion on $p$-values. I am quite surprised of the bad performance of both Kirchenbauer and Aaronson in this paper. Following Appendix K of [3], text of around 250 tokens should lead to good detectability at FPR around $ 10^{-3} $  under the chosen settings (though the Watermax paper uses $\gamma =0.5$  which might account for the difference). What I would like to know is: is it because an exact $p$-value is not used (which I would find bizarre), is it because of the low entropy setting? (Aaronson is especially dependent on high entropy). In any case, I would appreciate the authors computing the $p$-value for their scores and reporting them -- this should be very simple if the scores have been saved, by using the formulas in [2]. This is critical to securing my recommendation as this will explain the *apriori* low performance of the SOTA on the author's setting.

- **Hashing function**: Please provide the hashing function used for Kirchenbauer and Aaronson. I expect it to be the original one which has been shown to lead to be insecure and lead to unreliable scores in [2] and [5]. Though not necessary, I believe using a secure hash function that also guarantees the independence of token scores (at the cost of some robustness) would lead to a stronger work. At the very least, the hashing function needs to be spelled out in the appendix.

- **The setting**: I am only expecting the authors to defend their approach by answering my counter-claim. I do not necessarily ask for further experiments -- though a comparison with WaterMax or another scheme able to reach arbitrary performance would be welcome. In this case, the question would be, how much compute do I save by using an hybrid approach instead of WaterMax? This is critical to securing my recommendation, since, to the practitioner, an hybrid approach might seem counter-productive if the watermark is well-designed.

- **Synonym attack**: As said in the discussion, I would appreciate adding one more attack which does not clearly favor forensics detectors. This would strengthen the work in my view.

- **Choice of title**: The current title is very generic and does not convey anything of the paper's content. I would advise the authors to rename their work by making explicit the use of hybrid methods between forensics and watermarking. This would strengthen the work in my view.
- **Minor nitpicking**: p7 - an attack is always adversarial by definition, otherwise it is an augmentation. Attack is sufficient in this case.

---

> ### Author Response · Authors · 2025-10-24
> **rebuttal 1/2**
>
> Thank you for the substantial review – wow!
>
> **RE: Following Appendix K of [3], text of around 250 tokens should lead to good detectability at FPR around under the chosen settings**
>
> Firstly, to clarify, you mean Appendix K of [2] (page 27 of https://hal.science/hal-04766606/file/17001_WaterMax_breaking_the_LL.pdf), right?
> If so, it’s hard to make sense of it because we cannot find where the estimation of this quantity is actually defined. My understanding is that they’re reporting FPR@TPR=0.5. In the experimental description, they say they use 256 generations from 3 tasks, but also run “the detector on 100k wikipedia entries”. If the human samples from the 256 examples were used as negatives, then the lowest FPR they could be discussing is 1/256 = 0.004, but since we’re talking about even lower FPRs, it must be the case that they’re taking the entire negative class to be random 100k texts from wikipedia. For all the schemes we discuss (Aaronson, Kirchenbauer included) what influences performance is the amount of overlap in n-grams between the positive and negative text samples. The more n-grams they share, the worse the discrimination. In our work, we take human samples (and model generations) for the test prompts as negatives. These are going to share far more n-grams with the watermark model response than a random wikipedia snippet is – as a result, our discrimination task is inherently harder. When the negative class is another model, the number of common n-grams is even higher and the task is even tougher.
> So the TLDR is that because of our choice of using hard negatives, our task is harder and reported FPR is higher than would be the case if the negative class were random text snippets like wikipedia. This is a very important distinction to make and we can update the paper to explicitly point this out. With this consideration, a max FPR of 1% like the one we use is not all that high! It’s even what SynthID uses (see Figure 3 in the main text)!
>
> **RE: hashing function used for Kirchenbauer and Aaronson**
>
> For Aaronson:
>
> ```
> import itertools
> import hashlib
> import numpy as np
>
> def hash_list(values, private_key):
>     vals = itertools.chain([private_key], values)
>     m = hashlib.sha256()
>     code = bytes(''.join(map(str, vals)), 'utf-8')
>     m.update(code)
>     return int(m.hexdigest(), 16) % (2**32)-1
>
>
> def aaronson_prf(context_tokens, vocab_size, private_key):
>     seed = hash_list([int(t) for t in context_tokens], private_key)
>     generator = np.random.default_rng(seed)
>     return generator.uniform(0, 1, size=vocab_size)
> ```
>
>
> For Kirchenbauer, we use the most recent logic from their github codebase: https://github.com/jwkirchenbauer/lm-watermarking/tree/82922516930c02f8aa322765defdb5863d07a00e which includes self hashing.
>
> ```
> KB_FIXED_TABLE = np.random.default_rng(2971215073).permutation(1_000_003)
>
> def _kb_hashint(x):
>     return KB_FIXED_TABLE[x % len(KB_FIXED_TABLE)] + 1
>
> def kb_anchored_minhash_prf(input_ids, private_key):
>     input_ids = np.asarray(input_ids)
>     return (private_key * _kb_hashint(input_ids) * _kb_hashint(input_ids[-1])).min()
> ```
>
>
> **RE: the setting**
>
> Here is the way we see it: watermarking schemes can either be “mostly non-distortionary” or “distortionary”. Examples of the former are Bahri, Kirchenbauer with small $\delta$ bias, Kuditipudi, SynthId, Aaronson. Examples of the latter are: Kirchenbauer with very large $\delta$ bias, or even the following function (not one you would want to use, but a valid option nonetheless):
>
> ```
> def watermark(prompt, LLM):
>     return “hello world! I am a piece of watermarked text and I want to be detected.”
> ```
>
> These latter cases offer strong performance because they can embed the signal even when entropy is low, but they ruin the text. The less your watermarking scheme distorts, the worse the performance, especially in the low entropy regime, and relying on just watermarking becomes inadequate and can benefit substantially by a non-watermark detector. In fact, we note in the paper that “non-watermark detectors can outperform watermark ones in many cases. For example, under Aaronson, Binoculars and likelihood-based detection (LLh) achieve 93.1% and 91.7% accuracy respectively whereas watermark can only obtain 90.4%” Hybrid detection is necessary to overcome inherent limitations of low-distortion watermarks.
>
>
> RE: your point – “Small answers generated by LLMs -- such as answers to quizzes or factual questions -- are not meant to be watermarked as they do not contain the "added-value" of the LLM””
> Firstly, whether we should care about watermarking these is subjective and a question for the end developer. Regardless, many of the responses we consider here aren’t that short. Furthermore, while factual questions with short answers have low entropy, not all low entropy cases are short factual questions – one can find cases in more open-ended writing tasks like creative writing where the response is long but entropy is relatively low.

---

> ### Author Response · Authors · 2025-10-24
> **rebuttal 2/2**
>
> **RE: Synonym attack: As said in the discussion, I would appreciate adding one more attack which does not clearly favor forensics detectors.**
>
> This is a reasonable suggestion and we can add this in time permitting. I guess you are referring to the RoBERTa classifier as having an advantage for the paraphrasing attack? If so, note that the RoBERTa classifier can be trained to be robust / invariant to synonym attacks. The way these classifiers would be trained in practice is the detecting party will use a kitchen sink of binary labeled training data that include examples of all the expected forms of attack. We expect, apriori, for the classifier to do well test time on an attack type it got to learn during training – synonym attack included.
>
> **RE: Choice of title**
>
> Thank you, this is a good suggestion. If you have one in mind, let us know – we are open to it!
>
> **RE: Minor nitpicking**
>
> Good observation; we’ll change adversarial attack → attack. Thank you!

---

### Review · Reviewer_j18Y · 2025-10-09

**Summary Of Contributions:**

This paper investigates how combining watermark-based and non-watermark-based detectors can improve the detection of AI-generated text. The authors provide an extensive empirical study covering multiple watermarking schemes, several non-watermark detectors, and three hybrid strategies (two cascades and one learned fusion). The experiments are systematic and the results convincingly show that hybrid detection can significantly boost accuracy, especially in low-entropy conditions where traditional watermark detection is weak.

**Additional Comments:**

I recommend acceptance after major revisions focusing on addressing the above concerns.

**Audience:**

Yes

**Audience Explanation:**

The paper addresses a timely and practically relevant topic for the TMLR audience—how to improve provenance and authenticity detection for large language models. Researchers and practitioners in AI safety, watermarking, and responsible AI would find the results valuable. The study also complements recent work on watermark robustness and AI text forensics.

Although the conceptual novelty is modest, the empirical insights are important and broadly applicable, making it an interesting contribution for both academic and applied communities.

**Broader Impact Concerns:**

I do not see major ethical concerns in the presented work. The paper deals with AI-generated text detection and does not involve human subjects, sensitive data, or downstream decision-making. Its potential societal impact is positive: improving content attribution and AI transparency.

**Claims And Evidence:**

Yes

**Claims Explanation:**

The experimental results are thorough and well-documented. The paper evaluates a wide variety of watermark and non-watermark detectors under consistent settings, across multiple models and datasets. The evidence clearly supports the claim that hybrid detection improves performance under low-entropy conditions. Statistical comparisons and robustness analyses are convincing.

However, while the empirical evidence is strong, the paper lacks a deeper theoretical explanation of why certain hybrid combinations outperform others. The relationship between entropy and detectability remains observational rather than analytically grounded.

**Requested Changes:**

1. While the proposed hybrid detection framework (watermark + non-watermark) is executed carefully and demonstrates empirical value, its conceptual novelty is modest.
> Suggestions: 1) Introduce adaptive weighting mechanisms that automatically adjust the fusion parameters based on text entropy, token length, or detector confidence. or 2) Provide a mathematical formulation of the decision boundary learned by the logistic regression model to highlight what it learns beyond heuristic thresholding.

2. The study assumes white-box (1P) access to model logits or watermark keys, which limits its generality.
In real-world applications such as platform moderation, media provenance, or forensic analysis, detectors typically operate in a third-party (3P) or black-box setting, without access to internal model information.
> Suggestions: Extend the framework to 3P-compatible detectors that use only surface-level statistics or model-agnostic proxies (e.g., perplexity ratios).

3. The hybrid models (especially logistic regression) are calibrated on a dataset different from the test domain (ELI5 vs. Dolly), implicitly assuming cross-domain consistency. This calibration dependency raises questions about robustness to domain shifts, prompt distribution changes, or stylistic differences in text.
> Suggestions: 1) Conduct explicit cross-domain generalization experiments (e.g., calibrate on Dolly, test on news or social media text). 2) Quantify the sensitivity of hybrid thresholds to calibration data by measuring AUC variance under dataset perturbations.

4. The paper empirically confirms that detection improves with higher entropy prompts, but provides no formal analysis linking entropy to detection capability. This remains an empirical observation rather than a theoretical understanding.
> Suggestions: 1) Provide a probabilistic model describing how text entropy bounds the achievable ROC-AUC of watermark detectors. 2) Analyze mutual information between watermark scores and non-watermark features to quantify their complementarity.

5.  Attack evaluation is narrow: paraphrasing uses only Gemini-1.5-Pro with a single paraphrase intensity; random token replacement is synthetically defined and may not reflect realistic obfuscation.

6. Negative samples are limited to “human” or “different-model” text; mixed human–AI co-authored text, which is prevalent in practice, is not tested.

7. Although cascade “hit rate” (γ) is reported as a proxy for computational savings, the paper does not quantify real-world cost metrics such as FLOPs, latency, or energy use. This makes it difficult to assess trade-offs between detection accuracy and efficiency.

---

> ### Author Response · Authors · 2025-10-24
> **rebuttal 1/2**
>
> Thank you for the detailed review!
>
> **RE: Introduce adaptive weighting mechanisms**
>
> This is something we did think about, but here are some reasons against it. Regarding text entropy – remember that this quantity is actually latent during inference time. In order to estimate it for a text y, we need to know the prompt x that the user had issued to get y – we need $P(\cdot | x)$, which we generally don’t have for this task. Regarding non-watermark detector confidence, this is already used in the models – for example, the decision tree splits on the value of the confidence, possibly multiple times. In general, our findings (and the reason why logistic regression generally outperforms decision tree) is that it’s pretty easy to overfit to the calibration or training data. Unless the user was confident that the test distribution of text is very similar to that of available training data, adding more parameters or making the model complex via adaptive weighting will very likely hurt performance.
>
>
> **RE: extend to 3P-compatible detectors**
>
> Could you further elaborate on exactly what you mean? The scope of our work is how to best combine watermark and non-watermark signals together, and since (AFAIK) it’s almost always the model owners (i.e. 1P) that have the necessary watermark key needed to compute the watermark signal, we just say that the paper is about 1P detection for simplicity. The paper is still applicable if the watermark keys are shared with the 3P content moderator. The non-watermark detectors applicable in this case are those that we explicitly call out as being 3P – Binoculars (which uses a ratio involving perplexity-like quantities), RoBERTa classifier, and RADAR (likelihood and rank cannot be as they involve inference on the model).
>
> **RE: cross-domain consistency**.
>
> You are correct that the hybrid models are calibrated on a dataset different from the test one. This was intentional since this is the most likely scenario in a real world deployment – practitioners will tune their detection on whatever training data is available, which may or may not be similar to what is seen at test time. Do note that the RoBERTa classifier is trained on the eli5 train set – when the classifier is evaluated on its own on eli5 test (generator = Mistral-7B-instruct, watermark = Aaronson), it achieves 94.4% accuracy (table 7) whereas it drops to 62.1% (table 5) on databricks-dolly-15k. One thing we could do for a subset of experimental settings is: take the test dataset (say dolly15k), and compute a cross-validated accuracy score, where we partition the test into 4 splits, calibrate on one, test on the rest – do this 4 times and average. This will provide some insight opn how the cascade does when tested exactly on the same data distribution it’s been calibrated on. We’d probably put this in the Appendix since we think it’s less representative of real world deployment though. What do you think?
>
> **RE:  no formal analysis linking entropy to detection capability**
>
> Regarding your first suggestion – the connection between entropy and watermarking detection performance is well established. For example, see Theorem 4.2 in “A Watermark for Language Models”, Bahri et. al (https://arxiv.org/pdf/2410.02099) for the Bahri baseline we use in the paper.
>
> The goal of the work isn’t in providing performance guarantees of watermarking schemes – it’s in the observation that non-watermark detectors can step in and be helpful in the low entropy regimes where watermarks are known to struggle.
> Regarding your second suggestion – computing the mutual information between the two scores is a reasonable thing to do (e.g. using `sklearn.feature_selection.mutual_info_regression` since both are continuous in nature), but while this quantity is meaningful and interesting, what we really care about at the end of the day is how the presence of the two scores affects prediction of the label (watermarked or not watermarked) vs each one individually, and that’s what the current experimental setup is directly measuring. If using both scores don’t add any power to predicting the label, then the performance of logistic regression with 2 features should not improve over that with one feature.

---

> ### Author Response · Authors · 2025-10-24
> **rebuttal 2/2**
>
> **RE: Attack evaluation is narrow**
>
> What would you suggest? We based our choice of attacks based on prior work – Bahri et. al (https://arxiv.org/abs/2410.02099) and Kuditipudi et. al (https://arxiv.org/abs/2307.15593) – both evaluate these paraphrasing and random token attacks. We could try roundtrip translation (e.g. English -> Spanish -> English) but apriori we expect it to be roughly similar to paraphrasing performance.
>
> **RE: mixed human–AI co-authored text, which is prevalent in practice**
>
> This is a good point and something we were aware of at the time of writing. We agree that many real world text samples may be AI-assisted human samples, or human-edited AI samples. We think this can make for an interesting extension for future work – we like that the existing evaluation is black and white and we can treat the problem cleanly as binary classification. Mixed text complicates this because, for example…do you treat all the mixed text as a third class? If so, then a sample that’s 99% AI and 1% human will be treated the same as one that’s 99% human and 1% AI. If you treat “mixedness”on a continuum, then the task becomes more regression-oriented (e.g. predict the fraction of AI text) than classification, and then our experimental setup will need to be extended substantially, and we feel the paper is already a good length.
>
> **RE: Although cascade “hit rate” ($\gamma$) is reported as a proxy for computational savings, the paper does not quantify real-world cost metrics such as FLOPs, latency, or energy use.**
>
> This is a good point and one we were expecting from reviewers. Here’s the thing – wall clock time execution speed, energy use, and real world FLOPs are all strongly implementation dependent – code written for research purposes and code written optimized for production will give different numbers, but hit rate is implementation agnostic. Practitioners can take the FLOPs they’ve measured for their watermarking scheme and non-watermarking detector and just use the hit rates to estimate the overall footprint of the hybrid approach. Watermark detection is usually so fast that it’s negligible to that of inference on a RoBERTa detector. In the paper we say “For Aaronson, Bahri, and Kirchenbauer, $\gamma$ generally hovers around 20-40%. $\gamma$ under Kuditipudi is very low since its watermark detector is not much better than random. If the cost of watermark detection is negligible compared to its counterpart, as it often is, then cascading improves non-watermark computational efficiency by 20-40%.”

---

### Review · Reviewer_GCTQ · 2025-10-10

**Summary Of Contributions:**

This work proposed a hybrid approach that ensembles watermark and non-watermark detectors to handle the low-entropy prompt where the watermark is unreliable. The experiments show that across watermark schemes and detectors on two datasets, hybrid approaches consistently beat either alone. The authors also show the robustness performance under two attacks.

**Audience:**

Yes

**Audience Explanation:**

AI detection is an important topic, and this paper explores a hybrid of watermark and non-watermark detectors which is of interest to many in the TMLR audience

**Claims And Evidence:**

Yes

**Claims Explanation:**

- The experiment is extensive, including a mixture of multiple watermark methods and non-watermark detectors, trying multiple hybrid approaches, and evaluating robustness. The authors also present massive results in the appendix.
- The summary of experimental findings in Section 4 is insightful and easy to follow.
- The proposed method is well-motivated.

**Requested Changes:**

- The ensemble method is modest. The author is encouraged to explore this aspect more deeply, such as why *Weaker non-watermark detectors can offer stronger performance in a hybrid setup.*
- The performance comparison between watermarking methods and detectors on low-entropy prompts could be presented in the introduction to more clearly highlight the motivation.
- The adversarial attack evaluation is relatively weak. Stronger methods [1,2] should be considered or at least discussed. For example, RoBERTa achieves high performance (85–90% accuracy) on paraphrase attacks.
- Some important experimental results should be moved to the main paper instead of the appendix, such as overfitting. A concise summary table could be used to present these results clearly.

[1] Paraphrasing evades detectors of AI-generated text, but retrieval is an effective defense. NeurIPS 2023.

[2] Large Language Models can be Guided to Evade AI-generated Text Detection. TMLR 2024.

---

> ### Author Response · Authors · 2025-10-24
> **rebuttal**
>
> Thank you for the review! Addressing your concerns one by one:
>
> **RE: “Weaker non-watermark detectors can offer stronger performance in a hybrid setup.”**
>
> Concretely, what further analysis do you think would be insightful here? In the paper we say the “RoBERTa classifier provides more complementary signal than the other approaches. One factor here is its strong performance in low entropy regimes”. Why this is the case may be hard to pinpoint, but if you have suggestions, let us know.
>
> **RE: Motivate introduction with results for low-entropy prompts**
>
> What do you think about updating a paragraph in the introduction as follows:
> “We show how non-watermark detectors, when configured in a hybrid setup, can substantially
> bolster watermarking in low entropy regimes. We study how these two approaches to detection…"
> →
> “We show how non-watermark detectors, when configured in a hybrid setup, can substantially
> bolster watermarking in low entropy regimes. For example, in Figure 1 we see how watermark detection accuracy on databricks-dolly15k is poor when entropy is low. When combined with a non-watermark detector in the hybrid setup we propose, the accuracy remains consistently high across entropy levels. We study how these two approaches to detection…"
>
> **RE: adversarial attack evaluation is weak**
>
> Thank you for bringing up these two references. We will add a discussion of them to the paper.
> RE: “Paraphrasing evades detectors of AI-generated text, but retrieval is an effective defense” – Here, the DIPPER paraphrasing model isn’t something that was trained explicitly to evade detection – it’s more or less a regular paraphrasing model (focused on maintaining quality paraphrasing) and so we think using Gemini 1.5 pro as we do is an appropriate substitute.
> RE: “Large Language Models can be Guided to Evade AI-generated Text Detection.” – this is an interesting approach. The authors propose a way to find prompts that lead to undetectable generations. This search, or optimization, is a little involved and requires use of a proxy detector, etc.
>
> **RE: moving results like overfitting to the main paper**
>
> Thank you for this suggestion. We can certainly move these results to the main text. Which results / metrics (besides examples of overfitting) do you think are most important to be moved over?

---

> ### Comment · Reviewer_GCTQ · 2025-10-24
>
> Thanks for authors' response.
>
> - Regarding the analysis, one possible approach is to train a weaker non-watermark detector to help validate the findings. However, I would like to clarify that I’m not specifically focused on this experiment, as indicated by the phrase “such as” in my original review. My main concern is that **the proposed method appears rather modest.** To strengthen the contribution of the paper, I recommend conducting more in-depth experiments.
>
> - For the performance comparison, I suggest presenting the results in a table or figure rather than only in text.
>
> - I appreciate the addition of a discussion on adversarial attack evaluations.
>
> - As I mentioned in my initial review, a concise summary table would be useful for clearly presenting the results of the moving scenario.
>
> I suggest the authors work on this themselves using the feedback, rather than asking reviewers for detailed step-by-step guidance.

---

> > ### Author Response · Authors · 2025-10-24
> >
> > Thanks for the quick response.
> >
> > **RE: "one possible approach is to train a weaker non-watermark detector to help validate the findings", "proposed method appears rather modest", and "more in-depth experiments"**
> >
> > I think there may be some misunderstanding. What do you mean here? What is modest? What would training a weaker non-watermark detector accomplish? The point we are making in that paragraph is that just because non-watermark method A is better than non-watermark method B when evaluated in isolation, when configured in a cascade or another hybrid setup B may provide more boost than A, and this is determined by how correlated and complementary the two signals are.
> >
> > **RE: performance comparison**
> >
> > The results are in Figure 1 and we are referencing it in this revised intro paragraph? Do you think we should do something else?
> >
> > **RE: As I mentioned in my initial review, a concise summary table would be useful for clearly presenting the results of the moving scenario.**
> >
> > Thank you. We had asked you what results you would like to have moved over? What do you mean by "results of the moving scenario"? What is the "moving scenario"?
> >
> > **RE:  authors work on this themselves using the feedback, rather than asking reviewers for detailed step-by-step guidance**
> >
> > With all due respect, we would like to improve based on feedback, and we asked clarifying questions about the feedback because we're confused by what you mean exactly.

---

### Decision · Action_Editor_uZYo · 2025-12-08

**Recommendation:** Accept with minor revision

**Additional Comments:**

In addition to incorporating their responses to the reviewers’ feedback, I ask the authors to also address the following requests in the final version of their manuscript:
1. Following reviewer GCTQ’s recommendations: please include a table or figure summarizing the key points discussed in Section 4, and highlight your findings on low-entropy prompts in the introduction to better motivate the paper.
2. Following reviewer j18Y, please include a discussion about:
- Adaptive Watermark Removal Attacks — Evaluate attacks explicitly targeting watermark traces (e.g., entropy-smoothing paraphrase, reinforcement-guided rewriting, or gradient-free methods such as Warfare and Warfare-plus).
- Detector-Evasion Attacks — Test adversarial text-generation techniques that minimize detection confidence (e.g., surrogate-detector optimization, RLHF-based evasion, or black-box paraphrasing with semantic constraints).
- Adaptive Evaluation Protocol — Define attacker knowledge levels (white-box, gray-box, black-box) and assess whether the hybrid detector maintains complementary robustness when one subsystem is compromised.

**Audience:**

Yes

**Audience Explanation:**

Detection of LLM generated content is a timely and important topic. The main conclusions of the paper should be of significant interest to the community, as they challenge assumptions about the sufficiency of watermarking and highlight weaknesses in this class of methods.

**Claims And Evidence:**

Yes

**Claims Explanation:**

Most reviewers agree that the paper presents a modest but solid and well-executed contribution. The key strength highlighted by all reviewers is the consistent empirical finding that hybrid detection frameworks, combining watermark-based and non-watermark detectors, outperform either method alone, especially in low-entropy settings where watermarking becomes unreliable. Reviewers also praise the careful experimental design and overall high quality of the writing and reproducibility.

The general opinion therefore leans toward acceptance with revisions. I concur with this assessment and recommend acceptance pending the revisions detailed above.